

# Algorithms for Optimization of Branching Gravity-Driven Water Networks

Ian Dardani[1], Gerard F. Jones[1]

[1]Villanova University, College of Engineering, Villanova, 19085, United States

*Correspondence to*: Ian Dardani (ian.dardani@villanova.edu)

**Abstract.** The design of a water network requires the selection of pipe diameters that satisfy pressure and flow requirements while optimizing for cost. This work focuses on the design of moderate-scale branching Gravity-Driven Water Networks (GDWNs), in contrast to large urban-scale looping networks, where budgets are highly constrained and where PVC pipe is typically used. In order to help designers of GDWNs select an appropriate design approach for a given network problem,

three cost-minimization algorithms are developed and compared on five GDWN test cases. Two algorithms, a backtracking algorithm and a genetic algorithm, use a set of discrete pipe diameters, while a new calculus-based algorithm produces a continuous-diameter solution, which is mapped onto a discrete-diameter solution. The backtracking algorithm produced the overall lowest-cost solutions with relative efficiency for the test cases, while the calculus-based algorithm produced slightly higher-cost results but with greater scalability to networks with more links. Furthermore, the new calculus-based algorithm's

continuous-diameter and mapped solutions provided lower and upper bounds, respectively, on the discrete-diameter global optimum cost, where the mapped solutions were typically within one diameter size of the global optimum. Overall, the genetic algorithm as implemented did not produce results, which deemed it compelling over deterministic methods as applied to GDWNs. However, for more complex networks and problem formulations, a genetic algorithm may be more advantageous, particularly if it incorporates improvements reported in the literature.

## 1 Introduction

A gravity-driven water network (GDWN) is commonly constructed to deliver potable water to a community in a developing region. These systems draw water from a source at a high elevation, such as a natural spring or a stream, and deliver it through a branching pipe network to household taps or public tapstands (Fig. 1). In principal, loops and loop/branching constructs may be added to networks for greater reliability, but material cost considerations often restrict

attention to just branch networks in GDWNs. The methodologies presented in this paper, however, may be extended to all networks, including those with loops. When feasible, gravity water networks are very attractive compared with pumped networks because of their simplicity and lower capital, operational, and maintenance costs. In addition, in most locations where GDWN are considered, there may be little or no access to reliable grid-based electrical power for pumps.

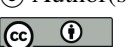



Water networks are modelled as a collection of nodes, each representing a point of water demand or supply, which are connected with links representing pipes. Typically, the layout of the site is known, including water source and demand locations and elevations of all other nodes. For the present work, design flow rates are determined from community survey data, which are extrapolated for future population growth. Networks in this category are referred to as "demand-driven"

designs. Thus, to design a network of this type, pipe diameters for each link must be chosen such that acceptable but arbitrary minimum pressure heads are maintained at each node given a design flow rate at the node. Furthermore, application of the energy equation to this network demonstrates that the design problem is non-unique; i.e., choosing different pressure heads at the nodes will result in a different pipe diameter solution for the network, and thus different networks costs.

In practice, gravity-driven water networks are commonly designed by a marching method, where diameters for each

link of the network are chosen sequentially. After selecting a reasonable diameter for each link, the designer calculates the static pressure head at the link outlet, and proceeds to the next link if this result is acceptable. In this way, the designer marches through the network until all pipe diameters have been selected. This method produces a feasible solution, but not a cost-optimized one. As noted by Bhave (2003), cost savings of 20-30% can result from the use of optimization techniques. In developing regions, the cost of a water network can be prohibitive, adding to the importance of optimizing network design.

Within the provided framework, the global optimum can be found through an exhaustive search of the solution space, known as complete enumeration, although this is infeasible when considering networks with many links and diameter choices (Kadu et al. 2008; González-Cebollada 2011). To reduce the computational time required by enumeration, authors have proposed methods to prune the search space (Kadu et al. 2008), although some of these techniques may remove the global optimum (Simpson et al. 1994). The most common types of algorithms that have been applied to optimize water network

design include deterministic methods, heuristic methods, metaheuristic methods, multi-objective methods, and decomposition methods (Zhao et al. 2016).

Deterministic methods include linear programming (LP), dynamic programming, and nonlinear programming (NLP), and typically involve rigorous mathematical approaches (Zhao et al. 2016). A brief overview and comparison of these algorithms is given in Kansal, et al. (1996), who use a single-part cost correlation for metric pipe diameters between 100 mm

and 350 mm. Linear programming techniques have relatively low computational complexity and allow each link to be composed of two diameters, called a split-pipe solution, although these may not always be practical to implement (Kessler and Shamir 1989, Swamee and Sharma 2000, Somani and Mottaghi 2006). LP can also get stuck in a local optimum (Zhao et al. 2016), although combining LP with metaheuristic techniques can help with the problem's non-smoothness properties (Krapivka and Ostfeld 2009). Dynamic programming has been used by Yang et al. (1975) and Martin (1980) to optimize

networks in stages. This approach begins at the discharge nodes, proceeding to select feasible diameters and joints for upstream stages and storing these partial candidates in memory until the source node is reached. At this point, the algorithm reviews the feasible segment design options and selects a combination of stage solutions producing the lowest cost overall solution. This method, however, requires the designer to allow a relatively narrow range for the design pressure of each node, or otherwise





store a large set of feasible candidate solutions in memory and also allow adjoining branches to arrive at different heads at the same node.

Nonlinear programming, a calculus-based method, deals with each link's diameter as a continuous variable. Using Lagrange multipliers and a one-part, pipe-cost model with minor-lossless flow, Swamee and Sharma (2000) developed systems of equations for both continuous and discrete pipe diameters for branch networks, assuming constant friction factor. When solved, the solution gives D values that minimize distribution main cost, not network cost. In carrying out the solution, iteration is required to update the value of the friction factor. For the discrete diameter case, large computational times were noted by Swamee and Sharma because of the stiffness of the mathematical system. Cases where one or more nodal pressure heads are not acceptable need to be treated manually by the designer in various ways as discussed by the authors.

For branching networks, Jones (2011) showed that by restricting the focus to smooth-turbulent, minor-lossless flow, and the use of a one-part, pipe-cost model, a simple nonlinear algebraic equation for each internal node in the distribution main could be developed. When solved simultaneously with the energy equation for each link, a unique solution for D and nodal pressure head values are obtained that produces minimum network cost, as opposed to the distribution main cost as in Swamee and Sharma (2000). The method of Jones (2011) also applies to serial networks.

Heuristic methods follow specific rules to incrementally build better solutions, although the rules are not strictly formulated to trend towards local or global optima. An approach by Monbaliu et al. (1990) sets all network pipes to their minimum size, where the pipe that has a maximum head loss gradient is incremented to its next-highest size until all nodal head requirements are satisfied. Similarly, an algorithm by Keedwell and Khu (2006) selects an initial solution and iteratively responds to nodal head deficits and surpluses by incrementing or decrementing pipe sizes accordingly until a feasible solution is found. Suribabu (2012) proposed a heuristic that identifies pipes to increment or decrement in size based on flow velocity and alternative metrics such as proximity to the source node, achieving acceptable cost results with computational efficiency. While these algorithms are typically computationally efficient, they do not guarantee a global optimum.

Metaheuristic optimization methods allow for a set of solutions to evolve through random processes that are guided with an objective function which rewards low network costs and penalizes hydraulic insufficiencies. Examples include evolutionary algorithms, which are most commonly genetic algorithms (Krapivka and Ostfeld 2009, Simpson et al. 1994, Kadu et al. 2008, Prasad and Park 2004), simulated annealing (Vasan and Simonovic 2010; Tospornsampan et al. 2007), ant colony optimization (Maier et al. 2003), and differential evolution (Vasan and Simonovic 2010). As reviewed by Nicklow et al. (2010), evolutionary algorithms are an emerging popular alternative to the deterministic methods, and they offer the opportunity to accommodate unique constraints and multiple design objectives. The main challenges for evolutionary algorithms are the difficulty of incorporating constraints into objective functions, the optimum selection of parameters, and a relatively large amount of computational effort. In addition to optimizing for cost, multi-objective methods, often based on evolutionary algorithms, allow the designer to choose from a Pareto-optimal front of objectives, such as cost and reliability (Prasad and Park 2004).



Decomposition methods involve the partitioning of networks into smaller sub-networks which are each optimized using one of many types of techniques and then combined into an overall solution. In some cases, the loops in the sub-networks are removed, producing branching trees which are then optimized individually. Techniques used to optimize the sub-networks can involve multiple methods, including linear programming (Saldarriaga 2013) and differential evolution (Zheng et al. 2013),

with a later stage optimizing the network as a whole using the sub-network solutions as inputs. Note that another distinct use of the term 'decomposition' refers to the approach of iteratively solving "inner" and "outer" mathematical problem formulations, and has been used in the literature by Krapivka and Ostfeld (2009) who traces its use in this context back to Alperovits and Shamir (1977).

In the present study, we compare and contrast the results of the calculus-based (CB) optimization model of Jones

(2011), which is an NLP algorithm, for continuous D, minor-lossless, smooth-turbulent flow with two other discrete-diameter models, including backtracking (BT), a recursive partial enumeration algorithm, and a genetic algorithm (GA) as applied to GDWN design. Backtracking has not been widely utilized in the WDN literature, with the method from González-Cebollada et al. (2011) representing the most similar approach to the present study's BT algorithm. Unlike the González-Cebollada algorithm, the BT algorithm in the present study guarantees a global optimum by continuing its search of the solution tree even

after the first solution has been found. The CB algorithm optimizes the network design with continuous diameter choices that are then mapped onto diameters from a discrete set of those commercially available. BT and GA directly utilize the discrete set of diameters. For a direct comparison of techniques, pipe costs for the calculus-based algorithm are found by interpolating a two-part cost formula based on a curve-fit of real cost data for available diameter values. The three models are tested against field data from five actual GDWNs installed in Panama, Nicaragua, and the Philippines.

Within the broader context of water network problem formulations, this paper is concerned with single-objective material cost optimization of single-source, branching water distribution networks with steady-state demands and pre-specified pipe locations. By implication of being gravity-driven, the problem does not involve the use of pumping stations. This problem formulation is directly applicable to typical gravity-driven water networks, and is also useful for multi-objective algorithms, the consideration of sub-networks in a decomposition technique, pumped networks, and looped system optimization, which

can involve reformulating the problem into a branching configuration.

The results of this study highlight the advantages and weaknesses of each GDWN design method including computational time, scalability, closeness to the global optimum, and features like the ability to prune the solution space of infeasible and sub-optimal candidates without missing the global optimum. We also extend the Jones closed-form model to include minor losses, a more-comprehensive two-part cost model, which realistically applies to pipe sizes that span a broad

range typical of GDWNs of interest in this work, and for smooth and commercial steel roughness values.



## 2 Problem Formulation

Branching networks are considered (Fig. 1), where all branches connect a distribution main node with a delivery node, shown as tapstands or houses. For each link in a network of $N_L$ links, pipe length ($L$) and the net elevation change ($\Delta z$) are considered fixed. Steady-state flow rates ($Q$) are prescribed for each link based on the demand flow data at delivery nodes.

As noted above, demand flows are determined by community surveys and extrapolated in time to quantitatively account for population growth. Minor losses are accounted for through a minor loss coefficient $K$ or a dimensionless equivalent pipe length, ($L_e/D$, or in symbol form, $L_{ebyd}$), where $L_e$ is the pipe length of diameter $D$ whose frictional loss results in the corresponding minor loss. An optimal solution is obtained by selecting pipe diameters ($D$) from a set of commercially available diameters such that the network's material cost is minimized. With $N_D$ choices of diameters for $N_L$ links, the problem has $N_D^{N_L}$

candidate solutions.

For all nodes, static pressure, $h$, is greater than or equal to a chosen minimum, $h_{min}$. The value for $h_{min}$ is selected to eliminate possible leakage of contaminated ground water into the network should the operating conditions change in an unanticipated way. The change in static pressure head, $\Delta h$, across each link is calculated with the energy equation for pipe flow,

$$\Delta h = -\Delta z + \left( \alpha + K + f\left( \frac{L}{D} + L_{ebyD} \right) \right) \frac{8Q^2}{\pi^2 g D^4} \tag{1}$$

where for each link, $\alpha$ is the kinetic energy correction factor and $f$ is the Darcy friction factor, calculated with the Colebrook-White equation (Colebrook and White 1937) or Churchill correlation (Churchill 1977), and $g$ is acceleration of gravity. The kinetic energy correction factor, $\alpha$, is considered only in the first link, where acceleration from a zero-velocity source is sometimes non-negligible for the smallest of GDWNs that have been encountered. Thus,

$$\alpha = \begin{cases} 2 & \text{Re} \leq 2300 \\ 1.05 & \text{Re} > 2300 \end{cases}$$

where Re is the Reynolds number for pipe flow, $4Q/\pi \nu D$, and $\nu$ is the kinematic viscosity of water. The possibility of laminar flow (Re $\leq 2300$) is permitted since branches from the smallest GDWN observed in practice have been in this regime.

The pressure upper bound is not incorporated into the optimization process. Worst-case pressure conditions occur under hydrostatic conditions, which are directly related to the maximum elevation change in the network and where no flow occurs. Therefore, the selection of appropriate pressure ratings and, if needed, break-pressure tanks are left to the correct

judgment of the designer under no-flow conditions. In addition, precautions against water hammer are left to the designer.

## 3 New Calculus-Based Algorithm

In this section we develop a new calculus-based algorithm for pipe diameters that minimize overall pipe cost for the network. First appearing in the text by Jones (2011), this algorithm is solved simultaneously with the energy equation for each



link to produce unique solutions for $D$ and nodal pressure head values that minimize network pipe cost, as opposed to only the distribution main cost as in Swamee and Sharma (2000). The method also applies to serial networks.

First consider the physical basis for the existence of a unique set of pipe diameters and static pressures for the demand-driven design problem with cost minimization included. Several works reviewed in the previous section have considered

optimization of GDWN and combined pumped and gravity-driven networks. We assume continuous pipe diameters in this section; values that result from the solution of the energy equation. Mapping between continuous diameters and the discrete nominal sizes, required to complete the design, will not be addressed in the present work.

Consider the three-pipe network shown in Fig. 2. Pipes 1-2, 2-3, and 2-4 meet where head $h_2$ is unknown. Each pipe has prescribed volume flow rate and length and unknown diameter $D$ as shown. The change in elevation between the top and

bottom of each pipe is $\Delta z$ and $\Delta h$ is the change in static pressure head. There is a prescribed head at each outlet for pipes 2-3 and 2-4.

To facilitate insight, we at first assume turbulent flow, which can be verified post-calculation if necessary, ***in smooth pipe*** and that minor losses are negligible. Two sources for the friction factor for smooth-turbulent flow are considered, namely the classical Blasius equation (reported in Streeter et al. 1998), $f = 0.316\,\mathrm{Re}^{-1/4}$, and the Swamee-Jain correlation (Swamee

and Jain 1976), $f = 0.175\,\mathrm{Re}^{-0.1923}$ (though not explicitly appearing in this reference, $f$ from the Swamee-Jain correlation is obtained by writing it for smooth pipe and comparing this with the energy equation, where $f$ is assumed to be in the form $a\,\mathrm{Re}^n$). The Blasius equation has higher accuracy (2% / 3%) in the range $10^4 < \mathrm{Re} < 10^5$, over which most of the GDWNs in this work operate, compared with the Swamee-Jain correlation of +8% / -3%, thus prompting the Blasius equation to be chosen for this work. A combination of the Blasius equation with the energy equation gives explicit formulas for $D$ for the three links

in Fig. 2. For simplicity, and to reduce the number of free parameters, the conditions for pipes 2-3 and 2-4 are assumed to be identical without loss of generality. We obtain

$$D_{12} = 0.741\,(\frac{\Delta z_{12} + \Delta h_{12}}{L_1})^{-4/19}(\frac{Q_{12}\,\nu^{1/7}}{g^{4/7}})^{7/19}$$

$$D_{23} = D_{24} = 0.741\,(\frac{\Delta z_{23} + \Delta h_{23}}{L_2})^{-4/19}(\frac{Q_{23}\,\nu^{1/7}}{g^{4/7}})^{7/19} \tag{2}$$

With our assumptions and inspection of Fig. 2, $\Delta h_{12} = -h_2$ and $\Delta h_{23} = \Delta h_{24} = h_2 - h_3 = h_2 - h_4$, obtain

$$D_{12} = 0.741\,(\frac{\Delta z_{12} - h_2}{L_1})^{-4/19}(\frac{Q_{12}\,\nu^{1/7}}{g^{4/7}})^{7/19}$$

$$D_{23} = D_{24} = 0.741\,(\frac{\Delta z_{23} - h_3 + h_2}{L_2})^{-4/19}(\frac{Q_{23}\,\nu^{1/7}}{g^{4/7}})^{7/19} \tag{3}$$

The pipe cost model can be assumed to follow a power-law relationship (Swamee and Sharma 2008)

$$C' = a\,(\frac{D}{D_u})^b \tag{4}$$





where $a$ is a constant coefficient, $b$ is a constant exponent, and $D_u$ an assumed unit diameter. A more robust, two-part model, valid for a greater range of pipe sizes than that of Swamee and Sharma (2008), will be used below. The use of pipe material cost as the objective function was assumed because of relevance. In most GDWNs of interest in this work, installation labor comes from the local community and has no well-defined associated cost. The material cost for the network is of prime importance since it normally comes from funds raised by nongovernmental organizations or grants, where there is seldom a required repayment but are always in short supply. For a more-general case, the economics of a GDWN may be more encompassing and include materials, labor, operation and maintenance, depreciation, taxes, and salvage, among others. The time value of money may also need to be considered, which includes interest rates and estimation of the network lifetime.

With Eq. (4) the general expression for the total cost for the pipe material, $C_T$, is obtained by summing over all links $ij$,

$$C_T = a \sum_{ij} L_{ij} \left(\frac{D_{ij}}{D_u}\right)^{b} \tag{5}$$

which, for the present problem, becomes

$$
\begin{aligned}
C_T &= a \left[ L_{12}\left(\frac{D_{12}}{D_u}\right)^{b} + L_{23}\left(\frac{D_{23}}{D_u}\right)^{b} + L_{24}\left(\frac{D_{24}}{D_u}\right)^{b} \right] \\
&= a \left[ L_{12}\left(\frac{D_{12}}{D_u}\right)^{b} + 2L_{23}\left(\frac{D_{23}}{D_u}\right)^{b} \right]
\end{aligned}
\tag{6}
$$

A close inspection of Eq. (3) in combination with Eq. (6) will reveal the origin of the existence of an optimal $h_2$ for the design of the network in Fig. 2. Because of its arbitrariness, we are free to vary the value of $h_2$. As $h_2$ increases, say from a small value like 1 m, the pressure difference between the junction and the bottom of pipe 2-3 (and 2-4) increases. Since the volume flow rates in each pipe are fixed, an increase in pressure drop across pipe 2-3 (and 2-4) requires a reduction in $D_{23}$ (and $D_{24}$). This is evident from our inspection of the second of Eq. (3), where we see that $D_{23}$ and $D_{24}$ are both proportional to $(\Delta z_{12} - h_3 + h_2)^{-4/19}$; that is $(\Delta z_{12} - h_3 + h_2)^{-4/19}$ *decreases* as $h_2$ increases.

Because the top of pipe 1-2 is at atmospheric pressure, an increase in $h_2$ will decrease the pressure drop between the top of pipe 1-2 and the junction. Thus, compared with pipes 2-3 and 2-4, the opposite effect occurs in pipe 1-2; $D_{12}$ increases with increasing $h_2$. For insight on how the energy equation supports this explanation, note that the first of Eqs. (3) requires that $D_{12} \approx (\Delta z_{12} - h_2)^{-4/19}$ *increases* as $h_2$ increases.

From this discussion it is clear that for an increasing $h_2$ there is a competition between the *decrease* of $D_{23}$ (and $D_{24}$) and an *increase* in $D_{12}$. Once the effect of $D$ on pipe cost is included through Eq. (6), as $h_2$ increases we see that the cost for pipes 2-3 and 2-4 *decrease*, and the cost for pipe 1-2 *increases*. A consequence of this competition is the existence of an optimum, in this case an optimal $h_2$, which produces the smallest possible cost.

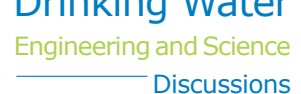

The mathematical basis for a unique solution for $h_2$ with cost minimization is now presented. In addition to the fixed pipe lengths, the total cost depends on the diameters for all of the pipes in the network. For the case of Fig. 2, where we now allow pipe 2-3 and pipe 2-4 to be different, get

$$C_T = C_T\big(D_{12}(h_2), D_{23}(h_2), D_{24}(h_2)\big) \tag{7}$$

Using the chain rule from the calculus, the total differential of Eq. (7) is

$$dC_T = \frac{\partial C_T}{\partial D_{12}} \frac{\partial D_{12}}{\partial h_2} dh_2 + \frac{\partial C_T}{\partial D_{23}} \frac{\partial D_{23}}{\partial h_2} dh_2 + \frac{\partial C_T}{\partial D_{24}} \frac{\partial D_{24}}{\partial h_2} dh_2 \tag{8}$$

5    The minimum value of $C_T$ is found once $dC_T = 0$ (and once it is verified that the second derivative of $C_T$ is positive thus indicating that $C_T$ is indeed a minimum). Requiring this, obtain

$$0 = \frac{\partial C_T}{\partial D_{12}} \frac{\partial D_{12}}{\partial h_2} + \frac{\partial C_T}{\partial D_{23}} \frac{\partial D_{23}}{\partial h_2} + \frac{\partial C_T}{\partial D_{24}} \frac{\partial D_{24}}{\partial h_2} \tag{9}$$

The cost $C_T$ is from Eq. (5), so the derivatives like $\partial C_T/\partial D_{12}$ in Eq. (9) are written in general as

$$\frac{\partial C_T}{\partial D_{ij}} = a\,b\,\frac{D_{ij}^{b-1}}{D_u^b} L_{ij} \tag{10}$$

10    for any link $ij$.

The derivatives like $\partial D_{12}/\partial h_2$ in Eq. (9) are obtained by taking the partial derivative of the pipe diameter with respect to the relevant pressure head in the appropriate energy equation. For the full energy equation, where $D$ appears in a nonlinear way in more than one location, this would be done using numerical methods. However, if we restrict our interest to minor-lossless, smooth-turbulent flow as noted above, we can use the energy equations like Eq. (3). Obtain for pipe 1-2

$$\frac{\partial D_{12}}{\partial h_2} = 0.156 \left(\frac{\Delta z_{12} - h_2}{L_{12}}\right)^{-23/19} \left(\frac{\nu^{1/7} Q_{12}}{g^{4/7} L_{12}^{19/7}}\right)^{7/19} \tag{11}$$

15    For pipe 2-3, we get

$$\frac{\partial D_{23}}{\partial h_2} = -0.156 \left(\frac{\Delta z_{23} + h_2 - h_3}{L_{23}}\right)^{-23/19} \left(\frac{\nu^{1/7} Q_{23}}{g^{4/7} L_{23}^{19/7}}\right)^{7/19} \tag{12}$$

and for pipe 2-4,

$$\frac{\partial D_{24}}{\partial h_2} = -0.156 \left(\frac{\Delta z_{24} + h_2 - h_4}{L_{24}}\right)^{\frac{23}{19}} \left(\frac{\nu^{\frac{1}{7}} Q_{24}}{g^{\frac{4}{7}} L_{24}^{\frac{19}{7}}}\right)^{\frac{7}{19}} \tag{13}$$

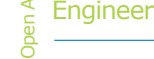

Equations (10)–(13) are combined with Eq. (9) to produce a single algebraic equation that depends on $h_2$, as well as $D_{12}$, $D_{23}$, and $D_{24}$. Introducing $D_{12}$, $D_{23}$, and $D_{24}$ from Eqs (3), we get

$$0 = Q_{12}^{7b/19}(\frac{\Delta z_{12} - h_2}{L_{12}})^{-(1+4b/19)} - Q_{23}^{7b/19}(\frac{\Delta z_{23} + h_2 - h_3}{L_{23}})^{-(1+4b/19)}$$
$$- Q_{24}^{7b/19}(\frac{\Delta z_{24} + h_2 - h_4}{L_{24}})^{-(1+4b/19)} \tag{14}$$

The general form of Eq. (14), written at any internal node is

$$0 = \sum_{ij,in} Q_{ij}^{7b/19} S_{ij}^{-(1+4b/19)} - \sum_{ij,out} Q_{ij}^{7b/19} S_{ij}^{-(1+4b/19)} \tag{15}$$

where the hydraulic gradient, $S_{ij}$, is

$$S_{ij} = \frac{\Delta z_{ij} + \Delta h_{ij}}{L_{ij}} \tag{16}$$

In Eq. (15) the indices $ij,in$ and $ij,out$ on the summations refer to inflows and outflows at the node (e.g., in Fig. 2, $ij,in$ $=12$ and $ij,out = 23$ and 24). Equation (15), the new CB algorithm proposed in this work, is written for each internal node in the network and solved simultaneously with the energy equation for each link to obtain unique and optimal values of $D$ for all links and $h$ for all internal nodes. It is understood that the nodal pressure heads determined from the solution of this system must be greater than or equal to the $h_{min}$ prescribed for the network. For nodes that do not satisfy this condition, the pressure head is set equal to $h_{min}$, as part of the CB algorithm.

Minor losses using the equivalent-length method can be included in the above developments by artificially extending the length of the link by $L_e$ in which minor loss occurs, thus contributing a non-zero $L_{ebyd}$ term in Eq. (1). We also extend the cost model of Eq. (5) from Swamee and Sharma (2008) to encompass two different ranges of pipe diameters having two different coefficients $a$ and exponents $b$. The link between the two ranges starts at discrete pipe size $D_{co}$, at and below which the cost model for the small (subscript $s$) pipe sizes applies, and discrete pipe size $D_{co+1}$, at and above which the cost model for the large (subscript $l$) pipe sizes applies. The cutoff diameter, $D_{co}$ is chosen by the designer based on inspection of cost vs. diameter data. Thus,

$$C_{ij} = L_{ij} \begin{cases} a_s(\frac{D_{ij}}{D_u})^{b_s}, & D_{ij} \leq D_{co} \\ c_1 + c_2 \frac{D_{ij}}{D_u} + c_3 \left(\frac{D_{ij}}{D_u}\right)^2 + c_4 \left(\frac{D_{ij}}{D_u}\right)^3, & D_{co} < D_{ij} < D_{co+1} \\ a_\ell(\frac{D_{ij}}{D_u})^{b_\ell}, & D_{ij} \geq D_{co+1} \end{cases} \tag{17}$$

In Eq. (17), $a_s$ and $a_\ell$ are the coefficients for the small and large pipe size regions, respectively, and $b_s$ and $b_\ell$ are the exponents for the small and large pipe size regions, respectively. A cubic spline is fit between pipe sizes $D_{co}$ and $D_{co+1}$ to complete the transition between small and large pipe sizes. The coefficients of this polynomial are $c_1, c_2, c_3$, and $c_4$ as seen in

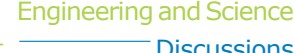



Eq. (17). These coefficients are evaluated by matching the cubic polynomial and pipe data at $D_{co}$ and $D_{co+1}$ and the first derivative of the polynomial with respect to $D_{ij}/D_u$ to $a_s b_s (\frac{D_{co}}{D_u})^{b_s-1}$ at $D_{ij} = D_{co}$ and to $a_\ell b_\ell (\frac{D_{co+1}}{D_u})^{b_\ell-1}$ at $D_{ij} = D_{co+1}$. An example of data for PVC pipe and the curvefit is shown in Fig. 3. The results of the curvefit are: $D_{co} = 2.067$ in., $D_{co+1} =$ 2.469 in., $a_s = \$1.349$ m$^{-1}$, $b_s = 1.157$, $a_\ell = \$1.381$ m$^{-1}$, $b_\ell = 1.344$, $c_1 = \$237.516$ m$^{-1}$, $c_2 = -\$316.125$ m$^{-1}$, $c_3 =$
5    $\$140.450$ m$^{-1}$, $c_4 = -\$20.499$ m$^{-1}$. It is clear from inspection of Fig. 3 that a one-part cost model would not have produced an acceptable curve-fit to pipe-cost data.

      With the inclusion of the two-part cost model and minor loss term, Eq. (15) becomes

$$0 = \sum_{ij,in} \frac{C'_{ij} A_{ij}^{\frac{4}{19}} (1+\epsilon_{ij})^{\frac{4}{19}} S_{ij}^{-\frac{23}{19}} \left(\frac{Q_{ij}^7 \nu}{g^4 D_u^{19}}\right)^{\frac{1}{19}}}{1-B A_{ij}^{\frac{4}{19}} \epsilon_{ij}' (1+\epsilon_{ij})^{-\frac{15}{19}} S_{ij}^{-\frac{4}{19}} \left(\frac{Q_{ij}^7 \nu}{g^4 D_u^{19}}\right)^{\frac{1}{19}}} - \sum_{ij,out} \frac{C'_{ij} A_{ij}^{\frac{4}{19}} (1+\epsilon_{ij})^{\frac{4}{19}} S_{ij}^{-\frac{23}{19}} \left(\frac{Q_{ij}^7 \nu}{g^4 D_u^{19}}\right)^{1/19}}{1-B A_{ij}^{\frac{4}{19}} \epsilon_{ij}' (1+\epsilon_{ij})^{-\frac{15}{19}} S_{ij}^{-\frac{4}{19}} \left(\frac{Q_{ij}^7 \nu}{g^4 D_u^{19}}\right)^{1/19}} \tag{18}$$

where $B = 0.1989$ and

$$\epsilon_{ij} = \sum_k \left(\frac{L_e}{D}\right)_{k,ij} \frac{D_{ij}}{L_{ij}} \tag{19}$$

$$\epsilon_{ij}' = \sum_k \left(\frac{L_e}{D}\right)_{k,ij} \frac{D_u}{L_{ij}} \tag{20}$$

$$A_{ij} = \begin{cases} 0.318, & \text{smooth pipe} \\ 0.420, & \text{steel pipe} \end{cases}$$

10    and $A$ accounts for the effect of pipe roughness (smooth and commercial steel). The term $C'_{ij}$ is the derivative of the cost function per unit length with respect to $D/D_u$. For the two-part cost model from above, obtain

$$C'_{ij} = \begin{cases} a_s b_s (\frac{D_{ij}}{D_u})^{b_s-1}, & D_{ij} \leq D_{co} \\ c_2 + 2c_3 (\frac{D_{ij}}{D_u}) + 3c_4 (\frac{D_{ij}}{D_u})^2, & D_{co} < D_{ij} < D_{co+1} \\ a_\ell b_\ell (\frac{D_{ij}}{D_u})^{b_\ell-1}, & D_{ij} \geq D_{co+1} \end{cases} \tag{21}$$

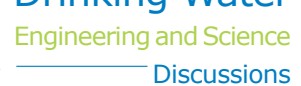

Equation (18), which forms the basis for calculus-based optimization in this work, is applied at all internal nodes to uniquely determine $h$. Equation (18) is valid over the range of $\sim$4000 $<$ Re $<$ $\sim$300,000.

## 4 Backtracking Algorithm and Genetic Algorithm

Backtracking (BT) and genetic algorithm (GA) assess candidate solutions composed of discrete diameters from a commercially available set. These candidates are represented by a vector of size $[1, N_L]$ where each element corresponds to a network link. The values of the vector specify a diameter from the commercially available set that are indexed from smallest ($i_D = 1$) to largest ($i_D = N_D$). To reduce the computational time associated with these evaluations, the constraints imposed by the energy equation and cost minimization may be more efficiently evaluated through lookup tables. With fixed $L$, $\Delta z$, $K$, $L_{ebyD}$, and $\alpha$, the change in static pressure head $\Delta h$ is evaluated for all $N_D x N_L$ combinations of link index and pipe diameter

$$\Delta \boldsymbol{h} = \begin{bmatrix} \Delta h_{11} & \cdots & \Delta h_{1N_L} \\ \vdots & \ddots & \vdots \\ \Delta h_{N_D1} & \cdots & \Delta h_{N_DN_L} \end{bmatrix} \tag{22}$$

While an algorithm evaluates a candidate solution, the static pressure head at each node is sequentially calculated by "marching" through the network. Starting with the fixed source pressure head, the algorithm finds the pressure head $h_i$ for a given node by adding the head at the upstream node, $h_{i-1}$ to the change in head for that link $i_L$ and the diameter $i_D$ under consideration. Thus,

$$h_i = h_{i-1} + \Delta \boldsymbol{h}(i_D, i_L) \tag{23}$$

Along with the hydraulic evaluation of a candidate solution, the cost of the partial candidate is found through the use of a lookup table $\boldsymbol{C}$,

$$\boldsymbol{C} = \begin{bmatrix} C_{11} & \cdots & C_{1N_L} \\ \vdots & \ddots & \vdots \\ C_{N_D1} & \cdots & C_{N_DN_L} \end{bmatrix} \tag{24}$$

where $\boldsymbol{C}(i_D, i_L)$ returns the additional cost of assigning a diameter with index $i_D$ to link $i_L$. In this way, the candidate solution's hydraulic performance and cost are incorporated into the genetic algorithm and backtracking approaches. In contrast to GA, the backtracking algorithm evaluates static pressure head and cost upon consideration of each partial candidate, where GA calculates these values on full candidates as part of the objective function.

### 4.1 BT and GA Pre-processor 1: Maximum Available Diameter

To increase the efficiency of BT and GA, it is advantageous to limit the number of pipe diameters in the available set, especially those outside of the range of the optimal solution. In particular for the BT algorithm, larger diameters can require considerable computational effort, since they tend not to violate static head requirements and require multiple-link partial candidates for the algorithm to reject them once their cost exceeds that of an already-found viable candidate. Therefore, a pre-





processor is used to provide a maximum diameter ($D_{max}$) that should be considered during the optimization process. This procedure, which produces a conservative estimate, finds the smallest diameter at which a network with a single pipe diameter choice produces no nodes with a static pressure head below $h_{min}$, similar to the technique used by Mohan and Jinesh Babu (2009). After this diameter is found, the next-larger diameter in the set is selected as $D_{max}$ in order to allow the algorithm to

select a larger-than-necessary diameter if this is able to save cost elsewhere. If $D_{max}$ appears in the optimum solution, the designer can elect to further increase this maximum diameter. It worth noting that Kadu et al. (2008) presents another method to further prune the search space with the critical path concept, where Dongre and Gupta (2011) noted the computational advantages of having just four diameter choices per link. This method, however, may prune the global optimum and may not produce feasible HGL values at intermediate nodes, as in the case of networks with a local high point.

**4.2   BT and GA Pre-processor 2: Adjusted Minimum Static Pressure Head**

A second pre-processor adjusts the minimum static pressure head requirement for each internal node by considering the total head required at downstream nodes. It can be recognized that, without the use of a pump, the total head cannot increase at nodes downstream of a given node $i$. Furthermore, the total head must decline at a minimum grade that is determined by the demand volume flow rate and the largest pipe diameter available ($D_{max}$) for selection. This energy constraint is utilized to

reduce the number of candidates to be considered by increasing the minimum static pressure head at nodes where these rules produce a higher minimum head than the original $h_{min}$. For example, nodes upstream of a local network high point can have their minimum static pressure head increased beyond the normal minimum, since the static pressure head must be great enough to ensure adequate flow to the higher-elevation downstream node. To begin this process, each node $i$ is initialized with a baseline minimum total head,

$$th_{min,i} = z_i + h_{min} \tag{25}$$

$th_{min,i}$ is thus initialized by considering only the node's hydraulic requirements in isolation, i.e., without acknowledging the neighboring downstream nodes. The pre-processor then considers updating $th_{min,i}$ by checking the following condition, which is false when the minimum static pressure head at downstream nodes produces further constraints on an upstream node $i$. Thus, for all nodes $i$ which are upstream of some node $j$, the following inequality can be evaluated

$$th_{min,i} - th_{min,j} \geq \left( \alpha_{i-j} + K_{i-j} + f_{i-j} \left( \frac{L_{i-j}}{D_{i-j}} + L_{ebyD_{i-j}} \right) \right) \frac{8Q_{i-j}{}^2}{\pi^2 g D_{max}{}^4} \tag{26}$$

Also, consider that when flow rate $Q_{i-j}$ is small and $D_{max}$ is large, the right hand side of Eq. (26) approaches zero, representing the simple statement that upstream total head must always be greater than downstream total head. When the condition in Eq. (26) is false, the minimum total head can be updated in node $i$ such that the maximum diameter size in link i-j is able to meet the downstream node's minimum total head, or



$$th_{min,i} = th_{min,j} + \left( \alpha_{i-j} + K_{i-j} + f_{i-j} \left( \frac{L_{i-j}}{D_{i-j}} + L_{ebyD_{i-j}} \right) \right) \frac{8Q_{i-j}{}^2}{\pi^2 g D_{max}{}^4} \tag{27}$$

In this way, $th_{min,i}$ may be updated for each node until the condition in Eq. (26) is true for all nodes $i$ with a downstream node $j$ connected by a single link.

After the values for $th_{min,i}$ are updated, they are converted back into minimum static pressure head values by subtracting the elevation $z_i$ from $th_{min,i}$. This pre-processor serves to narrow the search for viable candidate solutions by potentially increasing the minimum static pressure head. Since backtracking and GA consider network links in the downstream direction, these algorithms are otherwise blind to future downstream static pressure head requirements. This limitation is alleviated by the pre-processor, which allows these algorithms some implicit information about what local diameter choices will be viable for the full network solution. Note that both pre-processors discussed will not prune the global optimum from the solution.

## 4.3 Backtracking Algorithm (BT)

The backtracking algorithm employs a systematic search of candidate solutions to find a global optimum. The algorithm operates within a recursive structure to incrementally build candidate solutions while checking the candidates for hydraulic and cost acceptability. The strength of the BT is that, upon discovery of an infeasible partial candidate, all extensions of that candidate can be eliminated from consideration. In this way, a large number of solutions can be pruned from the solution tree to achieve greater computational efficiency. Backtracking is a type of partial enumeration method, which Raad (2011) notes can drastically reduce the number of solutions to be evaluated based on two rejection criteria. The first rejection criterion is that when a candidate violates static pressure head constraints, all candidates with equal or lesser diameter sizes can be discarded. This condition is leveraged even more effectively with pre-processor 3 above, which can increase static pressure heads. The second rejection criterion is that once a feasible candidate has been found, all other partial candidates with a higher cost can also be discarded. The BT algorithm further extends this criteria by considering that the links yet to be considered in a partial candidate, an "extension" to the partial candidate, will cost at a minimum that of the entire extension being composed of the smallest available diameter. Thus, when considering whether the partial candidate will necessarily be more expensive than the running optimum, this minimum extension cost can be added to the partial candidate cost.

The backtracking algorithm begins its search of the solution tree by considering the partial candidate with the smallest diameter size assigned to the first network link. The static pressure head and the partial candidate cost at the outlet node are calculated with the $\Delta \boldsymbol{h}$ and $\boldsymbol{C}$ lookup tables. If this partial candidate meets static pressure head and cost requirements, the algorithm extends this partial candidate by assigning the smallest diameter to the downstream link. If a partial candidate produces a node that is rejected on static pressure head basis, the next largest larger diameter is chosen for the link upstream of the node. If no diameter satisfies the pressure head condition, the algorithm backtracks to the upstream link and assigns a larger diameter to the link. If a node is connected to a delivery node by a single link, the smallest feasible diameter for that

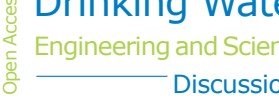


link is found, and if no such diameter exists, the partial candidate is rejected. In this way, the algorithm continues to extend and reject candidate solutions until a full candidate satisfies the static pressure head requirements. Once this has been achieved, the diameter choices and cost of the network are stored as a running optimum.

Once a working solution has been found, candidate solutions may be rejected on the basis of cost. For each new

candidate, cost is calculated by adding the cost of diameters that have already been assigned to the cost of assigning all downstream links with the smallest diameter available. If this cost exceeds the cost of the running optimum, the partial candidate is rejected. Unlike a candidate rejection based on static pressure head, a rejection based on cost does not consider siblings with larger diameters, since these would only add cost, rather, the algorithm backtracks immediately to re-assigning the upstream link.

In this way, the rejection criteria based on a minimum static pressure head and cost are used to prune the solution tree and largely reduce the number of non-optimal candidates that need to be considered. The minimum static pressure head criterion tends to prune candidates with diameters that are too small, while the running optimal cost criterion tends to prune candidates of diameters that are too large. Another pruning technique noted by Raad (2011) is to group together adjacent links that are sized identically. This technique, in contrast to the former two mentioned, cannot guarantee an optimal result, and is

therefore not included in the present study's BT algorithm. The present study's BT algorithm operates similarly to the method presented by González-Cebollada et al. (2011), with the major difference being that the BT algorithm continues searching once it has found its first feasible solution. The BT algorithm could also be used in this way to find an initial solution very quickly, and then continue as normal to find progressively better solutions until the end of the search space, or a predefined condition such as calculation time, are met.

**4.4  Modified Backtracking Algorithm (BT-NoUp)**

A modification to the BT algorithm was made to further improve its computational speed, although at the risk of pruning the global optimum from the search. This modified algorithm (BT-NoUp) rejects all candidates that feature a smaller diameter that is upstream of a larger diameter when an equal or smaller flow rate is present in the downstream link. Typically, optimal networks would not exhibit this feature, and in cases where a single source feeds into a network with constant-length links, it

is advantageous (or equivalent) to place larger diameters upstream of smaller diameters. However, due to the discrete nature of diameter choices and link lengths, an optimization problem may, in fact, select a candidate with a larger diameter downstream from smaller ones. For this reason, the BT-NoUp algorithm, unlike the BT algorithm, may miss the global optimum at the expense of its greater computational efficiency.



## 4.5 Genetic Algorithm (GA)

Genetic algorithms are stochastic optimization techniques that mimic the process of natural selection, and numerous recent variations of GAs have demonstrated improved performance on WDN design (Nicklow et al. 2010). Given their popularity, the GA included in this study is meant to provide a point of comparison to the BT and CB algorithms when applied to GDWNs, and represents a straightforward GA implementation with an attempt to select appropriate operators and well-tuned parameters.

When implemented in water network design, each candidate solution represents a selection of pipe diameters. The algorithm is initialized with a population of candidates of size $N_c$ that repeatedly undergoes the processes of mutation, crossover, and selection

$$c_i = [D_{1,i} \quad D_{2,i} \quad ... \quad D_{N_L,i}] \tag{28}$$

where each candidate in the population $c_i$ contains $N_L$ diameters. In the present work, candidates are represented as a string of natural numbers, which is used over a binary representation to improve the efficiency of encoding and ensure uniqueness of coded strings (Vairavamoorthy and Ali 2000). The mutation operator replaces pipe diameters with a diameter from a uniform random distribution, where each link diameter has a probability of $p_{mut}$ of mutating on each generation. The crossover operator randomly pairs all individuals in the population and performs a single-point crossover of the two individuals with probability $p_{xover}$, where the point of crossover is chosen randomly in the string of diameters. The fitness, $f_i$, of each candidate is assessed with penalties associated with the solution's pipe cost, $C_{pipe,i}$, and violations of the static pressure head requirement, $C_{hyd,i}$, or

$$f_i = \frac{1}{C_{pipe,i} + C_{hyd,i}} \tag{29}$$

The hydraulic cost is obtained for each individual by identifying nodes in which the static pressure head is less than $h_{min}$ and multiplying the total amount of head violation by a hydraulic penalty coefficient, $a_{hyd}$:

$$C_{hyd,i_c} = a_{hyd} \sum_1^{N_L} \left(h_{min} - h_{i_N}\right) \mid h_{i_N} < h_{min} \tag{30}$$

To allow for a hydraulic penalty coefficient to produce similar results in both small-scale (inexpensive) network and a large-scale (more expensive) cases, the hydraulic penalty coefficient is made directly proportional to the average solution cost. With each generation, $a_{hyd}$ is updated by multiplying the normalized penalty coefficient, $a_{hyd,norm}$, by the average pipe cost of the population,

$$a_{hyd} = a_{hyd,norm} \frac{\sum_1^{N_c} C_{pipe,i_C}}{N_c} \tag{31}$$

The algorithm then selects candidates to be carried into the next generation through a pinwheel lottery (with replacement), where each candidate has a probability of being selected, $p_{sel,i}$ in direct proportion to its fitness relative to the sum of all fitness values in the population



$$p_{sel,i} = \frac{f_i}{\sum_1^{N_c} f_i} \qquad (32)$$

The algorithm replaces the parent generation with a generation of equal size and tends to select more fit individuals in successive generations. In this study, the genetic algorithm parameters used were $p_{mut} = 0.02$, $N_c = 50$, $p_{xover} = 0.5$, $N_{gen} = 100$, $a_{hyd,norm} = 0.1$. The first four of these parameters were chosen based on typical values presented in the literature and then tuned with a sensitivity analysis for the first test case. Simspon et al. (1994) present typical values for $N_c$ (30 - 200),

$p_{xover}$ (0.7 - 1.0), $p_m$ (0.01 - 0.05), and $N_{gen}$ (100 - 1000). The normalized hydraulic penalty coefficient, $a_{hyd,norm}$, was chosen such that the GA converged on solutions which tended to satisfy the minimum static pressure constraint, but still allowed the population to gravitate towards smaller diameters with static pressures close to $h_{min}$.

## 5   Cases Studied

Five cases were studied based on actual GDWN in Panama, Nicaragua, and the Philippines. Global characteristics of

each network are presented in Table 1 and the details of each network are presented in Table 3(a)-(e). Each network is a branching type without loops. The total lengths of the networks range from less than 1 km to over 15 km. Two serial networks are tested in order to demonstrate the effect of a local high point on the algorithm solutions. Elevation plots for each case are shown in Fig. 4.

In the present study $h_{min} = 7$ m, although this requirement was reduced at selected nodes at the beginning of networks

where changes in elevation are still small. At the source node, the static pressure head is fixed at atmospheric. All cases assumed minor-lossless flow, although all algorithms (e.g., Eq. (18) for CB-Theor) are capable of handling minor loss coefficients through the equivalent length method as presented above.

## 6   Results

The current study evaluated three types of algorithms that optimize the design of gravity-driven water networks

(GDWN). The algorithms include a calculus-based (CB) algorithm, a backtracking algorithm (BT), a modified backtracking version (BT-NoUp), and a genetic algorithm (GA). The algorithms were applied to five test cases that are based on real GDWN.

The global optimum network cost, found with BT, is shown in Table 2. The costs of solutions from all other algorithms are expressed as a percentage difference in cost from the global optimum cost. To visually compare the algorithm solutions, the hydraulic grade lines from BT, BT-NoUp, CB-Theor, and CB-Disc are presented in Fig. 4 along with the network elevation

for each test case. For clarity, the hydraulic grade lines of branch links are omitted from the figure. In addition, the GA solutions are omitted since 100 solutions were obtained for each test case. Collectively, the hydraulic grade lines reveal a close alignment of the BT solution (the global optimum) with the CB-Theor solution which utilizes a continuous diameter set. Furthermore,



the mapping scheme used to generate a CB-Disc solution is shown to increase pipe sizes in some cases far beyond the limit imposed by $h_{min}$, which was set to 7 m in the present work.

In practice, a GDWN must be designed with pipe diameters that are selected from a discrete, commercially available set. With a given number of network links, $N_L$, and a number of available diameters, $N_D$, a total of $N_D^{N_L}$ candidate solutions exist, yet with only one global optimum (except in the case of no viable solutions or unique solutions with identical costs). For example, a GDWN of 20 links and 13 commercially available pipe sizes will, in principal, produce approximately $1.9x10^{22}$ candidate solutions. However, BT is able to find the global optimum without needing to check all of the possible solutions by using a set of rules to prune infeasible and sub-optimal candidates. In this study, BT evaluated only a fraction of the candidate solutions, where the fraction ranged from $4x10^{-18}$ to $7x10^{-4}$.

To further reduce the number of evaluations required to arrive at a solution, the BT algorithm was modified (BT-NoUp) to prune all solutions that include a smaller diameter that is upstream of a larger diameter. This criterion, which seems intuitive to the designer, may actually miss the global optimum due to trade-offs associated with discrete solutions. In fact, BT-NoUp missed the global optimum in cases 2 and 3, although by a small percentage increase in cost (2.1% and 0.35% respectively). BT-NoUp, however, finished its search in a shorter amount of time in comparison to BT. Using a Dell Latitude (i5 CPU at 2.50 GHz), the time to evaluate one candidate was around 0.2 ms. In addition to approximately 2 s of pre-processing time for BT and GA, the computation times for BT ranged from 0.08 s (case 1) to 79 s (case 3), while BT-NoUp ranged from 0.06 s (case 4) to 0.3 s (case 3). The number of available diameters used in the BT, BT-NoUp, and GA runs are listed in Table 1.

The CB algorithm, unlike the other algorithms in this work, finds a solution with theoretical diameters that are drawn from a continuous domain (CB-Theor). For all test cases, the costs of the CB-Theor solutions was less when compared to the discrete-diameter global optimum (-5.46% to -2.60%). In fact, because of the discrete pipe sizes needed for an actual network, *the continuous model will always produce the smallest theoretical cost for the network*. The CB algorithm then maps this solution to a commercially-available discrete set (CB-Disc). The mapping process used in this study simply mapped each theoretical diameter to the nearest available diameter of a larger size, thus producing a solution which still satisfies static head requirements but with a higher associated material cost. This tended to oversize the diameters, although the CB-Disc solutions were always within two diameters of the BT global optimum solutions, as shown in Fig. 5. From all of the test cases combined, all but one (71 out of 72) of the diameter selections were within one diameter of the global optimum. More sophisticated mapping schemes, like independently adjusting $D$ for each link in the distribution main in a step-by-step manner starting with the source while ensuring all pressure head constraints are satisfied, would more likely produce results identical to the global optimum. This was not performed in the current study. The CB-Disc solution costs were, in all cases, larger than the global optimum, with a percentage difference ranging from 3.86% to 22.6%. Thus, for all cases, the calculus-based algorithm bounded the cost of the global optima with a lower-cost CB-Theor solution and a higher-cost CB-Disc solution. This trend is a result of the additional constraints imposed by the finite set of diameter choices. If the algorithm is allowed a greater number of discrete





diameter choices, i.e., through adding a less-common nominal diameter size to the available set, the cost of the CB-Disc solution would approach the CB-Theor solution.

GA was run on each case a total of 100 times, each run itself produced 100 generations of 50 candidates. The least-cost candidate in that did not violate the static pressure head condition was chosen as the optimum. Because GA is a stochastic search algorithm producing different results from run-to-run, the costs of the optima from all 100 runs were averaged, with this averaged value presented in Table 2 as a percentage increase from the global optimum. Out of the 100 GA runs for each test case, nearly all runs failed to achieve the global optimum, with the exception of 8 runs of the Kiangan network. On a Dell Latitude (i5 CPU at 2.50 GHz), GA runs took between 0.9 s (case 1) and 1.2 s (case 3), not including about 2 s of pre-processor time. We note that many variations of GAs have been reported in the literature and several of these would likely improve upon the GA results obtained in this study. Potential improvements to the GA include scaling of the fitness function to magnify the rewards towards slightly fitter candidates (Dandy et al. 1996), a self-adapting penalty function (Wu and Walski 2005), the use of elitism to preserve the best solutions (Kadu et al. 2008), the systematic optimization of operator parameters (Reed et al. 2000), and a reduction in the search space (Kadu et al. 2008). The GA results, however, allow for a point of comparison to the BT and CB algorithms for a simple GA on real-world GDWNs.

## 7 Conclusions

Algorithms to optimize the cost of branching gravity-driven water networks are evaluated on five test cases from real networks in the Philippines, Nicaragua, and Panama. A calculus-based algorithm produced a solution composed of theoretical diameters from a continuous set (CB-Theor), which are then mapped onto discrete commercially available diameters (CB-Disc). Backtracking (BT), a recursive algorithm, systematically searches discrete candidate solutions and is guaranteed to find the global optimum by following rules that prune only higher-cost or hydraulically infeasible candidates. The BT algorithm was modified (BT-NoUp) to improve computational speed by also rejecting all candidates that included a small diameter directly upstream of a larger diameter. This criterion allowed BT-NoUp to prune more candidate solutions but allowed for the possibility of missing the global optimum. The third type of algorithm evaluated was a genetic algorithm (GA).

Backtracking was able to find the global optimum in all test cases with relatively little computational effort, and could be applied to other GDWNs composed of a similar number of links. This approach, however, could become prohibitively time-consuming when dealing with networks with significantly more links. The calculus-based algorithm produced consistently good results for the networks tested, although a more robust mapping scheme from theoretical diameters to discrete diameters would further improve on these results as discussed above. In another novel approach, the CB-Theor solutions could be used to prune the BT search space, similar to Kadu et al. (2008), by only including the two diameters above and below the CB-Theor diameters, producing four diameter choices per link. The calculus-based methodology provides an additional benefit to



the designer by explicitly revealing the sensitivities to cost for a design. The calculus-based algorithm requires greater computational effort than backtracking for smaller networks, however, this effort scales more linearly with the number of network links, while backtracking scales exponentially. Furthermore, backtracking's computational time is sensitive to the number of available diameters. Still, when applied to GDWNs with a similar number of links to the test cases, backtracking

can quickly find a global optimum. In addition, because it is guaranteed to find the global optimum, it can be useful for benchmarking the performance of other algorithms which scale better with more network links. While the genetic algorithm produced solutions with decent closeness to the global optimum, run-to-run results vary due to the stochastic nature of the algorithm. Overall, the genetic algorithm as implemented did not produce results which deemed it compelling over deterministic methods as applied to GDWNs. However, for more complex networks and problem formulations, a genetic

algorithm may be more advantageous. In this case, the present study's GA could be greatly improved on through many improvements reported in the literature (Nicklow et al. 2010).

For all test cases, the calculus-based algorithm's theoretical diameter solutions (CB-Theor) produced a lower cost than the discrete-domain global optimum. This result is made possible because of it is not constrained to a discrete set of diameters. As such, the *CB-Theor results represent a lower-bound on the optimum solution within the problem formulation*,

which could be approached with a finer selection of pipe diameters.

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

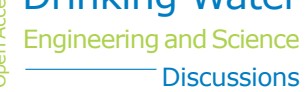

**Table 1: Characteristics of Test Cases.**

| Test Case | Type | Number of Diameter Choices | Number of Links | $Q_{tot}$ ( L s$^{-1}$ ) | $L_{tot}$ ( km ) |
|---|---|---|---|---|---|
| 1.  Kiagan, Philippines | Branching | 8 | 9 | 4.37 | 0.82 |
| 2.  Los Modulus, Nicaragua | Serial | 4 | 13 | 0.39 | 1.24 |
| 3.  Cañazas, Panama | Branching | 10 | 23 | 6.29 | 15.2 |
| 4.  San Miguel, Nicaragua | Serial | 9 | 10 | 0.40 | 1.18 |
| 5.  El Guabo, Nicaragua | Branching | 12 | 17 | 17.7 | 4.71 |



**Table 2: Solution Costs.**

| Test Case | Global Optimum | Percentage cost increase from global optimum | | | |
|---|---|---|---|---|---|
| | BT | BT-NoUp | CB-Theor | CB-Disc | GA |
| 1.  Kiagan, Philippines | $   2,331 | 0 | -3.16 | 11.3 | 4.80 |
| 2.  Los Modulos, Nicaragua | $   1,441 | 2.10 | -2.60 | 22.6 | 12.1 |
| 3.  Cañazas, Panama | $ 72,190 | 0.35 | -5.46 | 17.0 | 20.7 |
| 4.  San Miguel, Nicaragua | $   5,418 | 0 | -4.54 | 3.86 | 6.20 |
| 5.  El Guabo, Nicaragua | $ 61,445 | 0 | -3.16 | 20.2 | 13.3 |



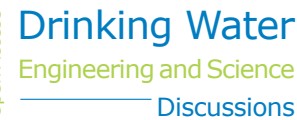

**Table 3(a): Case #1 Kiangan network properties, diameter results (inch nominal sizes, with CB-Theor in inches), and nodal h.**

|  | Link | 1-2 | 2-3 | 3-4 | 4-5 | 5-6 | 2-7 | 3-8 | 4-9 | 5-10 |
|---|---|---|---|---|---|---|---|---|---|---|
| **Network** | Length (m) | 76 | 113 | 19 | 54 | 75 | 80 | 99 | 170 | 135 |
|  | Q (L/s) | 4.37 | 3.68 | 2.94 | 1.46 | 0.69 | 0.69 | 0.74 | 1.48 | 0.77 |
|  | Δz (m) | 14.0 | 1.0 | 0.0 | 0.0 | -1.0 | 0.0 | -2.0 | 3.0 | 2.0 |
| **D solutions** | BT | 3 | 2-½ | 2 | 1-½ | 1-½ | 1 | 1-¼ | 1-½ | 1-¼ |
|  | BT-NoUp | 3 | 2-½ | 2 | 1-½ | 1-½ | 1 | 1-¼ | 1-½ | 1-¼ |
|  | CB-Theor | 2.75 | 2.56 | 2.14 | 1.83 | 1.36 | 1.06 | 1.38 | 1.58 | 1.13 |
|  | CB-Disc | 3 | 3 | 2-½ | 2 | 1-¼ | 1-¼ | 1-¼ | 1-½ | 1-¼ |

|  | Node | 1 | 2 | 3 | 4 | 5 | 6 | 7 | 8 | 9 | 10 |
|---|---|---|---|---|---|---|---|---|---|---|---|
| **h (m)** | BT | 0 | 13.09 | 11.43 | 10.72 | 8.81 | 7.10 | 7.27 | 7.21 | 7.57 | 7.58 |
|  | BT-NoUp | 0 | 13.09 | 11.43 | 10.72 | 8.81 | 7.10 | 7.27 | 7.21 | 7.57 | 7.58 |
|  | CB-Theor | 0 | 12.48 | 11.24 | 10.65 | 9.61 | 7.00 | 6.99 | 7.00 | 7.00 | 3.19 |
|  | CB-Disc | 0 | 13.09 | 13.15 | 12.85 | 12.27 | 9.78 | 11.51 | 8.94 | 9.70 | 11.04 |



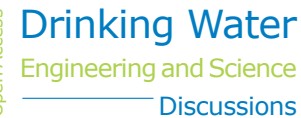

**Table 3(b): Case #2 Los Modulos network properties, diameter results (inch nominal sizes, with CB-Theor in inches), and nodal h.**

| | Link | 1-2 | 2-3 | 3-4 | 4-5 | 5-6 | 6-7 | 7-8 | 8-9 | 9-10 | 10-11 | 11-12 | 12-13 | 13-14 | |
|---|---|---|---|---|---|---|---|---|---|---|---|---|---|---|---|
| Network | Length (m) | 60 | 41 | 108 | 46 | 134 | 153 | 79 | 157 | 90 | 32 | 102 | 120 | 117 | |
| | Q (L/s) | 0.39 | 0.39 | 0.39 | 0.39 | 0.39 | 0.39 | 0.39 | 0.39 | 0.39 | 0.39 | 0.39 | 0.39 | 0.39 | |
| | Δz (m) | 11.2 | -0.5 | 32.8 | -3.7 | 36.6 | -2.3 | 15.7 | -6.8 | 7.3 | -7.4 | 4.5 | -1.2 | 8.4 | |
| | | | | | | | | | | | | | | | |
| D solutions | BT | 1 | 1 | ¾ | ¾ | ¾ | ¾ | 1 | ¾ | 1 | 1 | ¾ | ¾ | ¾ | |
| | BT-NoUp | 1 | 1 | 1 | 1 | 1 | ¾ | ¾ | ¾ | ¾ | ¾ | ¾ | ¾ | ¾ | |
| | CB-Theor | 0.99 | 0.98 | 0.85 | 0.85 | 0.85 | 0.85 | 0.85 | 0.85 | 0.85 | 0.85 | 0.85 | 0.85 | 0.85 | |
| | CB-Disc | 1 | 1 | 1 | 1 | 1 | 1 | 1 | 1 | 1 | 1 | 1 | 1 | 1 | |
| | | | | | | | | | | | | | | | |
| | Node | 1 | 2 | 3 | 4 | 5 | 6 | 7 | 8 | 9 | 10 | 11 | 12 | 13 | 14 |
| h (m) | BT | 0 | 9.55 | 7.94 | 31.59 | 24.00 | 49.25 | 33.99 | 47.55 | 27.45 | 32.32 | 24.05 | 19.91 | 8.55 | 7.04 |
| | BT-NoUp | 0 | 9.55 | 7.94 | 37.82 | 32.88 | 65.85 | 50.59 | 59.60 | 39.50 | 39.18 | 29.07 | 24.93 | 13.56 | 12.05 |
| | CB-Theor | 0 | 9.00 | 7.00 | 31.86 | 24.78 | 51.53 | 37.98 | 47.87 | 29.53 | 30.21 | 20.46 | 17.46 | 7.44 | 7.23 |
| | CB-Disc | 0 | 9.55 | 7.94 | 37.82 | 32.88 | 65.85 | 59.41 | 72.98 | 61.93 | 66.80 | 58.53 | 60.27 | 55.83 | 61.06 |



**Table 3(c): Case #3 Cañazas network properties, diameter results (inch nominal sizes, with CB-Theor in inches), and nodal h.**

| Network | | 1-2 | 2-3 | 3-4 | 4-5 | 5-6 | 6-7 | 7-8 | 8-9 | 9-10 | 10-11 | 11-12 | 12-13 | 2-14 | 3-15 | 4-16 | 5-17 | 6-18 | 7-19 | 8-20 | 9-21 | 10-22 | 11-23 | 12-24 |
|---|---|---|---|---|---|---|---|---|---|---|---|---|---|---|---|---|---|---|---|---|---|---|---|---|
| | Link | 1-2 | 2-3 | 3-4 | 4-5 | 5-6 | 6-7 | 7-8 | 8-9 | 9-10 | 10-11 | 11-12 | 12-13 | 2-14 | 3-15 | 4-16 | 5-17 | 6-18 | 7-19 | 8-20 | 9-21 | 10-22 | 11-23 | 12-24 |
| | Length (m) | 646 | 275 | 957 | 509 | 1102 | 291 | 1764 | 1256 | 2320 | 1580 | 2170 | 1217 | 160 | 100 | 1250 | 110 | 570 | 180 | 1400 | 50 | 400 | 260 | 100 |
| | Q (L/s) | 6.29 | 5.49 | 5.39 | 5.34 | 5.14 | 2.84 | 2.74 | 2.49 | 2.39 | 0.69 | 0.39 | 0.20 | 0.80 | 0.10 | 0.05 | 0.20 | 2.30 | 0.10 | 0.25 | 0.10 | 1.70 | 0.30 | 0.19 |
| | Δz (m) | 25.0 | 38.9 | 11.9 | 42.1 | -22.9 | 32.3 | -29.9 | 40.8 | -3.0 | -14.7 | 34.1 | -7.6 | -5.0 | 20.0 | -15.0 | 2.0 | -12.0 | 14.0 | -6.0 | 5.0 | -1.0 | -13.0 | 9.0 |

| D solutions | | 1-2 | 2-3 | 3-4 | 4-5 | 5-6 | 6-7 | 7-8 | 8-9 | 9-10 | 10-11 | 11-12 | 12-13 | 2-14 | 3-15 | 4-16 | 5-17 | 6-18 | 7-19 | 8-20 | 9-21 | 10-22 | 11-23 | 12-24 |
|---|---|---|---|---|---|---|---|---|---|---|---|---|---|---|---|---|---|---|---|---|---|---|---|---|
| | BT | 4 | 3 | 3 | 4 | 3 | 3 | 3 | 2-½ | 2-½ | 2 | 1-¼ | 1 | 1-¼ | ½ | ½ | ½ | 2 | ½ | 1 | ½ | 1-½ | 1-¼ | ½ |
| | BT-NoUp | 4 | 4 | 4 | 4 | 3 | 3 | 2-½ | 2-½ | 2-½ | 2 | 1-¼ | 1 | 1-¼ | ½ | ½ | ½ | 1-½ | ½ | 1 | ½ | 1-½ | 1-½ | ½ |
| | CB-Theor | 3.530 | 3.531 | 3.333 | 3.307 | 3.270 | 2.727 | 2.698 | 2.579 | 2.548 | 1.862 | 1.227 | 1.011 | 1.283 | 0.325 | 0.508 | 0.404 | 1.678 | 0.343 | 0.963 | 0.281 | 1.405 | 1.401 | 0.488 |
| | CB-Disc | 4 | 4 | 4 | 4 | 4 | 3 | 3 | 3 | 3 | 2 | 1-¼ | 1 | 1-¼ | ½ | ½ | ½ | 2 | ½ | 1 | ½ | 1-½ | 1-½ | ½ |

| h (m) | | 1 | 2 | 3 | 4 | 5 | 6 | 7 | 8 | 9 | 10 | 11 | 12 | 13 | 14 | 15 | 16 | 17 | 18 | 19 | 20 | 21 | 22 | 23 | 24 |
|---|---|---|---|---|---|---|---|---|---|---|---|---|---|---|---|---|---|---|---|---|---|---|---|---|---|
| | Node | 1 | 2 | 3 | 4 | 5 | 6 | 7 | 8 | 9 | 10 | 11 | 12 | 13 | 14 | 15 | 16 | 17 | 18 | 19 | 20 | 21 | 22 | 23 | 24 |
| | BT | 0 | 21.1 | 55.4 | 51.5 | 91.3 | 51.7 | 82.5 | 43.9 | 69.8 | 41.4 | 22.1 | 40.0 | 21.7 | 12.1 | 72.2 | 24.3 | 82.2 | 26.1 | 90.8 | 20.1 | 73.3 | 21.9 | 7.81 | 39.7 |
| | BT-NoUp | 0 | 21.1 | 58.8 | 66.4 | 106 | 66.6 | 97.4 | 42.8 | 68.8 | 40.4 | 21.0 | 39.0 | 20.7 | 12.1 | 75.6 | 39.2 | 97.1 | 9.5 | 106 | 19.0 | 72.2 | 20.9 | 7.43 | 38.7 |
| | CB-Theor | 0 | 17.8 | 54.3 | 55.6 | 91.9 | 56.7 | 86.3 | 40.3 | 69.0 | 44.1 | 21.9 | 27.9 | 7.64 | 6.99 | 8.02 | 7.70 | 7.98 | 7.72 | 7.79 | 7.70 | 8.18 | 7.68 | 7.69 | 7.71 |
| | CB-Disc | 0 | 21.1 | 58.8 | 66.4 | 106 | 78.8 | 110 | 70.9 | 106 | 94.4 | 75.1 | 93.0 | 74.7 | 12.1 | 75.6 | 39.2 | 97.1 | 53.1 | 118 | 47.1 | 110 | 74.9 | 61.4 | 92.7 |



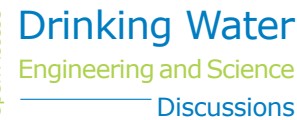

**Table 3(d): Case #4 San Miguel network properties, diameter results (inch nominal sizes, with CB-Theor in inches), and nodal h.**

|          |           | 1-2   | 2-3   | 3-4   | 4-5   | 5-6   | 6-7   | 7-8   | 8-9   | 9-10  | 10-11 |       |
|----------|-----------|-------|-------|-------|-------|-------|-------|-------|-------|-------|-------|-------|
| Network  | Link      | 1-2   | 2-3   | 3-4   | 4-5   | 5-6   | 6-7   | 7-8   | 8-9   | 9-10  | 10-11 |       |
|          | Length (m)| 189   | 168   | 139   | 81    | 32    | 92    | 225   | 115   | 52.3  | 85    |       |
|          | Q (L/s)   | 3.60  | 3.60  | 3.60  | 3.60  | 3.60  | 3.60  | 3.60  | 3.60  | 3.60  | 3.60  |       |
|          | Δz (m)    | 27.4  | 10.7  | -6.4  | 6.1   | -5.2  | -18.6 | 33.2  | 58.2  | -11.3 | 32.9  |       |
| D solutions | BT     | 3     | 3     | 3     | 3     | 3     | 2-½   | 2     | 1-¼   | 1-¼   | 1-¼   |       |
|          | BT-NoUp   | 3     | 3     | 3     | 3     | 3     | 2-½   | 2     | 1-¼   | 1-¼   | 1-¼   |       |
|          | CB-Theor  | 2.939 | 2.929 | 2.929 | 2.929 | 2.929 | 2.929 | 1.671 | 1.462 | 1.462 | 1.368 |       |
|          | CB-Disc   | 3     | 3     | 3     | 3     | 3     | 3     | 2     | 1-½   | 1-½   | 1-¼   |       |
| h (m)    | Node      | 1     | 2     | 3     | 4     | 5     | 6     | 7     | 8     | 9     | 10    | 11    |
|          | BT        | 0     | 25.88 | 35.20 | 27.68 | 33.13 | 27.70 | 7.02  | 28.27 | 43.86 | 13.19 | 14.60 |
|          | BT-NoUp   | 0     | 25.88 | 35.20 | 27.68 | 33.13 | 27.70 | 7.02  | 28.27 | 43.86 | 13.19 | 14.60 |
|          | CB-Theor  | 0     | 25.53 | 34.51 | 26.72 | 32.01 | 26.51 | 7.00  | 6.99  | 32.93 | 6.96  | 7.02  |
|          | CB-Disc   | 0     | 25.88 | 35.20 | 27.68 | 33.13 | 27.70 | 8.37  | 29.62 | 67.54 | 47.02 | 48.43 |



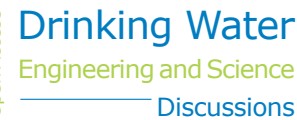

**Table 3(e): Case #5 El Guabo network properties, diameter results (inch nominal sizes, with CB-Theor in inches), and nodal h.**

| | Link | 383 | 486 | 1030 | 600 | 150 | 400 | 187 | 450 | 227 | 230 | 240 | 110 | 270 | 130 | 130 | 260 | 110 | |
|---|---|---|---|---|---|---|---|---|---|---|---|---|---|---|---|---|---|---|---|
| **Network** | Length (m) | 17.72 | 14.68 | 12.76 | 11.96 | 10.04 | 7.72 | 6.60 | 3.12 | 1.20 | 3.04 | 1.92 | 0.80 | 1.92 | 2.32 | 1.12 | 3.48 | 1.92 | |
| | Q (L/s) | 10.9 | 10.0 | -5.6 | 3.2 | -2.6 | 5.7 | -4.1 | 4.2 | -3.1 | 2.0 | 2.5 | -1.2 | 2.0 | -1.1 | 0.0 | 1.0 | 2.0 | |
| | $\Delta z$ (m) | | | | | | | | | | | | | | | | | | |
| | | | | | | | | | | | | | | | | | | | |
| **D solutions** | BT | 8 | 6 | 6 | 6 | 6 | 5 | 5 | 4 | 2 | 2-½ | 1-½ | 1-½ | 2 | 3 | 1-¼ | 3 | 1-½ | |
| | BT-NoUp | 8 | 6 | 6 | 6 | 6 | 5 | 5 | 4 | 2 | 2-½ | 1-½ | 1-½ | 2 | 3 | 1-¼ | 3 | 1-½ | |
| | CB-Theor | 6.88 | 6.41 | 6.14 | 6.01 | 5.69 | 4.8 | 4.58 | 3.49 | 2.65 | 2.36 | 1.61 | 1.53 | 1.93 | 3.25 | 1.4 | 3.08 | 1.65 | |
| | CB-Disc | 8 | 8 | 8 | 6 | 6 | 5 | 5 | 4 | 3 | 2-½ | 1-½ | 1-½ | 2 | 4 | 1-½ | 4 | 2 | |
| | | | | | | | | | | | | | | | | | | | |
| | Node | 1 | 2 | 3 | 4 | 5 | 6 | 7 | 8 | 9 | 10 | 11 | 12 | 13 | 14 | 15 | 16 | 17 | 18 |
| **h (m)** | BT | 0 | 10.34 | 18.50 | 9.91 | 11.53 | 8.61 | 13.16 | 8.65 | 12.11 | 7.25 | 8.48 | 7.23 | 7.35 | 8.84 | 7.03 | 7.16 | 7.68 | 7.80 |
| | BT-NoUp | 0 | 10.34 | 18.50 | 9.91 | 11.53 | 8.61 | 13.16 | 8.65 | 12.11 | 7.25 | 8.48 | 7.23 | 7.35 | 8.84 | 7.03 | 7.16 | 7.68 | 7.80 |
| | CB-Theor | 0 | 9.76 | 18.35 | 9.93 | 11.49 | 8.46 | 12.70 | 7.94 | 10.67 | 7.00 | 7.00 | 7.00 | 7.00 | 7.01 | 7.00 | 7.00 | 7.00 | 7.01 |
| | CB-Disc | 0 | 10.34 | 19.85 | 13.47 | 15.09 | 12.17 | 16.72 | 12.21 | 15.67 | 12.27 | 8.48 | 8.58 | 10.91 | 12.40 | 10.94 | 13.84 | 12.67 | 15.76 |

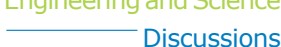



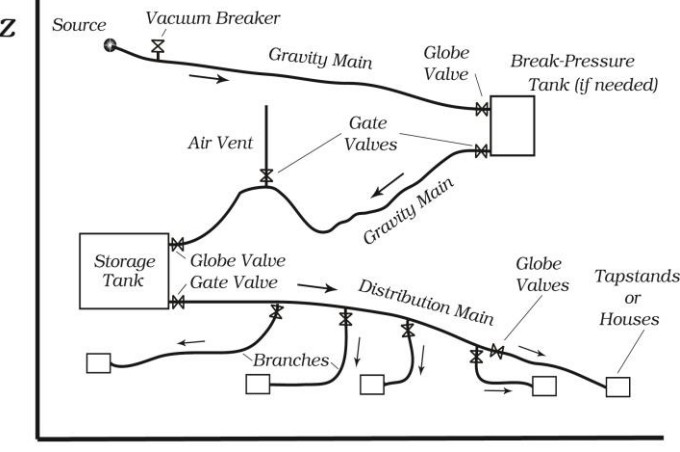

**Figure 1: Element schematic of a GDWN.**

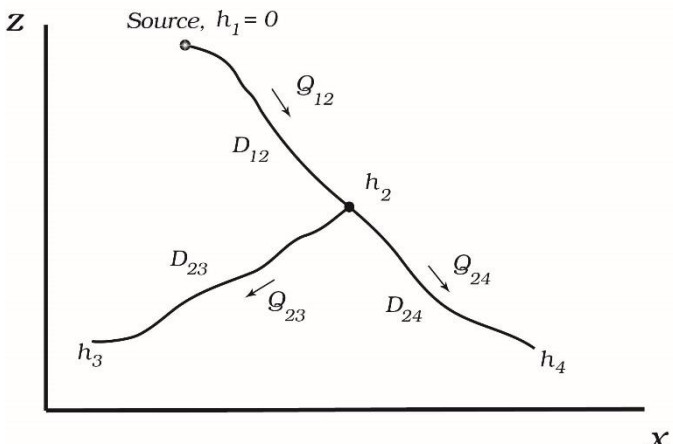

**Figure 2: Three-pipe branching network.**

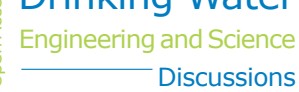



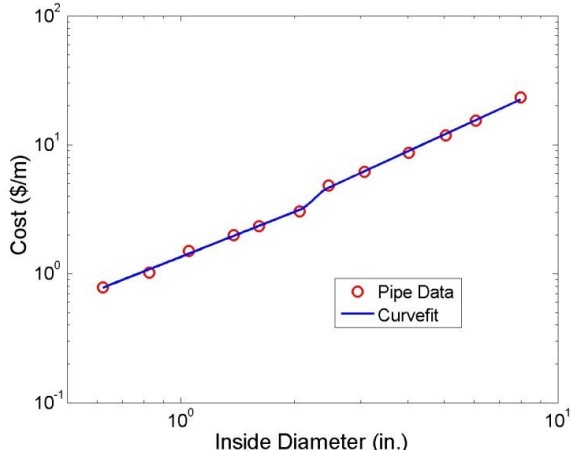

Figure 3: **PVC pipe cost from 2011 data.**



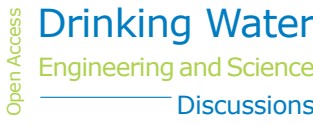

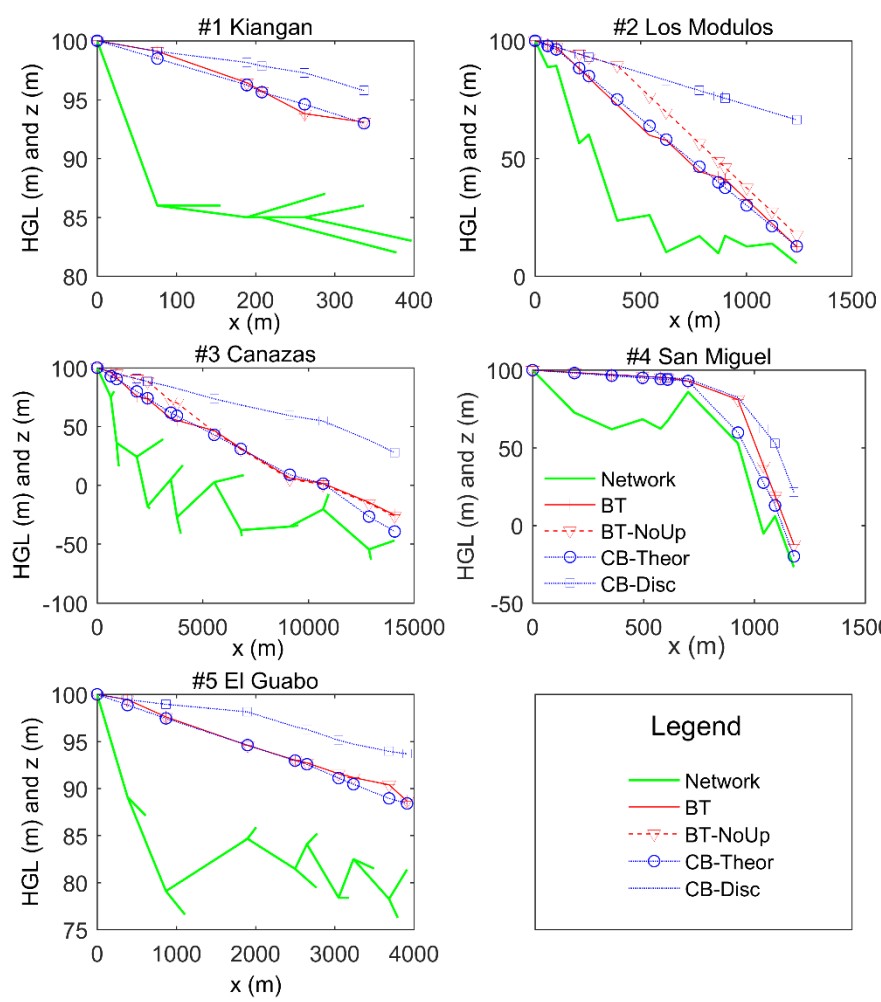

**Figure 4: Elevation and hydraulic grade lines of algorithm solutions for main distribution links.**



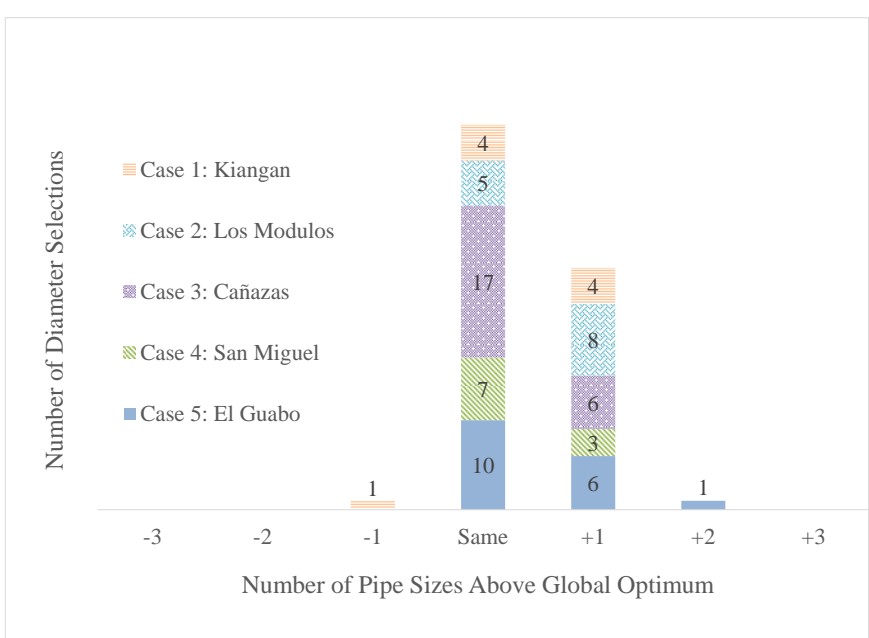

**Figure 5: Diameter sizes of calculus-based (CB-Disc) solutions above the global optimum solutions.**