# Peer review of "Algorithms for Optimization of Branching Gravity-Driven Water Networks"

_Drinking Water Engineering and Science, 2017_

## Referee Comment (RC1) · Anonymous Referee #1 · 15 Mar 2017

**Review Report**

**Paper Title: Algorithm for Optimization of Branching Gravity-Distribution Networks**

**Author: Ian Dardani, G. F. Jones**

Several algorithms are available in literature for optimal design of branched gravity-fed WDNs. Authors have chosen two out of them, a back tracking (BT) algorithm and a genetic algorithm (GA) without giving any proper justification of their selection. The chosen algorithms have been compared with a new calculus-based algorithm. I would like to suggest authors to provide advantage of proposed CB with those available in literature (Deb 1974, Bhave 1978, Chiplunkar and Khanna 1983, Fujiwara and Dey 1988, Young 1994, Johnson et al. 1996). Bhave's (1978) approach is general and applicable to branched as well as looped networks, gravity as well as pumped source networks, and new as well as existing networks. In case of looped networks, primary pipes forming a branching configuration is identified and designed to carry maximum flows by considering secondary loop-forming links of some minimum size.

Using calculus based approach, Bhave (1978) developed an optimal criterion similar to Eq. (9) of authors and expressed as

$$-\sum \left(\frac{bC}{h}\right)_{ij} + \sum \left(\frac{bC}{h}\right)_{jk} = 0$$

Where ij and jk are supply and distribution links at any node j; C and h are cost and head loss in any pipe.

Bhave (1978) suggested a univariant method in which nodal heads are assumed initially and corrected iteratively in order to satisfy the optimal criteria at all nodes. Gupta et al. (2003) improved the method of solution adopting Newton-Raphson method in which all correction values are obtained simultaneously for faster convergence of iterative methodology.

The equation (18) of author seems to be similar to Bhave's optimality criterion, if minor losses are ignored. Authors are requested to clearly point out the difference with Bhave's optimality criteria. Also, a systematic procedure or flow chart should be included to apply the proposed methodology to design water networks.

Even though outcome of the paper is general and nothing new in it, the paper can be recommended if the difference between the proposed CB method with Bhave's CB method is clearly indicated and proposed methodology is explained by giving procedure or flowchart.

**Additional References**

Deb, A. K. (1974). Least Cost Design of Branched Pipe Network Systems. *Journal of Environmental Engineering,* ASCE, 100(4): 821-835.

Bhave, P. R. (1978). Non-computer optimization of single source networks. *Journal of Environmental Engineering,* ASCE, 104(4): 799-814.

Chiplunkar, A. V. and Khanna, P. (1983). Least Cost Design of Branched Pipe Network Systemd. *Journal of Environmental Engineering,* ASCE, 109(3): 604-618.

Fujiwara, O. and Dey, D. (1988). Method of Optimal Design of Branched Network on Flat Terrain. *Journal of Environmental Engineering,* ASCE, 114(6): 1464-1475.

Young, B. (1994). Design of Branched Water Supply Network on Uneven Terrain. *Journal of Environmental Engineering,* ASCE, 120(4): 974-979.

Johnson, S. L., Gupta, R. and Bhave, P. R. (1996). Discussion of "Design of Branched Water Supply Network on Uneven Terrain" by B. Young. *Journal of Environmental Engineering,* ASCE, 122(5): 448.

Gupta, R., Sawarkar, V. R., and Bhave, P. R. (2003). Application of Newton-Raphson method in optimal design of Water Distribution Networks. *Journal of Indian Water Works Association*, 34(1), 31-37.

---

## Referee Comment (RC2) · Anonymous Referee #2 · 22 Mar 2017

1. The manuscript presents algorithms for optimization of branching gravity-driven water networks, which is interesting. The subject addressed is within the scope of the journal. 2. However, the manuscript, in its present form, contains several weaknesses. Appropriate revisions to the following points should be undertaken in order to justify recommendation for publication. 3. Full names should be shown for all abbreviations in their first occurrence in texts. For example, PVC in p.1, D in p.3, WDN in p.4, HGL in p.12, etc. 4. For readers to quickly catch your contribution, it would be better to highlight major difficulties and challenges, and your original achievements to overcome them, in a clearer way in abstract and introduction. 5. It is mentioned in p.1 that three cost-minimization algorithms are adopted for the design of moderate-scale branching Gravity-Driven Water Networks. What are the other feasible alternatives? What are the advantages of adopting these particular algorithms over others in this case? How will

this affect the results? More details should be furnished. 6. It is mentioned in p.4 that five actual GDWNs installed in Panama, Nicaragua, and the Philippines are adopted as test cases. What are the other feasible alternatives? What are the advantages of adopting these particular test cases over others in this case? How will this affect the results? More details should be furnished. 7. It is mentioned in p.5 that ". . .The pressure upper bound is not incorporated into the optimization process. . . ." Why this is not incorporated? More justifications should be furnished on this. 8. It is mentioned in p.5 that ". . .precautions against water hammer are left to the designer. . ." Why this is left to the designer? More justifications should be furnished on this. 9. It is mentioned in p.7 that a two-part model is adopted in this study. What are the other feasible alternatives? What are the advantages of adopting this particular model over others in this case? How will this affect the results? More details should be furnished. 10. It is mentioned in p.9 that a cubic spline is adopted to complete the transition between small and large pipe sizes. What are the other feasible alternatives? What are the advantages of adopting this particular function over others in this case? How will this affect the results? More details should be furnished. 11. It is mentioned in p.13 that two rejection criteria are adopted to reduce the number of solutions to be evaluated. What are the other feasible alternatives? What are the advantages of adopting these particular criteria over others in this case? How will this affect the results? More details should be furnished. 12. It is mentioned in p.14 that the method presented by González-Cebollada et al. (2011) is adopted as the BT algorithm. What are the other feasible alternatives? What are the advantages of adopting this particular method over others in this case? How will this affect the results? More details should be furnished. 13. It is mentioned in p.15 that a single-point crossover is adopted in this study. What are the other feasible alternatives? What are the advantages of adopting this particular crossover method over others in this case? How will this affect the results? More details should be furnished. 14. It is mentioned in p.15 that a pinwheel lottery (with replacement) is adopted to select candidates to be carried into the next generation. What are the other feasible alternatives? What are the advantages of adopting this particular approach over others in this case?

How will this affect the results? More details should be furnished. 15. It is mentioned in p.18 that "...and several of these would likely improve upon the GA results obtained in this study ..." Why they are not performed in this study? More justifications should be furnished on this. 16. Some key parameters are not mentioned. The rationale on the choice of the particular set of parameters should be explained with more details. Have the authors experimented with other sets of values? What are the sensitivities of these parameters on the results? 17. Some assumptions are stated in various sections. Justifications should be provided on these assumptions. Evaluation on how they will affect the results should be made. 18. The discussion section in the present form is relatively weak and should be strengthened with more details and justifications. 19. Moreover, the manuscript could be substantially improved by relying and citing more on recent literatures about real-life case studies of contemporary soft computing techniques in water resources engineering such as the followings: ïĄň Gholami, V., et al., "Modeling of groundwater level fluctuations using dendrochronology in alluvial aquifers", Journal of Hydrology 529 (3): 1060-1069 2015. ïĄň Taormina, R., et al., "Data-driven input variable selection for rainfall-runoff modeling using binary-coded particle swarm optimization and Extreme Learning Machines", Journal of Hydrology 529 (3): 1617-1632 2015. ïĄň Wu, C.L., et al., "Methods to improve neural network performance in daily flows prediction," Journal of Hydrology 372 (1-4): 80-93 2009. ïĄň Wang, W.C., et al., "Improving forecasting accuracy of annual runoff time series using ARIMA based on EEMD decomposition," Water Resources Management 29 (8): 2655-2675 2015. ïĄň Chen, X.Y., et al., "A Novel Hybrid Neural Network based on Continuity Equation and Fuzzy Pattern-recognition for Downstream Daily River Discharge Forecasting," Journal of Hydroinformatics 17 (5): 733-744 2015. ïĄň Saeidifarzad, B., et al., "Multi-site calibration of linear reservoir based geomorphologic rainfall-runoff models", Water 6 (9): 2690-2716 2014. 20. In the conclusion section, the limitations of this study, suggested improvements of this work and future directions should be highlighted.

---

## Referee Comment (RC3) · Anonymous Referee #3 · 14 Apr 2017

This paper looks at the classic problem of pipe diameter selection for branched gravity fed water distribution networks. Specifically it only allows for a single pipe diameter for each link in the network. It looks at three alternative approaches to the problems and compares them over 5 sample networks. A primary concern with the work is that the authors mention in the abstract "three cost-minimization algorithms are developed". But these algorithms are well known techniques as the authors themselves refer to the related past work. It is not made very explicit, what is the new contribution provided by the authors.

If indeed the primary purpose of the paper is to contrast existing techniques on different networks, this purpose should be made more explicit and clear. Further, why was the linear programming approach not considered? It is significantly faster for the problem in question than all the three approaches discussed and provides optimal results.

[Figure]

In a paper where the primary purpose is to test different approaches over different networks, it is a big shortcoming that the range of networks is so small, with the largest network only consisting of 23 links. I recommend including larger networks for the testing.

The problem statement being look at is a single pipe diameter for each link in the network. This is known to be NP hard even when restricted to branched networks [1]. But the more practical problem is one where multiple diameters are allowed (specifically upto two diameters in the optimal case [2]). Also the more general problem is "easier" since it can be solved by an ILP and thus is solved in polynomial time. Given that the allowing multiple diameters is more general, practical, easier and provides lower costs, why was it not considered except for a brief mention in the paper.

Some further specific points about the paper:

1. For the calculus based approach, the authors mention that "Mapping between continuous diameters and the discrete nominal sizes, required to complete the design, will not be addressed in the present work." But this is a non-trivial aspect of the problem and simply taking the nearest larger diameter can significantly impact the cost of the design as the authors themselves note in the results.

2. While the times for the backtracking and genetic algorithms has been reported in the results, there are no exact numbers provided for the CB algorithm. Only a qualitative comment is made as regard to its better scaling. As such why was a larger network not included in the testing process to make the comparison more explicit? Also the tabulation of costs for the different approaches across the sample networks should include the time taken, which will provided a better picture of the trade-off involved in choosing non optimal approaches.

References:

[1] : Yates, D.F., Templeman, A.B., and Boffey, T.B., 1984, "The Computational Complexity of the Problem of Determining Least Capital Cost Designs for Water Supply Networks", Engineering Optimization, 7(2), 142-155

[2] : Fujiwara, Okitsugu, and Debashis Dey., 1987, "Two adjacent pipe diameters at the optimal solution in the water distribution network models." Water Resources Research 23.8: 1457-1460

---

## Author Comment (AC1) · 27 May 2017

This paper looks at the classic problem of pipe diameter selection for branched gravity fed water distribution networks. Specifically, it only allows for a single pipe diameter for each link in the network. It looks at three alternative approaches to the problems and compares them over 5 sample networks. A primary concern with the work is that the authors mention in the abstract "three cost-minimization algorithms are developed". But these algorithms are well known techniques as the authors themselves refer to the related past work. It is not made very explicit, what is the new contribution provided by the authors.

Author's response:

1. We have changed the paper to read "…three cost-minimization algorithms are presented and results compared with five GDWN test cases."
2. We have added the new contributions to the Abstract.

If indeed the primary purpose of the paper is to contrast existing techniques on different networks, this purpose should be made more explicit and clear. Further, why was the linear programming approach not considered? It is significantly faster for the problem in question than all the three approaches discussed and provides optimal results. In a paper where the primary purpose is to test different approaches over different networks, it is a big shortcoming that the range of networks is so small, with the largest network only consisting of 23 links. I recommend including larger networks for the testing.

Author's response:

1. The problem considered in this paper is that of selecting a single pipe diameter for each link in a water network to minimize the material cost. This problem has three major categories of methods that are applicable to it: enumeration methods (including both complete enumeration and partial enumeration), nonlinear programming methods, and metaheuristic methods. For each of these categories we have proposed and tested one representative algorithm: backtracking (partial enumeration), the Jones calculus-based algorithm (nonlinear programming), and a genetic algorithm (metaheuristic). While there are many other types of metaheuristic algorithms (simulated annealing, Tabu search, cellular automata, ant colony optimization, and particle swarm optimization), the genetic algorithm is the most representative of these and is also the most commonly used (Zhao et al. 2016).

2. Note that we have categorized the backtracking algorithm as a partial enumeration method and not a heuristic algorithm. The backtracking algorithm, like heuristic methods, follows a set of deterministic rules to find better solutions, however, those rules are strictly formulated to find cost-optimal solutions and does so without missing the global optimum. In contrast, heuristic algorithms follow rules which achieve some proxy of an optimum solution but do not guarantee the global optimum. One example of a heuristic algorithm comes from Suribabu (2012), whose algorithm uses the uniformity of a solution's flow velocity as a proxy for its cost-optimality. As such, the Suribabu algorithm increments a pipe diameter when its flow velocity is high and

decrements the diameter when its flow velocity is low, in an attempt to approach more optimal solutions. It was not necessary to include such a heuristic algorithm for comparison in this paper, since the Backtracking algorithm presented follows a more strictly formulated set of rules that do guarantee a cost optimum. It should be noted, and now is done so more clearly in the paper in section 4.3, that BT scales poorly with larger network sizes and would not be appropriate for use on large urban networks. However, the present BT algorithm reduces the search space even more than previous partial enumeration methods without risk of pruning the global optimum, primarily through Pre-Processors 1 and 2. This technique proved successful on the cases tested, and in the context of GDWN design the larger network cases are in the higher range of what would be expected of a GDWN in practice. In fact, the network in Cañazas, Panama, on which Test Case 3 is based, is the largest GDWN in Central and South America. These example networks have been gathered from our GDWN design work in Panama, Nicaragua, and the Philippines.  Thus, these networks are most pertinent to our interests and those of other GDWN designers. Given the scant coverage of GDWNs in the literature, these networks are among the few for which we have detailed data.

Suribabu, C. R.: Heuristic-based pipe dimensioning model for water distribution networks, J. Pipeline Syst. Eng. Pract., 3(4), 115–124, doi:10.1061/(ASCE)PS.1949-1204.0000104, 2012.
Zhao, W., Beach, T., and Rezgui, Y.: Optimization of Potable Water Distribution and Wastewater Collection Networks: A Systematic Review and Future Research Directions, IEEE Transactions on Systems, Man, and Cybernetics: Systems, 46 (5), 659-681, doi:10.1109/TSMC.2015.2461188, 2016.

3. We note that other methods that can be applied to water network design do not address the problem of focus in our paper.  For example multi-objective optimization methods, which typically involve new implementations of metaheuristic methods, are outside the scope of (single-objective) cost optimization, although it should be noted that cost-optimization algorithms, such as the ones used in this paper, can be used within multi-objective implementations. In addition, decomposition methods, where networks are broken down into smaller sub-networks, can use any of the methods listed above, and are therefore not an exclusive method category. Given the appropriateness of the presented algorithms to the network sizes of gravity water networks, decomposition was not necessary. Lastly, linear programming methods, while providing an efficient algorithm design, do provide split-pipe solutions for each link, while the problem formulation here calls for each link to have a single diameter solution. While it is true that split-pipe solutions are often a satisfactory result, the focus on solutions with single-diameters is a meaningful area of research, as evidenced by the significant body of research dedicated to this problem formulation. Some of the reasons why linear programming can be less desirable include the practicality of dealing with split-pipe solutions, their path-dependent optimality, and the risk of converging on local optima particularly with poor initial guesses, which are referenced in more detail by by Zhao (2016).

Of course, the demand-driven (or $Q$-specified) design problem that we consider is nonlinear because of the appearance of the dependent variables $D_{ij}$ and $h_j$ in the energy (Equation (1)) and cost minimization equations (Equations (15) or (18)) in our paper, respectively. Iterative LP was not used in our work. Algorithms to solve a general set of independent, nonlinear algebraic equations using, for example, the Levenberg-Marquardt, Quasi-Newton, Newton-Raphson, or Conjugate Gradient methods are available in most commercial math packages including Matlab and Mathcad (we use the latter with the Quasi-Newton method).

We have added a more thorough description to our introduction section that gives clearer context to the selection of these algorithms, highlighting the category of method to which they belong and the key features that make them distinct from one another.

Zhao, W., Beach, T., and Rezgui, Y.: Optimization of Potable Water Distribution and Wastewater Collection Networks: A Systematic Review and Future Research Directions, IEEE Transactions on Systems, Man, and Cybernetics: Systems, 46 (5), 659-681, doi:10.1109/TSMC.2015.2461188, 2016.

The problem statement being look at is a single pipe diameter for each link in the network. This is known to be NP hard even when restricted to branched networks [1]. But the more practical problem is one where multiple diameters are allowed (specifically up to two diameters in the optimal case [2]). Also, the more general problem is "easier" since it can be solved by an ILP and thus is solved in polynomial time. Given that the allowing multiple diameters is more general, practical, easier and provides lower costs, why was it not considered except for a brief mention in the paper.

Some further specific points about the paper:

1. For the calculus based approach, the authors mention that "Mapping between continuous diameters and the discrete nominal sizes, required to complete the design, will not be addressed in the present work." But this is a non-trivial aspect of the problem and simply taking the nearest larger diameter can significantly impact the cost of the design as the authors themselves note in the results.

2. While the times for the backtracking and genetic algorithms has been reported in the results, there are no exact numbers provided for the CB algorithm. Only a qualitative comment is made as regard to its better scaling. As such why was a larger network not included in the testing process to make the comparison more explicit? Also, the tabulation of costs for the different approaches across the sample networks should include the time taken, which will provide a better picture of the trade-off involved in choosing non-optimal approaches.

Author's response:

1. We did not find the need to model dual pipe sizes per link because we were able to obtain convergence for all cases with a single pipe size. However, the mapping between continuous diameters and the discrete nominal sizes was accomplished in our solution by one of the following ways:
   a. For small and moderate size networks the designer may manually adjust the pipe sizes (downward, normally one pipe size) starting from the first link downstream from the

source and continuing along the rest of the distribution main to the end. A nearby plot of the static pressure heads compared with the theoretical $D_{ij}$ from our CB approach (on the same Mathcad page) will highlight the acceptability or unacceptability of any change. This exercise also gives the designer an understanding of the sensitivity of the design to small changes in pipe sizes.

    b. Based on the theoretical $D_{ij}$ from the CB approach, a composite pipeline can be created for each link after each link of diameter $D_{ij}$ is calculated. That is, the lengths for the two discrete pipes sizes that bound the theoretical $D_{ij}$ from above and below are calculated such that the pressure drop between two consecutive nodes in the distribution main matches between the composite pipeline and the CB approach. This also provides discrete pipe sizes that nearly matches the CB solution in terms of cost.

    c. We have included these two bullets in the paper.

2. The focus of this work was prompted by our GDWN design work in Panama, Nicaragua, and the Philippines, with the primary focus on finding designs of minimal cost. The algorithms we compared on solution cost were all able to find solutions in less than a few minutes, which does not present a meaningful burden on a designer, unlike many large urban networks which could take hours for various optimization algorithms to complete. The larger networks used in this paper are among the higher end that would be expected for a GDWN design. In fact, the network in Cañazas, Panama, on which Test Case 3 is based, is the largest GDWN in Central and South America. Thus, the ability of the algorithms to scale beyond this upper end of the cases tested are a lesser concern than the cost comparison is on the cases tested. However, for clarity and consistency we have incorporated these explanations to your comments into the paper and have referred to the CB algorithm's computational time.

3. Thank you for your review of our paper.

Reviewer References:

[1] : Yates, D.F., Templeman, A.B., and Boffey, T.B., 1984, "The Computational Complexity of the Problem of Determining Least Capital Cost Designs for Water Supply Networks", Engineering Optimization, 7(2), 142-155

[2] : Fujiwara, Okitsugu, and Debashis Dey., 1987, "Two adjacent pipe diameters at the optimal solution in the water distribution network models." Water Resources Research 23.8: 1457-1460 Interactive comment on Drink. Water Eng. Sci. Discuss., doi:10.5194/dwes-2017-7, 2017.

[revised manuscript text omitted]
} \le D_{co} \\ c_1 + c_2 \dfrac{D_{ij}}{D_u} + c_3 \left(\dfrac{D_{ij}}{D_u}\right)^2 + c_4 \left(\dfrac{D_{ij}}{D_u}\right)^3, & D_{co} < D_{ij} < D_{co+1} \\ a_\ell (\dfrac{D_{ij}}{D_u})^{b_\ell}, & D_{ij} \ge D_{co+1} \end{cases} \tag{17}$$

5 In Eq. (17), $a_s$ and $a_\ell$ are the coefficients for the small and large pipe size regions, respectively, and $b_s$ and $b_\ell$ are the exponents for the small and large pipe size regions, respectively. A cubic spline is fit between pipe sizes $D_{co}$ and $D_{co+1}$ to complete the transition between small and large pipe sizes. The coefficients of this polynomial are $c_1$, $c_2$, $c_3$, and $c_4$ as seen in Eq. (17). These coefficients are evaluated by matching the cubic polynomial and pipe data at $D_{co}$ and $D_{co+1}$ and the first derivative of the polynomial with respect to $D_{ij}/D_u$ to $a_s b_s (\frac{D_{co}}{D_u})^{b_s-1}$ at $D_{ij} = D_{co}$ and to $a_\ell b_\ell (\frac{D_{co+1}}{D_u})^{b_\ell-1}$ at $D_{ij} = D_{co+1}$. An

10 example of data for Polyvinyl chloride (PVC) pipe and the curvefit is shown in Fig. 3. The results of the curvefit are: $D_{co} = 2.067$ in., $D_{co+1} = 2.469$ in., $a_s = \$1.349$ m$^{-1}$, $b_s = 1.157$, $a_\ell = \$1.381$ m$^{-1}$, $b_\ell = 1.344$, $c_1 = \$237.516$ m$^{-1}$, $c_2 = -\$316.125$ m$^{-1}$, $c_3 = \$140.450$ m$^{-1}$, $c_4 = -\$20.499$ m$^{-1}$. It is clear from inspection of Fig. 3 that a one-part cost model would not have produced an acceptable curve-fit to pipe-cost data.

With the inclusion of the two-part cost model and minor loss term, Eq. (15) becomes

$$0 = \sum_{ij,in} \frac{C'_{ij} A_{ij}^{\frac{4}{19}} (1+\epsilon_{ij})^{\frac{4}{19}} S_{ij}^{-\frac{23}{19}} \left(\dfrac{Q_{ij}^7 \nu}{g^4 D_u^{19}}\right)^{\frac{1}{19}}}{1 - B A_{ij}^{\frac{4}{19}} \epsilon_{ij}'(1+\epsilon_{ij})^{-\frac{15}{19}} S_{ij}^{-\frac{4}{19}} \left(\dfrac{Q_{ij}^7 \nu}{g^4 D_u^{19}}\right)^{\frac{1}{19}}}$$
$$- \sum_{ij,out} \frac{C'_{ij} A_{ij}^{\frac{4}{19}} (1+\epsilon_{ij})^{\frac{4}{19}} S_{ij}^{-\frac{23}{19}} \left(\dfrac{Q_{ij}^7 \nu}{g^4 D_u^{19}}\right)^{1/19}}{1 - B A_{ij}^{\frac{4}{19}} \epsilon_{ij}'(1+\epsilon_{ij})^{-\frac{15}{19}} S_{ij}^{-\frac{4}{19}} \left(\dfrac{Q_{ij}^7 \nu}{g^4 D_u^{19}}\right)^{1/19}} \tag{18}$$

15 where $B = 0.1989$ and

$$\epsilon_{ij} = \sum_k \left(\dfrac{L_e}{D}\right)_{k,ij} \dfrac{D_{ij}}{L_{ij}} \tag{19}$$

$$\epsilon_{ij}' = \sum_k \left(\frac{L_e}{D}\right)_{k,ij} \frac{D_u}{L_{ij}}$$

$$A_{ij} = \begin{cases} 0.318, & \text{smooth pipe} \\ 0.420, & \text{steel pipe} \end{cases}$$

(20)

and $A$ accounts for the effect of pipe roughness (smooth and commercial steel). The term $C'_{ij}$ is the derivative of the cost function per unit length with respect to $D/D_u$. For the two-part cost model from above, obtain

$$C'_{ij} = \begin{cases} a_s b_s (\frac{D_{ij}}{D_u})^{b_s-1}, & D_{ij} \leq D_{co} \\ c_2 + 2c_3(\frac{D_{ij}}{D_u}) + 3c_4(\frac{D_{ij}}{D_u})^2, & D_{co} < D_{ij} < D_{co+1} \\ a_\ell b_\ell (\frac{D_{ij}}{D_u})^{b_\ell-1}, & D_{ij} \geq D_{co+1} \end{cases}$$

(21)

Equation (18), and its simpler form Eq. (15), forms the basis for calculus-based optimization in this work and is applied at all internal nodes to uniquely determine $h_j$. Equation (18) is valid over the range of ~4000 < Re < ~300,000. Algorithms to solve a general set of independent, nonlinear algebraic equations using, for example, the Levenberg-Marquardt, Quasi-Newton, Newton-Raphson, or Conjugate Gradient methods are available in most commercial math packages including Matlab (1 Apple Hill Drive, Natick, MA USA 01760) and Mathcad (http://www.ptc.com). We used the package Mathcad in the present work. Thus, compared with an iterative solution procedure, a solution flowchart is not relevant here.

Bhave (1978) first proposed an algorithm like Eq. (15). However, Bhave used an iterative method to solve the design problem. As such, there are several qualifications leading up to the cost minimization equation in Bhave. These include the assumption of smallness in variation of the static pressure head between two iterations. This allowed the terms in the cost function to be approximated as constants. In the present work, the cost-function coefficient and exponent are not assumed constant at any node joining two sets of links; see Equation (18). Nor do we make any assumptions on the orders of magnitude of the terms in our equations to simplify them. For clarity, we re-present Eq. (15) using Bhave's (1978) notation as

$$0 = \sum Q_{ij}^{7b/19} S_{ij}^{-(1+4b/19)} - \sum Q_{jk}^{7b/19} S_{jk}^{-(1+4b/19)}$$

where the $ij$ and $jk$ notation are shown in Fig. 4. Index $j$ spans all internal nodes along the distribution main.

[revised manuscript text omitted]

Two backtracking methods can be found in the literature, namely those by Gessler (1985) and González-Cebollada et al. (2011). The algorithm proposed by Gessler, however, also proposes a pipe-grouping criteria that risks pruning the global optimum and represents a tricky optimization problem in and of itself (Raad 2011). The González-Cebollada algorithm, on the other hand, does not include such criteria to potentially prune a global optimum from consideration, but it halts its search after finding the first feasible solution. In contrast, the BT algorithm in the present study guarantees a global optimum by continuing its search of the solution tree even after the first solution has been found. In addition, the present BT algorithm utilizes Pre-Processors 1 and 2 to further reduce the search space, without risk of pruning the global optimum, in advance of its search routine. Thus, out of the two reported backtracking algorithms in the literature, both do not guarantee a global optimum, while the BT algorithm presented in this work does. It should be noted that BT is known to scale poorly with large network sizes and would not be appropriate for use on large urban networks, though its appropriateness is demonstrated here for GDWNs, given that the test cases used in this paper representative of the sizes of GDWNs that would be expected in practice.

[revised manuscript text omitted]
. While a two-point crossover technique was considered, the results were not found to have any benefit over a single-point technique, which was chosen for its greater simplicity. The fitness, $f_i$, of each candidate is assessed with penalties associated with the solution's pipe cost, $C_{pipe,i}$, and violations of the static pressure head requirement, $C_{hyd,i}$, or

$$f_i = \frac{1}{C_{pipe,i} + C_{hyd,i}} \tag{29}$$

The hydraulic cost is obtained for each individual by identifying nodes in which the static pressure head is less than $h_{min}$ and multiplying the total amount of head violation by a hydraulic penalty coefficient, $a_{hyd}$:

$$C_{hyd,i_c} = a_{hyd} \sum_1^{N_L} \left(h_{min} - h_{i_N}\right) \mid h_{i_N} < h_{min} \tag{30}$$

To allow for a hydraulic penalty coefficient to produce similar results in both small-scale (inexpensive) network and a large-scale (more expensive) cases, the hydraulic penalty coefficient is made directly proportional to the average solution

cost. With each generation, $a_{hyd}$ is updated by multiplying the normalized penalty coefficient, $a_{hyd,norm}$, by the average pipe cost of the population,

$$a_{hyd} = a_{hyd,norm} \frac{\sum_1^{N_c} C_{pipe,i_C}}{N_c} \tag{31}$$

The algorithm then selects candidates to be carried into the next generation through a proportionate selection method, where each candidate has a probability of being selected, $p_{sel,i}$ in direct proportion to its fitness relative to the sum of all fitness values in the population

$$p_{sel,i} = \frac{f_i}{\sum_1^{N_c} f_i} \tag{32}$$

The algorithm replaces the parent generation with a generation of equal size and tends to select more fit individuals in successive generations. In this study, the genetic algorithm parameters used were $p_{mut} = 0.02$, $N_c = 50$, $p_{xover} = 0.5$, $N_{gen} = 100$, $a_{hyd,norm} = 0.1$. The first four of these parameters were chosen based on typical values presented in the literature and then tuned with a sensitivity analysis for the first test case. Simspon et al. (1994) present typical values for $N_c$ (30 - 200), $p_{xover}$ (0.7 - 1.0), $p_m$ (0.01 - 0.05), and $N_{gen}$ (100 - 1000). The normalized hydraulic penalty coefficient, $a_{hyd,norm}$, was chosen such that the GA converged on solutions which tended to satisfy the minimum static pressure constraint, but still allowed the population to gravitate towards smaller diameters with static pressures close to $h_{min}$.

**5  Cases Studied**

Five cases were studied based on actual GDWN in Panama, Nicaragua, and the Philippines. Global characteristics of each network are presented in Table 1 and the details of each network are presented in Table 3(a)-(e). Each network is a branching type without loops. The total lengths of the networks range from less than 1 km to over 15 km. Two serial networks are tested to demonstrate the effect of a local high point on the algorithm solutions. Elevation plots for each case are shown in Fig. 5.

The choice of $h_{min}$ is not standardized, and should appropriately balance the risk of negative pressure in pipes and the increase in network cost due to the requirement of using larger diameters. The choice of $h_{min}$ in GDWN design is typically in the range of 5 m – 20 m (Arnalich 2010; Bouman 2014; Swamee and Sharma 2008). In the present study $h_{min} = 7$ m, although this requirement was reduced at selected nodes at the beginning of networks where changes in elevation are still small. At the source node, the static pressure head is fixed at atmospheric pressure. All cases assumed minor-lossless flow, although all algorithms (e.g., Eq. (18) for CB-Theor) are capable of handling minor loss coefficients through the equivalent length method as presented above.

**6  Mapping the Theoretical $D$ to Discrete Pipe Sizes**

The mapping between continuous diameters and the discrete nominal pipe sizes was accomplished in our solution by one of the following ways:

1. For small and moderate size networks, the designer may manually adjust the pipe sizes (downward, normally one pipe size) starting from the first link downstream from the source and continuing along the rest of the distribution main to the end.  A nearby plot of the static pressure heads compared with the theoretical $D_{ij}$ from our CB approach (on the same Mathcad page) will highlight the acceptability or unacceptability of any change.  This exercise also gives the designer an understanding of the sensitivity of the design to small changes in pipe sizes.

2. Based on the theoretical $D_{ij}$ from the CB approach, a composite pipeline can be created for each link. That is, the lengths for the two discrete pipes sizes that bound the theoretical $D_{ij}$ from above and below are calculated such that the pressure drop between two consecutive nodes in the distribution main matches between the composite pipeline and the CB approach.  This also provides discrete pipe sizes that nearly matches the CB solution in terms of cost.

[revised manuscript text omitted]
. 1996) were attempted in a less systematic way (Ie. a parameter tuning study and three attempts at scaling the fitness function with an increasing exponent), but these did not result in a noticeable on performance. However, it is possible that if these techniques were followed systematically in full, the GA performance may have been improved. Still, the GA algorithm presented has undergone

10   reasonable attempts to adapt its design and parameters for real-world GDWN cases, and therefore presents a useful point of comparison to the BT and CB algorithms.

**8   Conclusions**

Algorithms to optimize the cost of branching gravity-driven water networks are evaluated on five test cases from real networks in the Philippines, Nicaragua, and Panama. A calculus-based algorithm produced a solution composed of theoretical

15   diameters from a continuous set (CB-Theor), which are then mapped onto discrete commercially available diameters (CB-Disc). Backtracking (BT), a recursive algorithm, systematically searches discrete candidate solutions and is guaranteed to find the global optimum by following rules that prune only higher-cost or hydraulically infeasible candidates. The BT algorithm was modified (BT-NoUp) to improve computational speed by also rejecting all candidates that included a small diameter directly upstream of a larger diameter. This criterion allowed BT-NoUp to prune more candidate solutions but allowed for the

20   possibility of missing the global optimum. The third type of algorithm evaluated was a genetic algorithm (GA) that used single-point crossover and proportionate selection

BT was able to find the global optimum in all test cases with relatively little computational effort, and could be applied to other GDWNs composed of a similar number of links. In addition, while BT-NoUp completed its search in less time than BT, the time required to complete BT would not be burdensome on a designer and therefore BT-NoUp did not produce a

25   compelling relative advantage over BT. BT, however, could become prohibitively time-consuming when dealing with networks with significantly more links, as would be the case with large urban networks. While the test cases represent the range of GDWN sizes encountered in the authors' experience, future work would be needed to verify the suitability of the BT and BT-NoUp algorithms on other large GDWNs. The calculus-based algorithm produced consistently good results for the networks tested, although a more robust mapping scheme from theoretical diameters to discrete diameters would further

30   improve on these results as discussed above. In potential future work, the CB-Theor solutions could be used to prune the BT search space, similar to Kadu et al. (2008), by only including the two diameters above and below the CB-Theor diameters,

producing four diameter choices per link. The calculus-based methodology provides an additional benefit to the designer by explicitly revealing the sensitivities to cost for a design. The calculus-based algorithm requires greater computational effort than backtracking for smaller networks, however, this effort scales more linearly with the number of network links, while backtracking scales exponentially. Furthermore, backtracking's computational time is sensitive to the number of available

5     diameters. Still, when applied to GDWNs with a similar number of links to the test cases, backtracking can quickly find a global optimum. In addition, because it is guaranteed to find the global optimum, it can be useful for benchmarking the performance of other algorithms which scale better with more network links. While the genetic algorithm produced solutions with decent closeness to the global optimum, run-to-run results vary due to the stochastic nature of the algorithm. Overall, the genetic algorithm as implemented did not produce results which deemed it compelling over deterministic methods as applied

10     to GDWNs. However, for more complex networks and problem formulations, a genetic algorithm may be more advantageous. In this case, the present study's GA could be greatly improved on through many improvements reported in the literature (Nicklow et al. 2010).

      For all test cases, the calculus-based algorithm's theoretical diameter solutions (CB-Theor) produced a lower cost than the discrete-domain global optimum. This result is made possible because of it is not constrained to a discrete set of

15     diameters. As such, the *CB-Theor results represent a lower-bound on the optimum solution within the problem formulation*, which could be approached with a finer selection of pipe diameters.

[revised manuscript text omitted]

**Table 3(b): Case #2 Los Modulos network properties, diameter (D) results (inch nominal sizes, with CB-Theor in inches), and nodal h.**

| Network | | 1-2 | 2-3 | 3-4 | 4-5 | 5-6 | 6-7 | 7-8 | 8-9 | 9-10 | 10-11 | 11-12 | 12-13 | 13-14 |
|---|---|---|---|---|---|---|---|---|---|---|---|---|---|---|
| | Link | 1-2 | 2-3 | 3-4 | 4-5 | 5-6 | 6-7 | 7-8 | 8-9 | 9-10 | 10-11 | 11-12 | 12-13 | 13-14 |
| | Length (m) | 60 | 41 | 108 | 46 | 134 | 153 | 79 | 157 | 90 | 32 | 102 | 120 | 117 |
| | Q (L/s) | 0.39 | 0.39 | 0.39 | 0.39 | 0.39 | 0.39 | 0.39 | 0.39 | 0.39 | 0.39 | 0.39 | 0.39 | 0.39 |
| | $\Delta z$ (m) | 11.2 | -0.5 | 32.8 | -3.7 | 36.6 | -2.3 | 15.7 | -6.8 | 7.3 | -7.4 | 4.5 | -1.2 | 8.4 |

| D solutions | | | | | | | | | | | | | | |
|---|---|---|---|---|---|---|---|---|---|---|---|---|---|---|
| | BT | 1 | 1 | ¾ | ¾ | ¾ | ¾ | 1 | ¾ | 1 | 1 | ¾ | ¾ | ¾ |
| | BT-NoUp | 1 | 1 | 1 | 1 | 1 | ¾ | ¾ | ¾ | ¾ | ¾ | ¾ | ¾ | ¾ |
| | CB-Theor | 0.987 | 0.984 | 0.849 | 0.849 | 0.849 | 0.849 | 0.849 | 0.849 | 0.849 | 0.849 | 0.849 | 0.849 | 0.849 |
| | CB-Disc | 1 | 1 | 1 | 1 | 1 | 1 | 1 | 1 | 1 | 1 | 1 | 1 | 1 |

| h (m) | | 1 | 2 | 3 | 4 | 5 | 6 | 7 | 8 | 9 | 10 | 11 | 12 | 13 | 14 |
|---|---|---|---|---|---|---|---|---|---|---|---|---|---|---|---|
| | Node | 1 | 2 | 3 | 4 | 5 | 6 | 7 | 8 | 9 | 10 | 11 | 12 | 13 | 14 |
| | BT | 0 | 9.55 | 7.94 | 31.59 | 24.00 | 49.25 | 33.99 | 47.55 | 27.45 | 32.32 | 24.05 | 19.91 | 8.55 | 7.04 |
| | BT-NoUp | 0 | 9.55 | 7.94 | 37.82 | 32.88 | 65.85 | 50.59 | 59.60 | 39.50 | 39.18 | 29.07 | 24.93 | 13.56 | 12.05 |
| | CB-Theor | 0 | 9.00 | 7.00 | 31.86 | 24.78 | 51.53 | 37.98 | 47.87 | 29.53 | 30.21 | 20.46 | 17.46 | 7.44 | 7.23 |
| | CB-Disc | 0 | 9.55 | 7.94 | 37.82 | 32.88 | 65.85 | 59.41 | 72.98 | 61.93 | 66.80 | 58.53 | 60.27 | 55.83 | 61.06 |

**Table 3(c): Case #3 Cañazas network properties, diameter (D) results (inch nominal sizes, with CB-Theor in inches), and nodal h.**

| Network | Link | 1-2 | 2-3 | 3-4 | 4-5 | 5-6 | 6-7 | 7-8 | 8-9 | 9-10 | 10-11 | 11-12 | 12-13 | 2-14 | 3-15 | 4-16 | 5-17 | 6-18 | 7-19 | 8-20 | 9-21 | 10-22 | 11-23 | 12-24 |
|---|---|---|---|---|---|---|---|---|---|---|---|---|---|---|---|---|---|---|---|---|---|---|---|---|
| | Length (m) | 646 | 275 | 957 | 509 | 1102 | 291 | 1764 | 1256 | 2320 | 1580 | 2170 | 1217 | 160 | 100 | 1250 | 110 | 570 | 180 | 1400 | 50 | 400 | 260 | 100 |
| | Q (L/s) | 6.29 | 5.49 | 5.39 | 5.34 | 5.14 | 2.84 | 2.74 | 2.49 | 2.39 | 0.69 | 0.39 | 0.20 | 0.80 | 0.10 | 0.05 | 0.20 | 2.30 | 0.10 | 0.25 | 0.10 | 1.70 | 0.30 | 0.19 |
| | $\Delta z$ (m) | 25.0 | 38.9 | 11.9 | 42.1 | -22.9 | 32.3 | -29.9 | 40.8 | -3.0 | -14.7 | 34.1 | -7.6 | -5.0 | 20.0 | -15.0 | 2.0 | -12.0 | 14.0 | -6.0 | 5.0 | -1.0 | -13.0 | 9.0 |

| D solutions | | | | | | | | | | | | | | | | | | | | | | | | |
|---|---|---|---|---|---|---|---|---|---|---|---|---|---|---|---|---|---|---|---|---|---|---|---|---|
| | BT | 4 | 3 | 3 | 4 | 3 | 3 | 3 | 2-½ | 2-½ | 2 | 1-¼ | 1 | 1-¼ | ½ | ½ | ½ | 2 | ½ | 1 | ½ | 1-½ | 1-¼ | ½ |
| | BT-NoUp | 4 | 4 | 4 | 4 | 3 | 3 | 2-½ | 2-½ | 2-½ | 2 | 1-¼ | 1 | 1-¼ | ½ | ½ | ½ | 1-½ | ½ | 1 | ½ | 1-½ | 1-½ | ½ |
| | CB-Theor | 3.530 | 3.531 | 3.333 | 3.307 | 3.270 | 2.727 | 2.698 | 2.579 | 2.548 | 1.862 | 1.227 | 1.011 | 1.283 | 0.325 | 0.508 | 0.404 | 1.678 | 0.343 | 0.963 | 0.281 | 1.405 | 1.401 | 0.488 |
| | CB-Disc | 4 | 4 | 4 | 4 | 4 | 3 | 3 | 3 | 3 | 2 | 1-¼ | 1 | 1-¼ | ½ | ½ | ½ | 2 | ½ | 1 | ½ | 1-½ | 1-½ | ½ |

| h (m) | Node | 1 | 2 | 3 | 4 | 5 | 6 | 7 | 8 | 9 | 10 | 11 | 12 | 13 | 14 | 15 | 16 | 17 | 18 | 19 | 20 | 21 | 22 | 23 | 24 |
|---|---|---|---|---|---|---|---|---|---|---|---|---|---|---|---|---|---|---|---|---|---|---|---|---|---|
| | BT | 0 | 21.1 | 55.4 | 51.5 | 91.3 | 51.7 | 82.5 | 43.9 | 69.8 | 41.4 | 22.1 | 40.0 | 21.7 | 12.1 | 72.2 | 24.3 | 82.2 | 26.1 | 90.8 | 20.1 | 73.3 | 21.9 | 7.81 | 39.7 |
| | BT-NoUp | 0 | 21.1 | 58.8 | 66.4 | 106 | 66.6 | 97.4 | 42.8 | 68.8 | 40.4 | 21.0 | 39.0 | 20.7 | 12.1 | 75.6 | 39.2 | 97.1 | 9.5 | 106 | 19.0 | 72.2 | 20.9 | 7.43 | 38.7 |
| | CB-Theor | 0 | 17.8 | 54.3 | 55.6 | 91.9 | 56.7 | 86.3 | 40.3 | 69.0 | 44.1 | 21.9 | 27.9 | 7.64 | 6.99 | 8.02 | 7.70 | 7.98 | 7.72 | 7.79 | 7.70 | 8.18 | 7.68 | 7.69 | 7.71 |
| | CB-Disc | 0 | 21.1 | 58.8 | 66.4 | 106 | 78.8 | 110 | 70.9 | 106 | 94.4 | 75.1 | 93.0 | 74.7 | 12.1 | 75.6 | 39.2 | 97.1 | 53.1 | 118 | 47.1 | 110 | 74.9 | 61.4 | 92.7 |

**Table 3(d): Case #4 San Miguel network properties, diameter (D) results (inch nominal sizes, with CB-Theor in inches), and nodal h.**

| | Link | 1-2 | 2-3 | 3-4 | 4-5 | 5-6 | 6-7 | 7-8 | 8-9 | 9-10 | 10-11 |
|---|---|---|---|---|---|---|---|---|---|---|---|
| **Network** | Length (m) | 189 | 168 | 139 | 81 | 32 | 92 | 225 | 115 | 52.3 | 85 |
| | Q (L/s) | 3.60 | 3.60 | 3.60 | 3.60 | 3.60 | 3.60 | 3.60 | 3.60 | 3.60 | 3.60 |
| | $\Delta z$ (m) | 27.4 | 10.7 | -6.4 | 6.1 | -5.2 | -18.6 | 33.2 | 58.2 | -11.3 | 32.9 |
| | | | | | | | | | | | |
| **D solutions** | BT | 3 | 3 | 3 | 3 | 3 | 2-½ | 2 | 1-¼ | 1-¼ | 1-¼ |
| | BT-NoUp | 3 | 3 | 3 | 3 | 3 | 2-½ | 2 | 1-¼ | 1-¼ | 1-¼ |
| | CB-Theor | 2.939 | 2.929 | 2.929 | 2.929 | 2.929 | 2.929 | 1.671 | 1.462 | 1.462 | 1.368 |
| | CB-Disc | 3 | 3 | 3 | 3 | 3 | 3 | 2 | 1-½ | 1-½ | 1-¼ |

| | Node | 1 | 2 | 3 | 4 | 5 | 6 | 7 | 8 | 9 | 10 | 11 |
|---|---|---|---|---|---|---|---|---|---|---|---|---|
| **h (m)** | BT | 0 | 25.88 | 35.20 | 27.68 | 33.13 | 27.70 | 7.02 | 28.27 | 43.86 | 13.19 | 14.60 |
| | BT-NoUp | 0 | 25.88 | 35.20 | 27.68 | 33.13 | 27.70 | 7.02 | 28.27 | 43.86 | 13.19 | 14.60 |
| | CB-Theor | 0 | 25.53 | 34.51 | 26.72 | 32.01 | 26.51 | 7.00 | 6.99 | 32.93 | 6.96 | 7.02 |
| | CB-Disc | 0 | 25.88 | 35.20 | 27.68 | 33.13 | 27.70 | 8.37 | 29.62 | 67.54 | 47.02 | 48.43 |

**Table 3(e): Case #5 El Guabo network properties, diameter (D) results (inch nominal sizes, with CB-Theor in inches), and nodal h.**

| | Link | 1-2 | 2-3 | 3-4 | 4-5 | 5-6 | 6-7 | 7-8 | 8-9 | 9-10 | 2-11 | 3-12 | 4-13 | 5-14 | 6-15 | 7-16 | 8-17 | 9-18 | |
|---|---|---|---|---|---|---|---|---|---|---|---|---|---|---|---|---|---|---|---|
| Network | Length (m) | 383 | 486 | 1030 | 600 | 150 | 400 | 187 | 450 | 227 | 230 | 240 | 110 | 270 | 130 | 130 | 260 | 110 | |
| | Q (L/s) | 17.72 | 14.68 | 12.76 | 11.96 | 10.04 | 7.72 | 6.60 | 3.12 | 1.20 | 3.04 | 1.92 | 0.80 | 1.92 | 2.32 | 1.12 | 3.48 | 1.92 | |
| | $\Delta z$ (m) | 10.9 | 10.0 | -5.6 | 3.2 | -2.6 | 5.7 | -4.1 | 4.2 | -3.1 | 2.0 | 2.5 | -1.2 | 2.0 | -1.1 | 0.0 | 1.0 | 2.0 | |
| D solutions | BT | 8 | 6 | 6 | 6 | 6 | 5 | 5 | 4 | 2 | 2-½ | 1-½ | 1-½ | 2 | 3 | 1-¼ | 3 | 1-½ | |
| | BT-NoUp | 8 | 6 | 6 | 6 | 6 | 5 | 5 | 4 | 2 | 2-½ | 1-½ | 1-½ | 2 | 3 | 1-¼ | 3 | 1-½ | |
| | CB-Theor | 6.875 | 6.408 | 6.144 | 6.008 | 5.691 | 4.800 | 4.576 | 3.494 | 2.649 | 2.364 | 1.608 | 1.529 | 1.932 | 3.250 | 1.395 | 3.076 | 1.647 | |
| | CB-Disc | 8 | 8 | 8 | 6 | 6 | 5 | 5 | 4 | 3 | 2-½ | 1-½ | 1-½ | 2 | 4 | 1-½ | 4 | 2 | |
| | Node | 1 | 2 | 3 | 4 | 5 | 6 | 7 | 8 | 9 | 10 | 11 | 12 | 13 | 14 | 15 | 16 | 17 | 18 |
| h (m) | BT | 0 | 10.34 | 18.50 | 9.91 | 11.53 | 8.61 | 13.16 | 8.65 | 12.11 | 7.25 | 8.48 | 7.23 | 7.35 | 8.84 | 7.03 | 7.16 | 7.68 | 7.80 |
| | BT-NoUp | 0 | 10.34 | 18.50 | 9.91 | 11.53 | 8.61 | 13.16 | 8.65 | 12.11 | 7.25 | 8.48 | 7.23 | 7.35 | 8.84 | 7.03 | 7.16 | 7.68 | 7.80 |
| | CB-Theor | 0 | 9.76 | 18.35 | 9.93 | 11.49 | 8.46 | 12.70 | 7.94 | 10.67 | 7.00 | 7.00 | 7.00 | 7.00 | 7.01 | 7.00 | 7.00 | 7.00 | 7.01 |
| | CB-Disc | 0 | 10.34 | 19.85 | 13.47 | 15.09 | 12.17 | 16.72 | 12.21 | 15.67 | 12.27 | 8.48 | 8.58 | 10.91 | 12.40 | 10.94 | 13.84 | 12.67 | 15.76 |

[Figure]

**Figure 1: Element schematic of a GDWN.**

[Figure]

**Figure 2:  Three-pipe branch network.**

[Figure]

**Figure 3:  PVC pipe cost from 2011 data.**

[Figure]

**Figure 4. Bhave (1978) index notation at an internal node, *j*.**

[Figure]

**Figure 5: Network elevation (z) and hydraulic grade lines (HGLs) of algorithm solution for main distribution links.**

[Figure]

**Figure 6: Diameter sizes of calculus-based (CB-Disc) solutions above the global optimum solutions.**

---

## Author Comment (AC2) · 28 May 2017

1. The manuscript presents algorithms for optimization of branching gravity-driven water networks, which is interesting. The subject addressed is within the scope of the journal.

2. However, the manuscript, in its present form, contains several weaknesses. Appropriate revisions to the following points should be undertaken in order to justify recommendation for publication.

2. Thank you for reviewing our paper and providing detailed feedback. Please see our responses to each of your comments below.

3. Full names should be shown for all abbreviations in their first occurrence in texts. For example, PVC in p.1, D in p.3, WDN in p.4, HGL in p.12, etc.

3. We have updated our paper to include full names next to the first instances of all abbreviations in the text.

4. For readers to quickly catch your contribution, it would be better to highlight major difficulties and challenges, and your original achievements to overcome them, in a clearer way in abstract and introduction.

4. In response to your feedback, we have substantially revised the abstract and introduction sections to make the contribution and scope of the paper more clear. Please refer to those sections.

5. It is mentioned in p.1 that three cost-minimization algorithms are adopted for the design of moderate-scale branching Gravity-Driven Water Networks. What are the other feasible alternatives? What are the advantages of adopting these particular algorithms over others in this case? How will this affect the results? More details should be furnished.

**Author's Response:**

5. The problem considered in this paper is that of selecting a single pipe diameter for each link in a water network to optimize the material cost. This problem has three major categories of methods that are applicable to it: enumeration methods (including both complete enumeration and partial enumeration), nonlinear programming methods, and metaheuristic methods. For each of these categories we have proposed and tested one representative algorithm: backtracking (partial enumeration), the Jones calculus-based algorithm (nonlinear programming), and a genetic algorithm (metaheuristic). While there are many other types of metaheuristic algorithms (simulated annealing, Tabu search, cellular automata, ant colony optimization, and particle swarm optimization), the genetic algorithm is the most representative of these and is also the most commonly used (Zhao et al. 2016).

Note that we have categorized the backtracking algorithm as a partial enumeration method and not a heuristic algorithm. The backtracking algorithm, like heuristic methods, follows a set of deterministic rules to find better solutions, however, those rules are strictly formulated to find cost-optimal solutions and does so without missing the global optimum. In contrast, heuristic algorithms follow rules which

achieve some proxy of an optimum solution but do not guarantee the global optimum. One example of a heuristic algorithm comes from Suribabu (2012), whose algorithm uses the uniformity of a solution's flow velocity as a proxy for its cost-optimality. As such, the Suribabu algorithm increments a pipe diameter when its flow velocity is high and decrements the diameter when its flow velocity is low, in an attempt to approach more optimal solutions. It was not necessary to include such a heuristic algorithm for comparison in this paper, since the Backtracking algorithm presented follows a more strictly formulated set of rules that do guarantee a cost optimum.

We note that other methods that can be applied to water network design do not address the problem of focus in our paper. For example, linear programming methods provide split-pipe solutions for each link, while the problem addressed here calls for each link to have a single-diameter solution. Multi-objective optimization methods, which typically involve new implementations of metaheuristic methods, are also outside the scope of (single-objective) cost optimization, although it should be noted that cost-optimization algorithms, such as the ones used in this paper, can be used within multi-objective implementations. In addition, decomposition methods, where networks are broken down into smaller sub-networks, can use any of the methods listed above, and are therefore not an exclusive method category. Given the appropriateness of the presented algorithms to the network sizes of gravity water networks, decomposition was not necessary.

We have added a more thorough description to our introduction section that gives clearer context to the selection of these algorithms, highlighting the category of method to which they belong and the key features that make them distinct from one another.

- Suribabu, C. R.: Heuristic-based pipe dimensioning model for water distribution networks, J. Pipeline Syst. Eng. Pract., 3(4), 115–124, doi:10.1061/(ASCE)PS.1949-1204.0000104, 2012.
- Zhao, W., Beach, T., and Rezgui, Y.: Optimization of Potable Water Distribution and Wastewater
   Collection Networks: A Systematic Review and Future Research Directions, IEEE Transactions on
   Systems, Man, and Cybernetics: Systems, 46 (5), 659-681, doi:10.1109/TSMC.2015.2461188, 2016.

6. It is mentioned in p.4 that five actual GDWNs installed in Panama, Nicaragua, and the Philippines are adopted as test cases. What are the other feasible alternatives? What are the advantages of adopting these particular test cases over others in this case? How will this affect the results? More details should be furnished.

**Author's Response:**

6. This work was prompted by our GDWN design work in Panama, Nicaragua, and the Philippines. Thus, these networks are most pertinent to our interests and are networks for which we have detailed data. Despite an extensive review of the literature, we did not uncover any common test cases for GDWNs, nor many papers focusing on GDWN design, which is likely due to their use in mostly developing countries. While each network is a branching type without loops, the total lengths range to over 15 km. This is considered a very large network in the scale of the present work. In fact, this is the longest GDWN in Central and South America. Two serial networks were tested to demonstrate the effect of a local high point on the algorithm solutions.

7. It is mentioned in p.5 that "... The pressure upper bound is not incorporated into the optimization process...." Why this is not incorporated? More justifications should be furnished on this.

Author's Response:

7. It is correct that the pressure upper bound is not incorporated into the optimization process. Worstcase pressure conditions occur under hydrostatic conditions, which are directly related to the maximum elevation change in the network and where no flow occurs. Therefore, **before the optimization process is undertaken**, the selections of appropriate pressure ratings for the pipe and, if needed, break-pressure tanks are left to the correct judgment of the designer under **no-flow conditions**. In addition, precautions against water hammer are left to the designer since the design process presented in the paper specifically addresses network cost minimization and not this aspect of the hydraulic design.

8. It is mentioned in p.5 that "... precautions against water hammer are left to the designer..." Why this is left to the designer? More justifications should be furnished on this.

Author's Response:

8. See response to 7.

9. It is mentioned in p.7 that a two-part model is adopted in this study. What are the other feasible alternatives? What are the advantages of adopting this particular model over others in this case? How will this affect the results? More details should be furnished.

Author's Response:

9. Both single-part and two-part pipe cost models are considered in the paper, which result in two different equations for network cost minimization, Equations (15) and (18) respectively. The single-part cost model is simpler than the two-part model and results in the simpler Equation (15) compared with Equation (18). However, as Figure 3 in the paper shows, actual pipe-cost data are more accurately fit to the two-part cost model. There appears to not be the need for more elaborate pipe cost models, nor has one been proposed in the literature as far as the authors are aware.

10. It is mentioned in p.9 that a cubic spline is adopted to complete the transition between small and large pipe sizes. What are the other feasible alternatives? What are the advantages of adopting this particular function over others in this case? How will this affect the results? More details should be furnished.

Author's Response:

10. As seen in Equation (21) in the paper, a first derivative of the pipe cost w.r.t. pipe diameter is needed in the network cost minimization Equation (18), which is based on the two-part pipe cost model. Thus, a smooth, continuous function is fit between the small-*D* and large-*D* parts of the cost model to ensure the first derivative is well behaved. The cubic polynomial is the simplest one that ensures continuous function and first-derivative behavior between the small-*D* and large-*D* parts of the cost model.

11. It is mentioned in p.13 that two rejection criteria are adopted to reduce the number of solutions to be evaluated. What are the other feasible alternatives? What are the advantages of adopting these particular criteria over others in this case? How will this affect the results? More details should be furnished.

**Author's Response:**

11. In the BT algorithm, the selection of a diameter size results in a corresponding pressure head at a downstream node and a corresponding cost, both of which serve as the basis of the two rejection criteria used. Because the direct relationships between diameter and cost, and between diameter and pressure head, span the entire range of diameters, criteria to prune certain diameter choices can be safely proposed that never skip a global optimum. It can be noticed that the two criteria prune diameter solutions at the competing ends of the range of diameters, with the cost criteria pruning out larger diameters, which add more cost, and the head criteria pruning out smaller diameters, which contribute to greater head loss. These two criteria for enumeration methods have been the only ones in the literature that do not risk pruning a global optimum (Raad 2011, González-Cebollada et al. 2011). Other criteria, such as rejecting partial candidates that include a diameter that is larger than an upstream diameter, risk skipping the global optimum and were therefore not included in the BT algorithm. Instead, these were instead included in the modified algorithm, BT-NoUp, to compare the resulting effect on computation time and the cost of the solution.

These rejection criteria are further strengthened with the use of Pre-processor 1 (section 4.1) to adjust the maximum available diameter and Pre-processor 2 (section 4.2) to adjust the minimum static pressure head at each node (section 4.2), again without risk of pruning the global optimum and without the need to update any such calculations during the search process itself. To our knowledge, is the first implementation of Pre-processor 1 in enumeration methods and the first implementation of Pre-processor 2 in any method.

12. It is mentioned in p.14 that the method presented by González-Cebollada et al. (2011) is adopted as the BT algorithm. What are the other feasible alternatives? What are the advantages of adopting this particular method over others in this case? How will this affect the results? More details should be furnished.

**Author's Response:**

12. The use of backtracking for water network design has not been widely used, and only two other backtracking methods can be found in the literature, those by Gessler (1985) and González-Cebollada et al. (2011). However, the algorithm proposed by Gessler, unlike the present work's BT algorithm, includes a pipe-grouping criteria set that risks pruning the global optimum, and is therefore a less appropriate comparator to the present study's BT algorithm. The González-Cebollada algorithm, on the other hand, does not include such a criterion to potentially prune a global optimum from consideration, however, it halts its search after finding the first feasible solution. Thus, out of the two reported backtracking algorithms in the literature, both do not guarantee a global optimum, while the BT algorithm presented in this work does. The trade-off of this expanded search is an increase in computation time, however, the BT algorithm was able to execute in a satisfactory timeframe for all cases (with a maximum runtime of 79 seconds). More clarifications have been added to the paper to further clarify BT's comparison to the González-Cebollada backtracking algorithm.

Gessler J: Pipe network optimisation by enumeration, Proceedings of the Specialty Conference on Computer Applications in Water Resources, American Society of Civil Engineers, New York, 1985.

González-Cebollada, C., Macarulla, B., and Sallán, D.: Recursive Design of Pressurized Branched Irrigation Networks, J. Irrig. Drain Eng., 137(6), 375-382, doi:10.1061/(ASCE)IR.1943-4774.0000308, 2011.

13. It is mentioned in p.15 that a single-point crossover is adopted in this study. What are the other feasible alternatives? What are the advantages of adopting this particular crossover method over others in this case? How will this affect the results? More details should be furnished.

**Author's Response:**

13. The crossover operation in a genetic algorithm has a number of variations including: single-point, two-point, uniform, multi-parent single point, multi-parent two-point, multi-parent uniform, universal single-point, universal two-point, and universal parent uniform. Unlike a single-point crossover, in which two individuals exchange data to the right (or left) of a randomly selected point in their chromosome, a two-point crossover includes two such random points that distinguish the points at which data is exchanged. The uniform crossover is where each link's diameter has an equally likely probability of being exchanged with the other parent. The references to multi-parent techniques involve schemes where more than two parents are used to exchange information. Finally, references to universal parents means that, at low probabilities, one of the parent chromosomes is the set of all commercial diameters. This serves to add in more variation to the population, but effectively accomplishes the same thing as the mutation operator. There is no standard technique used in the literature, though single-point crossover has a long history of use in genetic algorithms for water distribution network design (Simpson et al. 1994, Krapivka and Ostfeld 2009, Dandy et al. 1996). While we did experiment with a two-point crossover design, we did not find the results to have any benefit over a single-point design, and decided on the single-point technique for its greater simplicity.

- Dandy, G. C., Simpson, A. R., and Murphy, L. J.: An improved genetic algorithm for pipe network optimization, Water Resour. Res., 32(2), 449–458, doi:10.1029/95WR02917, 1996.
- Krapivka, A., and Ostfeld, A.: Coupled genetic algorithm—Linear programming scheme for least cost design of water distribution systems, J. Water Resour. Plann. Manage., 135(4), 298–302, doi:10.1061/(ASCE)0733-9496(2009)135:4(298), 2009.
- Simpson, A. R., G. C. Dandy, and Murphy, L. J.: Genetic Algorithms Compared to Other Techniques for Pipe Optimization, J. Water Resour. Plann. Manage., 120(4), 423–443, doi:10.1061/(ASCE)0733-9496(1994)120:4(423), 1994.

14. It is mentioned in p.15 that a pinwheel lottery (with replacement) is adopted to select candidates to be carried into the next generation. What are the other feasible alternatives? What are the advantages of adopting this particular approach over others in this case? How will this affect the results? More details should be furnished.

**Author's Response:**

14. The technique used to select candidate solutions is more commonly known as a proportionate selection method, and the paper has been updated to include this term. With this technique, the fitness of each solution directly relates to its probability of being selected for the next generation. The proportionate selection method has a long history of use in genetic algorithms for water distribution network design (Goldberg 1989, Simpson et al. 1994). One feature of this selection process is that lessfit candidates have ability to be selected into successive rounds, albeit at a lower probability than fitter candidates. This ability to preserve less-fit candidates in the population increases genetic diversity, permitting genetic algorithms to avoid convergence into a local optimum. Alternates to proportionate selection include tournament selection, elitism, and stochastic universal sampling, which have been shown improved performance in genetic algorithms in certain cases of water network problems. Tournament selection pairs off small groups of candidates and selects the fittest candidate for subsequent crossover and has the added benefit of being less prone to genetic drift. Stochastic universal sampling has the same effect as proportionate selection, although it does so by generating only a single random number, thus saving computational time. Because the genetic algorithm runtimes were acceptable, within a few seconds each, there was no need to use more efficient selection techniques. Based on the literature, the proportionate selection technique is appropriate for use for this problem formulation.

Simpson, A. R., G. C. Dandy, and Murphy, L. J.: Genetic Algorithms Compared to Other Techniques for Pipe Optimization, J. Water Resour. Plann. Manage., 120(4), 423–443, doi:10.1061/(ASCE)0733-9496(1994)120:4(423), 1994.

Goldberg D.: Genetic algorithms in search, optimisation, and machine learning, Addison Wesley, San Francisco, 1989.

15. It is mentioned in p.18 that "...and several of these would likely improve upon the GA results obtained in this study ...." Why they are not performed in this study? More justifications should be furnished on this.

**Author's Response:**

15. The purpose of the present study is to offer a comparison of the major methods available for gravitydriven water network optimization in order to help inform designers of GDWNs on what to expect from each algorithm, where one of those is a genetic algorithm (GA). The computational research being done with genetic algorithms, and more broadly metaheuristic methods, is vast and presently developing. The number of variations available to implement in a genetic algorithm are too large to properly list in the present work. A good review of these developments in the context of water resources is given by Nicklow et al. (2010). Notably, while a theoretical basis for why genetic algorithms work has been proposed, the performance of a GA variant cannot be predicted on a given problem set, thus requiring experimental characterization (i.e., test runs on a range of cases). Even so, despite certain techniques showing benefits on water network test cases, it is not possible to generalize these findings to all water network cases.

The GA included in this paper is meant to present a benchmark for the results that could be expected when designing a GA with a reasonable amount of effort put into finding optimal parameters and the selection of proper operator schemes (selection, mutation, and crossover). It was not our intention with this paper to find the optimal GA scheme to implement for GDWNs. In fact, one improvement mentioned in the paper (scaling of the fitness function (Dandy et al. 1996)) was attempted in the present work. However, upon the trial of a few scaling schemes (Eg. a fitness exponent transitioning from 1 on the first generation to 2 on the last generation), no benefit was observed. Nonetheless, it is possible that a more thorough search of different scaling exponents may have improved the results, thus, this technique was not given a full consideration. Another improvement

mentioned, the systematic optimization of parameters (Reed et al. 2000), requires quite an extensive process to find optimal search parameters. A less intensive and systematic version of this was performed, as mentioned in the parameter tuning process, although it is possible that a more systematic and thorough search would have improved the GA parameters selected.

Seen in a practical context, such advanced GA techniques are not in line with GDWN design practice. A GDWN designer typically would rarely would have access to commercial water network design software, and would typically not have much time to dedicate to optimization. Instead, designers will often select pipe diameters manually by marching along a distribution main and selecting successive pipe diameters, each time checking for hydraulic acceptability. With this as common practice, the use of any feasible optimization technique represents a substantial improvement. In our experience, and as noted above, a designer rarely has access to commercial water network design software, nor would they have the capability to implement more advanced techniques in the literature. As such, the GA represented in the paper represents a feasible performance benchmark with which to compare with the other algorithms, which are already easier to implement even without more advanced GA techniques.

- Dandy, G. C., Simpson, A. R., and Murphy, L. J.: An improved genetic algorithm for pipe network optimization, Water Resour. Res., 32(2), 449–458, doi:10.1029/95WR02917, 1996.
- Nicklow, J., Reed, P., Savic, D., Dessalegne, T., Harrell, L., Chan-Hilton, A., Karamouz, M., Minsker, B., Ostfeld, A., Singh, A., Zechman, E., and ASCE Task Committee on Evolutionary Computation in Environmental and Water Resources Engineering: State of the Art for Genetic Algorithms and Beyond in Water Resources Planning and Management, J. Water Resour. Plann. Manage., 136(4), 412-432, doi:10.1061/(ASCE)WR.1943-5452.0000053, 2010.
- Reed, P., Minsker, B., and Goldberg, D. E.: Designing a competent simple genetic algorithm for search and optimization, Water Resour. Res., 36(12), 3757–3761, doi:10.1029/2000WR900231, 2000.

16. Some key parameters are not mentioned. The rationale on the choice of the particular set of parameters should be explained with more details. Have the authors experimented with other sets of values? What are the sensitivities of these parameters on the results?

Author's Response:

16. All parameters chosen for the GA algorithm are listed in Section 4.5. These include  $p_{mut} = 0.02$ ,  $N_c = 50$ ,  $p_{xover} = 0.5$ ,  $N_{gen} = 100$ ,  $a_{hyd,norm} = 0.1$ . The first four of these parameters were originally chosen based on typical values presented in the literature and then tuned with a parameter sensitivity study for the first test case. Simspon et al. (1994) present typical values for  $N_c$  (30 - 200),  $p_{xover}$  (0.7 - 1.0),  $p_m$  (0.01 - 0.05), and  $N_{gen}$  (100 - 1000). During the parameter tuning process, each parameter was sequentially varied until a decrease or increase in no parameter resulted in significantly lower-cost results, as determined by an average of 100 runs of each. The last parameter ( $a_{hyd,norm}$ ) was found through a parameter sensitivity study and then checked to be reasonably in line with literature-based hydraulic penalty coefficients, which normalized based on their respective network's optimum price. A value for the normalized hydraulic penalty coefficient,  $a_{hyd,norm}$ , was chosen such that the GA converged on solutions that tended to satisfy the minimum static pressure constraint, but still allowed the population to gravitate towards smaller diameters with static pressures close to  $h_{min}$ . The least sensitive parameters were  $p_{xover}$ ,  $N_c$ , and  $N_{gen} = 100$ , such that even a 50% change in their values

resulted in less than 1-5% change in average solution cost, while  $a_{hyd,norm}$  was moderately sensitive to changes and  $p_{mut}$  was the most sensitive, thus requiring finer-tuning in increments of 0.005, where the best performance was achieved at  $p_{mut} = 0.020$ . As further validation of the value chosen for  $p_{mut}$ , a generally acceptable standard for GA performance occurs when  $p_{mut} \approx 1/Nc$  (Reed 2000).

Reed, P., Minsker, B., and Goldberg, D. E.: Designing a competent simple genetic algorithm for search and optimization, Water Resour. Res., 36(12), 3757–3761, doi:10.1029/2000WR900231, 2000.

17. Some assumptions are stated in various sections. Justifications should be provided on these assumptions. Evaluation on how they will affect the results should be made.

**Author's Response:**

17. The main assumptions used in the problem formulation will be discussed below: turbulent flow, smooth pipe, and minimum static pressure head. The assumption of turbulent flow is reasonable for all practical potable water distribution networks, and is a traditionally-held assumption in the literature for this scale of water networks. The validity of this assumption is also verifiable post-analysis, as is mentioned in page 6 of our paper. It would not be appropriate to speculate on how the results would be different if the flow were laminar or transitional, since such cases do not normally occur in practice. The assumption of smooth pipe is consistent with conditions that exist in the field for nearly all the GDWNs we encounter in our work, which are nearly always constructed out of PVC pipe. A deeper discussion of this can be found in Jones (2011) pp. 39-44. Finally, there is no current standard for assumptions of minimum static pressure head in gravity water network design. Some examples from gravity water network design references include 5 m (Bouman, 2014), 10 m (Arnalich, 2010), 5-15 m (Action Contre La FAIM, 2008), and 8-20 m (Swamee and Sharma 2008). Thus, 7 m is a reasonable assumption and in line with our own experience of common practice. Including even larger values for hmin will unnecessarily increase pipe sizes and thus cost. This justification has been included in the paper.

Arnalich, S.: How to design a Gravity Flow Water System. Arnalich – Water and Habitat, 2010.

Bouman, D.: Hydraulic design for gravity based water schemes. Aqua for All, Den Haag, 2014.

Action Contre La FAIM: Design, Sizing, Construction and Maintenance of Gravity-Fed Systems in Rural Areas: Module 2, 2008.

Swamee, P. K., and Sharma, A. K.: Design of Water Supply Pipe Networks, p. 99, Wiley, Hoboken, NJ, 2008.

Jones, G. F.: Gravity-driven Water Flow in Networks: Theory and Design, pp. 39-44. Wiley, Hoboken, NJ, 2011.

18. The discussion section in the present form is relatively weak and should be strengthened with more details and justifications.

**Author's Response:**

18. We have added a number of additional elements to the discussion in order to strengthen it, per your suggestion. A discussion of the CB results in the context of the mapping schemes now appears in Section6. Even though our tabular results are based on up-sizing, each of the methods described in Section 6

produce a solution closer to the global minimum of the more-computationally intensive and not-soscalable BT method. These methods engage the designer to make decisions on pipe-ordering (e.g., large diameters are chosen upstream of small diameters for the compound-pipe approach) and the acceptability or unacceptability of a diameter change relative to the HGL corresponding to the CB continuous diameter results. In addition, discussion of the rationale behind GA techniques used and parameter tuning has been added, including a clearer explanation of the relevance of these criteria may have on the results of the study. Where appropriate, we have included more details in sections which were a better fit for their inclusion, for example, the comparison of our BT algorithm to the González-Cebollada algorithm (section 4.3).

19. Moreover, the manuscript could be substantially improved by relying and citing more on recent literatures about real-life case studies of contemporary soft computing techniques in water resources engineering such as the followings:

- Gholami, V., et al., "Modeling of groundwater level fluctuations using dendrochronology in alluvial aquifers", Journal of Hydrology 529 (3): 1060-1069 2015.
- Taormina, R., et al., "Data-driven input variable selection for rainfall-runoff modeling using binary-coded particle swarm optimization and Extreme Learning Machines", Journal of Hydrology 529 (3): 1617-1632 2015.
- Wu, C.L., et al., "Methods to improve neural network performance in daily flows prediction," Journal of Hydrology 372 (1-4): 80-93 2009.
- Wang, W.C., et al., "Improving forecasting accuracy of annual runoff time series using ARIMA based on EEMD decomposition," Water Resources Management 29 (8): 2655-2675 2015.
- Chen, X.Y., et al., "A Novel Hybrid Neural Network based on Continuity Equation and Fuzzy Pattern-recognition for Downstream Daily River Discharge Forecasting," Journal of Hydroinformatics 17 (5): 733-744 2015.
- Saeidifarzad, B., et al., "Multi-site calibration of linear reservoir based geomorphologic rainfallrunoff models", Water 6 (9): 2690-2716 2014.

19. We have looked at the literature examples you have provided and reviewed their scope in comparison with that of the current paper. Our paper focuses on water distribution network design, a significant body of work in its own right without including other aspects of the much broader field of water resources. We felt that the inclusion of references on aquifer levels and flows prediction for watersheds and runoff did not fit within the scope of our paper. We have included a reference to Taorimina (2015), however, in an attempt to highlight the many uses of metaheuristic methods in water resources engineering.

**20. In the conclusion section, the limitations of this study, suggested improvements of this work and future directions should be highlighted.**

**Author's Response:**

20. The conclusions section has been has been updated to highlight the limitations of the study (e.g., the scalability of the BT algorithm, the lack of inclusion of some GA improvements reported in the literature) and improvements and future directions for the work (e.g., integrating the strengths of the BT to map the CB algorithm continuous diameter solution to a discrete diameter solution and verifying the suitability of the BT and BT-NoUp algorithms on other large GDWNs).

Thank you again for your time spent reviewing our paper.

[revised manuscript text omitted]
. The equation has been extended in the present work to include minor losses and rough pipe. When solved simultaneously with the energy equation for each link, a unique solution for all link diameters and nodal pressure head values are obtained that produces minimum network cost, as opposed to the distribution main cost as in Swamee and Sharma (2000).

The method of Jones (2011) also applies to serial and loop networks because of its generality.

Heuristic methods follow specific rules to incrementally build better solutions, although the rules are not strictly formulated to trend towards local or global optima. An approach by Monbaliu et al. (1990) sets all network pipes to their minimum size, where the pipe that has a maximum head loss gradient is incremented to its next-highest size until all nodal

- 25 head requirements are satisfied. Similarly, an algorithm by Keedwell and Khu (2006) selects an initial solution and iteratively responds to nodal head deficits and surpluses by incrementing or decrementing pipe sizes accordingly until a feasible solution is found. Suribabu (2012) proposed a heuristic that identifies pipes to increment or decrement in size based on flow velocity and alternative metrics such as proximity to the source node, achieving acceptable cost results with computational efficiency. While these algorithms are typically computationally efficient, they do not guarantee a global optimum.
- 30

20

Metaheuristic optimization methods allow for a set of solutions to evolve through random processes that are guided with an objective function which rewards low network costs and penalizes hydraulic insufficiencies. Examples include evolutionary algorithms, which are most commonly genetic algorithms (Krapivka and Ostfeld 2009, Simpson et al. 1994, Kadu et al. 2008, Prasad and Park 2004), simulated annealing (Vasan and Simonovic 2010; Tospornsampan et al. 2007), ant colony optimization (Maier et al. 2003), and differential evolution (Vasan and Simonovic 2010). As reviewed by Nicklow et al. (2010), evolutionary algorithms are an emerging popular alternative to the deterministic methods, and they offer the opportunity to accommodate unique constraints and multiple design objectives. The main challenges for evolutionary algorithms are the difficulty of incorporating constraints into objective functions, the optimum selection of parameters, and a relatively large

- 5 amount of computational effort. In addition to optimizing for cost, multi-objective methods, often based on evolutionary algorithms, allow the designer to choose from a Pareto-optimal front of objectives, such as cost and reliability (Prasad and Park 2004). In addition to water network design, metaheuristic algorithms have been used for a range of problems in water resources engineering, such as rainfall and runoff modelling (Taormina et al. 2015).
- Decomposition methods involve the partitioning of networks into smaller sub-networks which are each optimized using one of many types of techniques and then combined into an overall solution. In some cases, the loops in the sub-networks are removed, producing branching trees which are then optimized individually. Techniques used to optimize the sub-networks can involve multiple methods, including linear programming (Saldarriaga 2013) and differential evolution (Zheng et al. 2013), with a later stage optimizing the network as a whole using the sub-network solutions as inputs. Note that another distinct use of the term 'decomposition' refers to the approach of iteratively solving "inner" and "outer" mathematical problem formulations, and has been used in the literature by Krapivka and Ostfeld (2009) who traces its use in this context back to

Alperovits and Shamir (1977).

In the present study, we present three algorithms, each from one of three major categories of methods applied to cost optimization of water distribution networks, and compare their performance on five cases adapted from real GDWNs. These algorithms include (1) the calculus-based (CB) optimization model of Jones (2011), an NLP method, (2) backtracking (BT), a

- 20 partial enumeration method, and (3) a genetic algorithm (GA), a metaheuristic method. Major distinguishing features of these algorithms include their working use of continuous diameters (CB) versus discrete diameters (BT and GA), their deterministic nature (CB and BT) versus a stochastic nature (GA), and their relative scalability as better (CB, GA) and worse (BT) for larger networks. In terms of their ability to find a global optimum solution for the problem formulation, CB finds a global optimum for continuous diameters but cannot guarantee a discrete diameter global optimum in its mapped solution, BT can guarantee a
- 25 discrete global optimum, and GA cannot guarantee an optimum. For a direct comparison of techniques, the pipe costs used for all algorithms are found by interpolating a two-part cost formula based on a curve-fit of real cost data for available diameter values. The three algorithms are tested against networks adapted from field data on five actual GDWNs installed in Panama, Nicaragua, and the Philippines.

Within the broader context of water network problem formulations, this paper is concerned with finding cost-optimal 30 single-diameter solutions to branching water distribution networks with steady-state demand flows and pre-specified pipe locations.. By implication of being gravity-driven, the problem does not involve the use of pumping stations. This problem formulation is directly applicable to typical gravity-driven water networks, and is also useful for multi-objective algorithms, the consideration of sub-networks in a decomposition technique, pumped networks, and looped system optimization, which can involve reformulating the problem into a branching configuration.

The results of this study highlight the advantages and weaknesses of each GDWN design method including closeness to the global optimum, the ability to prune the solution space of infeasible and sub-optimal candidates without missing the 5 global optimum, and also computational time. We present two pre-processors with which discrete-diameter search methods can use to reduce the search space without pruning the global optimum. To the authors' knowledge, is the first implementation of Pre-Processor 1 in enumeration methods and the first implementation of Pre-Processor 2 in any method. We also extend the Jones closed-form model to include minor losses, a more-comprehensive two-part cost model, which realistically applies to pipe sizes that span a broad range typical of GDWNs of interest in this work, and for smooth and commercial steel roughness values. 10

**2 **Problem Formulation**

Branching networks are considered (Fig. 1), where all branches connect a distribution main node with a delivery node, shown as tapstands or houses. For each link in a network of  $N_L$  links, pipe length (L) and the net elevation change ( $\Delta z$ ) are considered fixed. Steady-state flow rates (0) are prescribed for each link based on the demand flow data at delivery nodes. As noted above, demand flows are determined by community surveys and extrapolated in time to quantitatively account for

population growth. Minor losses are accounted for through a minor loss coefficient K or a dimensionless equivalent pipe length,  $(L_e/D)$ , or in symbol form,  $L_{ebvd}$ ), where  $L_e$  is the pipe length of diameter D whose frictional loss results in the corresponding minor loss. An optimal solution is obtained by selecting pipe diameters (D) from a set of commercially available diameters such that the network's material cost is minimized. With  $N_D$  choices of diameters for  $N_L$  links, the problem has  $N_D^{N_L}$ candidate solutions.

15

For all nodes, static pressure, h, is greater than or equal to a chosen minimum,  $h_{min}$ . The value for  $h_{min}$  is selected to eliminate possible leakage of contaminated ground water into the network should the operating conditions change in an unanticipated way. The change in static pressure head,  $\Delta h$ , across each link is calculated with the energy equation for pipe flow,

$$\Delta h = -\Delta z + \left(\alpha + K + f\left(\frac{L}{D} + L_{ebyD}\right)\right) \frac{8Q^2}{\pi^2 g D^4} \tag{1}$$

25 where for each link,  $\alpha$  is the kinetic energy correction factor and f is the Darcy friction factor, calculated with the Colebrook-White equation (Colebrook and White 1937) or Churchill correlation (Churchill 1977), and g is acceleration of gravity. The kinetic energy correction factor,  $\alpha$ , is considered only in the first link, where acceleration from a zero-velocity source is sometimes non-negligible for the smallest of GDWNs that have been encountered. Thus,

$$\alpha = \begin{cases} 2 & \text{Re} \le 2300 \\ 1.05 & \text{Re} > 2300 \end{cases}$$

where Re is the Reynolds number for pipe flow,  $4Q' \pi v D$ , and v is the kinematic viscosity of water. The possibility of laminar flow (Re  $\leq 2300$ ) is permitted since branches from the smallest GDWN observed in practice have been in this regime.

The pressure upper bound is not incorporated into the optimization process. Worst-case pressure conditions occur under hydrostatic conditions, which are directly related to the maximum elevation change in the network and where no flow occurs. Therefore, before the optimization process is undertaken, the selections of appropriate pressure ratings for the pipe and, if needed, break-pressure tanks are left to the correct judgment of the designer under no-flow conditions. In addition, precautions against water hammer are left to the designer.

**3 New Calculus-Based Algorithm**

- 10 In this section we develop a new calculus-based algorithm for pipe diameters that minimize overall pipe cost for the network. First appearing in the text by Jones (2011), this algorithm is solved simultaneously with the energy equation for each link to produce unique solutions for *D* and nodal pressure head values that minimize network pipe cost, as opposed to only the distribution main cost as in Swamee and Sharma (2000). The method also applies to serial and loop networks.
- First consider the physical basis for the existence of a unique set of pipe diameters and static pressures for the demand-15 driven design problem with cost minimization included. Several works reviewed in the previous section have considered optimization of GDWN and combined pumped and gravity-driven networks. We assume continuous pipe diameters in this section; values that result from the solution of the energy equation. Mapping between continuous diameters and the discrete nominal sizes, required to complete the design, will not be fully addressed in the present work. However, we will discuss two methods we have used for mapping the continuous *D* solutions onto the discrete pipe diameter set.
- 20 Consider the three-pipe network shown in Fig. 2. Pipes 1-2, 2-3, and 2-4 meet where head  $h_2$  is unknown. Each pipe has prescribed volume flow rate and length and unknown diameter *D* as shown. The change in elevation between the top and bottom of each pipe is  $\Delta z$  and  $\Delta h$  is the change in static pressure head. There is a prescribed head at each outlet for pipes 2-3 and 2-4.
- To facilitate insight, we at first assume turbulent flow, which can be verified post-calculation if necessary, *in smooth pipe* and that minor losses are negligible. Two sources for the friction factor for smooth-turbulent flow are considered, namely the classical Blasius equation (reported in Streeter et al. 1998),  $f = 0.316 \text{ Re}^{-1/4}$ , and the Swamee-Jain correlation (Swamee and Jain 1976),  $f = 0.175 \text{ Re}^{-0.1923}$  (though not explicitly appearing in this reference, f from the Swamee-Jain correlation is obtained by writing it for smooth pipe and comparing this with the energy equation, where f is assumed to be in the form  $a \text{ Re}^n$ ). The Blasius equation has higher accuracy (2% for low Re and 3% for high Re) in the range  $10^4 < \text{Re} < 10^5$ , over which
- 30 most of the GDWNs in this work operate, compared with the Swamee-Jain correlation of +8% / -3%, thus prompting the Blasius equation to be chosen for this work. A combination of the Blasius equation with the energy equation gives explicit

formulas for D for the three links in Fig. 2. For simplicity, and to reduce the number of free parameters, the conditions for pipes 2-3 and 2-4 are assumed to be identical without loss of generality. We obtain

$$D_{12} = 0.741 \left(\frac{\Delta z_{12} + \Delta h_{12}}{L_1}\right)^{-4/19} \left(\frac{Q_{12} \nu^{1/7}}{g^{4/7}}\right)^{7/19}$$

$$D_{23} = D_{24} = 0.741 \left(\frac{\Delta z_{23} + \Delta h_{23}}{L_2}\right)^{-4/19} \left(\frac{Q_{23} \nu^{1/7}}{g^{4/7}}\right)^{7/19}$$
(2)

[revised manuscript text omitted]

Equation (18), and its simpler form Eq. (15), forms the basis for calculus-based optimization in this work and is applied at all internal nodes to uniquely determine *hj*. Equation (18) is valid over the range of ~4000 < Re < ~300,000.</li>
Algorithms to solve a general set of independent, nonlinear algebraic equations using, for example, the Levenberg-Marquardt, Quasi-Newton, Newton-Raphson, or Conjugate Gradient methods are available in most commercial math packages including Matlab (1 Apple Hill Drive, Natick, MA USA 01760) and Mathcad (http://www.ptc.com). We used the package Mathcad in the present work. Thus, compared with an iterative solution procedure, a solution flowchart is not relevant here.

Bhave (1978) first proposed an algorithm like Eq. (15). However, Bhave used an iterative method to solve the design problem. As such, there are several qualifications leading up to the cost minimization equation in Bhave. These include the assumption of smallness in variation of the static pressure head between two iterations. This allowed the terms in the cost function to be approximated as constants. In the present work, the cost-function coefficient and exponent are not assumed constant at any node joining two sets of links; see Equation (18). Nor do we make any assumptions on the orders of magnitude of the terms in our equations to simplify them. For clarity, we re-present Eq. (15) using Bhave's (1978) notation as

$$0 = \sum Q_{ij}^{7b/19} S_{ij}^{-(1+4b/19)} - \sum Q_{jk}^{7b/19} S_{jk}^{-(1+4b/19)}$$

15 where the *ij* and *jk* notation are shown in Fig. 4. Index *j* spans all internal nodes along the distribution main.

[revised manuscript text omitted]

- Two backtracking methods can be found in the literature, namely those by Gessler (1985) and González-Cebollada et al. (2011). The algorithm proposed by Gessler, however, also proposes a pipe-grouping criteria that risks pruning the global optimum and represents a tricky optimization problem in and of itself (Raad 2011). The González-Cebollada algorithm, on the other hand, does not include such criteria to potentially prune a global optimum from consideration, but it halts its search after finding the first feasible solution. In contrast, the BT algorithm in the present study guarantees a global optimum by continuing its search of the solution tree even after the first solution has been found. In addition, the present BT algorithm utilizes Pre-
- 15 Processors 1 and 2 to further reduce the search space, without risk of pruning the global optimum, in advance of its search routine. Thus, out of the two reported backtracking algorithms in the literature, both do not guarantee a global optimum, while the BT algorithm presented in this work does. It should be noted that BT is known to scale poorly with large network sizes and would not be appropriate for use on large urban networks, though its appropriateness is demonstrated here for GDWNs, given that the test cases used in this paper representative of the sizes of GDWNs that would be expected in practice.

[revised manuscript text omitted]
. While a two-point crossover technique was considered, the results were not found to have any benefit over a single-point technique, which was chosen for its greater 20 simplicity. The fitness,  $f_i$ , of each candidate is assessed with penalties associated with the solution's pipe cost,  $C_{pipe,i}$ , and
  - violations of the static pressure head requirement,  $C_{hyd,i}$ , or

$$f_i = \frac{1}{C_{pipe,i} + C_{hyd,i}} \tag{29}$$

The hydraulic cost is obtained for each individual by identifying nodes in which the static pressure head is less than  $h_{min}$  and multiplying the total amount of head violation by a hydraulic penalty coefficient,  $a_{hvd}$ :

$$C_{hyd,i_c} = a_{hyd} \sum_{1}^{N_L} (h_{min} - h_{i_N}) | h_{i_N} < h_{min}$$
(30)

To allow for a hydraulic penalty coefficient to produce similar results in both small-scale (inexpensive) network and a large-scale (more expensive) cases, the hydraulic penalty coefficient is made directly proportional to the average solution cost. With each generation,  $a_{hyd}$  is updated by multiplying the normalized penalty coefficient,  $a_{hyd,norm}$ , by the average pipe cost of the population,

$$a_{hyd} = a_{hyd,norm} \frac{\sum_{i=1}^{N_c} C_{pipe,i_c}}{N_c}$$
(31)

The algorithm then selects candidates to be carried into the next generation through a proportionate selection method, where each candidate has a probability of being selected,  $p_{sel,i}$  in direct proportion to its fitness relative to the sum of all fitness values

5 in the population

10

$$p_{sel,i} = \frac{f_i}{\sum_{1}^{N_c} f_i} \tag{32}$$

The algorithm replaces the parent generation with a generation of equal size and tends to select more fit individuals in successive generations. In this study, the genetic algorithm parameters used were  $p_{mut} = 0.02$ ,  $N_c = 50$ ,  $p_{xover} = 0.5$ ,  $N_{gen} = 100$ ,  $a_{hyd,norm} = 0.1$ . The first four of these parameters were chosen based on typical values presented in the literature and then tuned with a sensitivity analysis for the first test case. Simspon et al. (1994) present typical values for  $N_c$  (30 - 200),  $p_{xover}$  (0.7 - 1.0),  $p_m$  (0.01 - 0.05), and  $N_{gen}$  (100 - 1000). The normalized hydraulic penalty coefficient,  $a_{hyd,norm}$ , was chosen such that the GA converged on solutions which tended to satisfy the minimum static pressure constraint but still

chosen such that the GA converged on solutions which tended to satisfy the minimum static pressure constraint, but still allowed the population to gravitate towards smaller diameters with static pressures close to  $h_{min}$ .

**5 Cases Studied**

- Five cases were studied based on actual GDWN in Panama, Nicaragua, and the Philippines. Global characteristics of each network are presented in Table 1 and the details of each network are presented in Table 3(a)-(e). Each network is a branching type without loops. The total lengths of the networks range from less than 1 km to over 15 km. Two serial networks are tested to demonstrate the effect of a local high point on the algorithm solutions. Elevation plots for each case are shown in Fig. 5.
- The choice of  $h_{min}$  is not standardized, and should appropriately balance the risk of negative pressure in pipes and 20 the increase in network cost due to the requirement of using larger diameters. The choice of  $h_{min}$  in GDWN design is typically 20 in the range of 5 m – 20 m (Arnalich 2010; Bouman 2014; Swamee and Sharma 2008). In the present study  $h_{min} = 7$  m, although 20 this requirement was reduced at selected nodes at the beginning of networks where changes in elevation are still small. At the 20 source node, the static pressure head is fixed at atmospheric pressure. All cases assumed minor-lossless flow, although all 20 algorithms (e.g., Eq. (18) for CB-Theor) are capable of handling minor loss coefficients through the equivalent length method
- as presented above.

**6 Mapping the Theoretical *D* to Discrete Pipe Sizes**

The mapping between continuous diameters and the discrete nominal pipe sizes was accomplished in our solution by one of the following ways:

- For small and moderate size networks, the designer may manually adjust the pipe sizes (downward, normally one pipe size) starting from the first link downstream from the source and continuing along the rest of the distribution main to the end. A nearby plot of the static pressure heads compared with the theoretical *Dij* from our CB approach (on the same Mathcad page) will highlight the acceptability or unacceptability of any change. This exercise also gives the designer an understanding of the sensitivity of the design to small changes in pipe sizes.
  - 2. Based on the theoretical  $D_{ij}$  from the CB approach, a composite pipeline can be created for each link. That is, the lengths for the two discrete pipes sizes that bound the theoretical  $D_{ij}$  from above and below are calculated such that the pressure drop between two consecutive nodes in the distribution main matches between the composite pipeline and the CB approach. This also provides discrete pipe sizes that nearly matches the CB solution in terms of cost.

[revised manuscript text omitted]
. 1996) were attempted in a less systematic way (Ie. a parameter tuning study and three attempts at scaling the fitness function with an increasing exponent), but these did not result in a noticeable on performance. However, it is possible that if these techniques were followed systematically in full, the GA performance may have been improved. Still, the GA algorithm presented has undergone
- 10 reasonable attempts to adapt its design and parameters for real-world GDWN cases, and therefore presents a useful point of comparison to the BT and CB algorithms.

**8 Conclusions**

Algorithms to optimize the cost of branching gravity-driven water networks are evaluated on five test cases from real networks in the Philippines, Nicaragua, and Panama. A calculus-based algorithm produced a solution composed of theoretical

- 15 diameters from a continuous set (CB-Theor), which are then mapped onto discrete commercially available diameters (CB-Disc). Backtracking (BT), a recursive algorithm, systematically searches discrete candidate solutions and is guaranteed to find the global optimum by following rules that prune only higher-cost or hydraulically infeasible candidates. The BT algorithm was modified (BT-NoUp) to improve computational speed by also rejecting all candidates that included a small diameter directly upstream of a larger diameter. This criterion allowed BT-NoUp to prune more candidate solutions but allowed for the
- 20 possibility of missing the global optimum. The third type of algorithm evaluated was a genetic algorithm (GA) that used singlepoint crossover and proportionate selection

BT was able to find the global optimum in all test cases with relatively little computational effort, and could be applied to other GDWNs composed of a similar number of links. In addition, while BT-NoUp completed its search in less time than BT, the time required to complete BT would not be burdensome on a designer and therefore BT-NoUp did not produce a

- 25 compelling relative advantage over BT. BT, however, could become prohibitively time-consuming when dealing with networks with significantly more links, as would be the case with large urban networks. While the test cases represent the range of GDWN sizes encountered in the authors' experience, future work would be needed to verify the suitability of the BT and BT-NoUp algorithms on other large GDWNs. The calculus-based algorithm produced consistently good results for the networks tested, although a more robust mapping scheme from theoretical diameters to discrete diameters would further
- 30 improve on these results as discussed above. In potential future work, the CB-Theor solutions could be used to prune the BT search space, similar to Kadu et al. (2008), by only including the two diameters above and below the CB-Theor diameters,

[revised manuscript text omitted]

| Test Case                 | Туре      | Number of
Diameter
Choices | Number of
Links | Q tot
( L s -1 ) | L tot
( km ) |
|---------------------------|-----------|----------------------------------|--------------------|-------------------------------------------|----------------------------|
| 1. Kiagan, Philippines    | Branching | 8                                | 9                  | 4.37                                      | 0.82                       |
| 2. Los Modulus, Nicaragua | Serial    | 4                                | 13                 | 0.39                                      | 1.24                       |
| 3. Cañazas, Panama        | Branching | 10                               | 23                 | 6.29                                      | 15.2                       |
| 4. San Miguel, Nicaragua  | Serial    | 9                                | 10                 | 0.40                                      | 1.18                       |
| 5. El Guabo, Nicaragua    | Branching | 12                               | 17                 | 17.7                                      | 4.71                       |

| Test Case                 | Globa | al Optimum | Percentage cost increase from global optimum |          |         |      |  |  |  |  |  |  |
|---------------------------|-------|------------|----------------------------------------------|----------|---------|------|--|--|--|--|--|--|
| Test Case                 |       | BT         | BT-NoUp                                      | CB-Theor | CB-Disc | GA   |  |  |  |  |  |  |
| 1. Kiagan, Philippines    | \$    | 2,331      | 0                                            | -3.16    | 11.3    | 4.80 |  |  |  |  |  |  |
| 2. Los Modulos, Nicaragua | \$    | 1,441      | 2.10                                         | -2.60    | 22.6    | 12.1 |  |  |  |  |  |  |
| 3. Cañazas, Panama        | \$    | 72,190     | 0.35                                         | -5.46    | 17.0    | 20.7 |  |  |  |  |  |  |
| 4. San Miguel, Nicaragua  | \$    | 5,418      | 0                                            | -4.54    | 3.86    | 6.20 |  |  |  |  |  |  |
| 5. El Guabo, Nicaragua    | \$    | 61,445     | 0                                            | -3.16    | 20.2    | 13.3 |  |  |  |  |  |  |

Table 2: Solution Costs.

| ~    | Link          | 1-2   | 2-3   | 3-4   | 4-5   | 5-6   | 2-7   | 3-8   | 4-9   | 5-10  |       |
|------|---------------|-------|-------|-------|-------|-------|-------|-------|-------|-------|-------|
| vorl | Length (m)    | 76    | 113   | 19    | 54    | 75    | 80    | 99    | 170   | 135   |       |
| Vetv | Q (L/s)       | 4.37  | 3.68  | 2.94  | 1.46  | 0.69  | 0.69  | 0.74  | 1.48  | 0.77  |       |
| ~    | $\Delta z(m)$ | 14.0  | 1.0   | 0.0   | 0.0   | -1.0  | 0.0   | -2.0  | 3.0   | 2.0   |       |
|      |               |       |       |       |       |       |       |       |       |       |       |
| SU   | BT            | 3     | 2-1/2 | 2     | 1-1/2 | 1-1/2 | 1     | 1-1⁄4 | 1-1/2 | 1-1⁄4 |       |
| utio | BT-NoUp       | 3     | 2-1/2 | 2     | 1-1/2 | 1-1/2 | 1     | 1-1⁄4 | 1-1/2 | 1-1⁄4 |       |
| soli | CB-Theor      | 2.751 | 2.562 | 2.141 | 1.830 | 1.356 | 1.062 | 1.376 | 1.584 | 1.128 |       |
|      | CB-Disc       | 3     | 3     | 2-1/2 | 2     | 1-1⁄4 | 1-1⁄4 | 1-1⁄4 | 1-1/2 | 1-1⁄4 |       |
|      |               |       |       |       |       |       |       |       |       |       |       |
|      | Node          | 1     | 2     | 3     | 4     | 5     | 6     | 7     | 8     | 9     | 10    |
| ~    | ВТ            | 0     | 13.09 | 11.43 | 10.72 | 8.81  | 7.10  | 7.27  | 7.21  | 7.57  | 7.58  |
| m)   | BT-NoUp       | 0     | 13.09 | 11.43 | 10.72 | 8.81  | 7.10  | 7.27  | 7.21  | 7.57  | 7.58  |
| -    | CB-Theor      | 0     | 12.48 | 11.24 | 10.65 | 9.61  | 7.00  | 6.99  | 7.00  | 7.00  | 3.19  |
|      | CB-Disc       | 0     | 13.09 | 13.15 | 12.85 | 12.27 | 9.78  | 11.51 | 8.94  | 9.70  | 11.04 |
|      | -             |       |       |       |       |       |       |       |       |       |       |

| × | Link          | 1-2   | 2-3   | 3-4   | 4-5   | 5-6   | 6-7   | 7-8   | 8-9   | 9-10  | 10-11 | 11-12 | 12-13 | 13-14 |       |
|----------|---------------|-------|-------|-------|-------|-------|-------|-------|-------|-------|-------|-------|-------|-------|-------|
| vor      | Length (m)    | 60    | 41    | 108   | 46    | 134   | 153   | 79    | 157   | 90    | 32    | 102   | 120   | 117   |       |
| Vetv     | Q (L/s)       | 0.39  | 0.39  | 0.39  | 0.39  | 0.39  | 0.39  | 0.39  | 0.39  | 0.39  | 0.39  | 0.39  | 0.39  | 0.39  |       |
| 4        | $\Delta z(m)$ | 11.2  | -0.5  | 32.8  | -3.7  | 36.6  | -2.3  | 15.7  | -6.8  | 7.3   | -7.4  | 4.5   | -1.2  | 8.4   |       |
|          |               |       |       |       |       |       |       |       |       |       |       |       |       |       |       |
| su       | BT            | 1     | 1     | 3⁄4   | 3⁄4   | 3⁄4   | 3⁄4   | 1     | 3⁄4   | 1     | 1     | 3⁄4   | 3⁄4   | 3⁄4   |       |
| utio     | BT-NoUp       | 1     | 1     | 1     | 1     | 1     | 3⁄4   | 3⁄4   | 3⁄4   | 3⁄4   | 3⁄4   | 3⁄4   | 3⁄4   | 3⁄4   |       |
| solı     | CB-Theor      | 0.987 | 0.984 | 0.849 | 0.849 | 0.849 | 0.849 | 0.849 | 0.849 | 0.849 | 0.849 | 0.849 | 0.849 | 0.849 |       |
| Ω        | CB-Disc       | 1     | 1     | 1     | 1     | 1     | 1     | 1     | 1     | 1     | 1     | 1     | 1     | 1     |       |
|          |               |       |       |       |       |       |       |       |       |       |       |       |       |       |       |
|          | Node          | 1     | 2     | 3     | 4     | 5     | 6     | 7     | 8     | 9     | 10    | 11    | 12    | 13    | 14    |
|          | ВТ            | 0     | 9.55  | 7.94  | 31.59 | 24.00 | 49.25 | 33.99 | 47.55 | 27.45 | 32.32 | 24.05 | 19.91 | 8.55  | 7.04  |
| E (B     | BT-NoUp       | 0     | 9.55  | 7.94  | 37.82 | 32.88 | 65.85 | 50.59 | 59.60 | 39.50 | 39.18 | 29.07 | 24.93 | 13.56 | 12.05 |
| Ч        | CB-Theor      | 0     | 9.00  | 7.00  | 31.86 | 24.78 | 51.53 | 37.98 | 47.87 | 29.53 | 30.21 | 20.46 | 17.46 | 7.44  | 7.23  |
|          | CB-Disc       | 0     | 9.55  | 7.94  | 37.82 | 32.88 | 65.85 | 59.41 | 72.98 | 61.93 | 66.80 | 58.53 | 60.27 | 55.83 | 61.06 |
|          |               |       |       |       |       |       |       |       |       |       |       |       |       |       |       |

Table 3(c): Case #3 Cañazas network properties, diameter (D) results (inch nominal sizes, with CB-Theor in inches), and nodal h.

| ×    | Link          | 1-2   | 2-3   | 3-4   | 4-5   | 5-6   | 6-7   | 7-8   | 8-9   | 9-10  | 10-11 | 11-12 | 12-13 | 2-14  | 3-15  | 4-16  | 5-17  | 6-18  | 7-19  | 8-20  | 9-21  | 10-22 | 11-23 | 12-24 |      |
|------|---------------|-------|-------|-------|-------|-------|-------|-------|-------|-------|-------|-------|-------|-------|-------|-------|-------|-------|-------|-------|-------|-------|-------|-------|------|
| VOL  | Length (m)    | 646   | 275   | 957   | 509   | 1102  | 291   | 1764  | 1256  | 2320  | 1580  | 2170  | 1217  | 160   | 100   | 1250  | 110   | 570   | 180   | 1400  | 50    | 400   | 260   | 100   |      |
| Vetv | Q (L/s)       | 6.29  | 5.49  | 5.39  | 5.34  | 5.14  | 2.84  | 2.74  | 2.49  | 2.39  | 0.69  | 0.39  | 0.20  | 0.80  | 0.10  | 0.05  | 0.20  | 2.30  | 0.10  | 0.25  | 0.10  | 1.70  | 0.30  | 0.19  |      |
| ~    | $\Delta z(m)$ | 25.0  | 38.9  | 11.9  | 42.1  | -22.9 | 32.3  | -29.9 | 40.8  | -3.0  | -14.7 | 34.1  | -7.6  | -5.0  | 20.0  | -15.0 | 2.0   | -12.0 | 14.0  | -6.0  | 5.0   | -1.0  | -13.0 | 9.0   |      |
| us   | BT            | 4     | 3     | 3     | 4     | 3     | 3     | 3     | 2-1/2 | 2-1/2 | 2     | 1-1⁄4 | 1     | 1-1⁄4 | 1⁄2   | 1⁄2   | 1⁄2   | 2     | 1/2   | 1     | 1⁄2   | 1-1/2 | 1-1⁄4 | 1⁄2   |      |
| utio | BT-NoUp       | 4     | 4     | 4     | 4     | 3     | 3     | 2-1/2 | 2-1/2 | 2-1/2 | 2     | 1-1⁄4 | 1     | 1-1⁄4 | 1⁄2   | 1⁄2   | 1⁄2   | 1-1/2 | 1⁄2   | 1     | 1⁄2   | 1-1⁄2 | 1-1/2 | 1⁄2   |      |
| solı | CB-Theor      | 3.530 | 3.531 | 3.333 | 3.307 | 3.270 | 2.727 | 2.698 | 2.579 | 2.548 | 1.862 | 1.227 | 1.011 | 1.283 | 0.325 | 0.508 | 0.404 | 1.678 | 0.343 | 0.963 | 0.281 | 1.405 | 1.401 | 0.488 |      |
| D    | CB-Disc       | 4     | 4     | 4     | 4     | 4     | 3     | 3     | 3     | 3     | 2     | 1-1⁄4 | 1     | 1-1⁄4 | 1⁄2   | 1⁄2   | 1⁄2   | 2     | 1⁄2   | 1     | 1⁄2   | 1-1/2 | 1-1/2 | 1⁄2   |      |
|      | Node          | 1     | 2     | 3     | 4     | 5     | б     | 7     | 8     | 9     | 10    | 11    | 12    | 13    | 14    | 15    | 16    | 17    | 18    | 19    | 20    | 21    | 22    | 23    | 24   |
|      | BT            | 0     | 21.1  | 55.4  | 51.5  | 91.3  | 51.7  | 82.5  | 43.9  | 69.8  | 41.4  | 22.1  | 40.0  | 21.7  | 12.1  | 72.2  | 24.3  | 82.2  | 26.1  | 90.8  | 20.1  | 73.3  | 21.9  | 7.81  | 39.7 |
| (m   | BT-NoUp       | 0     | 21.1  | 58.8  | 66.4  | 106   | 66.6  | 97.4  | 42.8  | 68.8  | 40.4  | 21.0  | 39.0  | 20.7  | 12.1  | 75.6  | 39.2  | 97.1  | 9.5   | 106   | 19.0  | 72.2  | 20.9  | 7.43  | 38.7 |
| Ч    | CB-Theor      | 0     | 17.8  | 54.3  | 55.6  | 91.9  | 56.7  | 86.3  | 40.3  | 69.0  | 44.1  | 21.9  | 27.9  | 7.64  | 6.99  | 8.02  | 7.70  | 7.98  | 7.72  | 7.79  | 7.70  | 8.18  | 7.68  | 7.69  | 7.71 |
|      | CB-Disc       | 0     | 21.1  | 58.8  | 66.4  | 106   | 78.8  | 110   | 70.9  | 106   | 94.4  | 75.1  | 93.0  | 74.7  | 12.1  | 75.6  | 39.2  | 97.1  | 53.1  | 118   | 47.1  | 110   | 74.9  | 61.4  | 92.7 |

Table 3(d): Case #4 San Miguel network properties, diameter (D) results (inch nominal sizes, with CB-Theor in inches), and nodal h.

| Network | Link
Length (m)
Q (L/s) | 1-2
189
3.60 | 2-3
168
3.60 | 3-4
139
3.60 | 4-5
81
3.60 | 5-6
32
3.60 | 6-7
92
3.60 | 7-8
225
3.60 | 8-9
115
3.60 | 9-10
52.3
3.60 | 10-11
85
3.60 |       |
|---------|-------------------------------|--------------------|--------------------|--------------------|-------------------|-------------------|-------------------|--------------------|--------------------|----------------------|---------------------|-------|
|         | $\Delta z(m)$                 | 27.4               | 10.7               | -6.4               | 6.1               | -5.2              | -18.6             | 33.2               | 58.2               | -11.3                | 32.9                |       |
| su      | BT                            | 3                  | 3                  | 3                  | 3                 | 3                 | 2-1/2             | 2                  | 1-1⁄4              | 1-1⁄4                | 1-1⁄4               |       |
| utio    | BT-NoUp                       | 3                  | 3                  | 3                  | 3                 | 3                 | 2-1/2             | 2                  | 1-1⁄4              | 1-1⁄4                | 1-1⁄4               |       |
| solı    | CB-Theor                      | 2.939              | 2.929              | 2.929              | 2.929             | 2.929             | 2.929             | 1.671              | 1.462              | 1.462                | 1.368               |       |
| Ω       | CB-Disc                       | 3                  | 3                  | 3                  | 3                 | 3                 | 3                 | 2                  | 1-1/2              | 1-1/2                | 1-1⁄4               |       |
|         |                               |                    |                    |                    |                   |                   |                   |                    |                    |                      |                     |       |
|         | Node                          | 1                  | 2                  | 3                  | 4                 | 5                 | 6                 | 7                  | 8                  | 9                    | 10                  | 11    |
|         | BT                            | 0                  | 25.88              | 35.20              | 27.68             | 33.13             | 27.70             | 7.02               | 28.27              | 43.86                | 13.19               | 14.60 |
| (B      | BT-NoUp                       | 0                  | 25.88              | 35.20              | 27.68             | 33.13             | 27.70             | 7.02               | 28.27              | 43.86                | 13.19               | 14.60 |
| Ч       | CB-Theor                      | 0                  | 25.53              | 34.51              | 26.72             | 32.01             | 26.51             | 7.00               | 6.99               | 32.93                | 6.96                | 7.02  |
|         | CB-Disc                       | 0                  | 25.88              | 35.20              | 27.68             | 33.13             | 27.70             | 8.37               | 29.62              | 67.54                | 47.02               | 48.43 |

| 4        | Link          | 1-2   | 2-3   | 3-4   | 4-5   | 5-6   | 6-7   | 7-8   | 8-9   | 9-10  | 2-11  | 3-12  | 4-13  | 5-14  | 6-15  | 7-16  | 8-17  | 9-18  |       |
|----------|---------------|-------|-------|-------|-------|-------|-------|-------|-------|-------|-------|-------|-------|-------|-------|-------|-------|-------|-------|
| vor      | Length (m)    | 383   | 486   | 1030  | 600   | 150   | 400   | 187   | 450   | 227   | 230   | 240   | 110   | 270   | 130   | 130   | 260   | 110   |       |
| Vetv     | Q (L/s)       | 17.72 | 14.68 | 12.76 | 11.96 | 10.04 | 7.72  | 6.60  | 3.12  | 1.20  | 3.04  | 1.92  | 0.80  | 1.92  | 2.32  | 1.12  | 3.48  | 1.92  |       |
| ~        | $\Delta z(m)$ | 10.9  | 10.0  | -5.6  | 3.2   | -2.6  | 5.7   | -4.1  | 4.2   | -3.1  | 2.0   | 2.5   | -1.2  | 2.0   | -1.1  | 0.0   | 1.0   | 2.0   |       |
|          | l             |       |       |       |       |       |       |       |       |       |       |       |       |       |       |       |       |       |       |
| su       | BT            | 8     | 6     | 6     | 6     | 6     | 5     | 5     | 4     | 2     | 2-1/2 | 1-1/2 | 1-1/2 | 2     | 3     | 1-1⁄4 | 3     | 1-1/2 |       |
| utic     | BT-NoUp       | 8     | 6     | 6     | 6     | 6     | 5     | 5     | 4     | 2     | 2-1/2 | 1-1/2 | 1-1/2 | 2     | 3     | 1-1⁄4 | 3     | 1-1/2 |       |
| solı     | CB-Theor      | 6.875 | 6.408 | 6.144 | 6.008 | 5.691 | 4.800 | 4.576 | 3.494 | 2.649 | 2.364 | 1.608 | 1.529 | 1.932 | 3.250 | 1.395 | 3.076 | 1.647 |       |
| Ω        | CB-Disc       | 8     | 8     | 8     | 6     | 6     | 5     | 5     | 4     | 3     | 2-1/2 | 1-1/2 | 1-1/2 | 2     | 4     | 1-1/2 | 4     | 2     |       |
|          |               |       |       |       |       |       |       |       |       |       |       |       |       |       |       |       |       |       |       |
|          | Node          | 1     | 2     | 3     | 4     | 5     | 6     | 7     | 8     | 9     | 10    | 11    | 12    | 13    | 14    | 15    | 16    | 17    | 18    |
|          | BT            | 0     | 10.34 | 18.50 | 9.91  | 11.53 | 8.61  | 13.16 | 8.65  | 12.11 | 7.25  | 8.48  | 7.23  | 7.35  | 8.84  | 7.03  | 7.16  | 7.68  | 7.80  |
|   | BT-NoUp       | 0     | 10.34 | 18.50 | 9.91  | 11.53 | 8.61  | 13.16 | 8.65  | 12.11 | 7.25  | 8.48  | 7.23  | 7.35  | 8.84  | 7.03  | 7.16  | 7.68  | 7.80  |
| д        | CB-Theor      | 0     | 9.76  | 18.35 | 9.93  | 11.49 | 8.46  | 12.70 | 7.94  | 10.67 | 7.00  | 7.00  | 7.00  | 7.00  | 7.01  | 7.00  | 7.00  | 7.00  | 7.01  |
|          | CB-Disc       | 0     | 10.34 | 19.85 | 13.47 | 15.09 | 12.17 | 16.72 | 12.21 | 15.67 | 12.27 | 8.48  | 8.58  | 10.91 | 12.40 | 10.94 | 13.84 | 12.67 | 15.76 |

Figure 1: Element schematic of a GDWN.

---

## Author Comment (AC3) · 28 May 2017

**Review Report**

**Paper Title: Algorithm for Optimization of Branching Gravity-Distribution Networks**

**Author: Ian Dardani, G. F. Jones**

Several algorithms are available in literature for optimal design of branched gravity-fed WDNs. Authors have chosen two out of them, a back tracking (BT) algorithm and a genetic algorithm (GA) without giving any proper justification of their selection. The chosen algorithms have been compared with a new calculus-based algorithm. I would like to suggest authors to provide advantage of proposed CB with those available in literature (Deb 1974, Bhave 1978, Chiplunkar and Khanna 1983, Fujiwara and Dey 1988, Young 1994, Johnson et al. 1996). Bhave's (1978) approach is general and applicable to branched as well as looped networks, gravity as well as pumped source networks, and new as well as existing networks. In case of looped networks, primary pipes forming a branching configuration is identified and designed to carry maximum flows by considering secondary loop-forming links of some minimum size.

Using calculus based approach, Bhave (1978) developed an optimal criterion similar to Eq. (9) of authors and expressed as
{EQUATION FROM BHAVE – not retyped here.  Refer to the original review.}
Where ij and jk are supply and distribution links at any node j; C and h are cost and head loss in any pipe.

Bhave (1978) suggested a univariant method in which nodal heads are assumed initially and corrected iteratively in order to satisfy the optimal criteria at all nodes. Gupta et al. (2003) improved the method of solution adopting Newton-Raphson method in which all correction values are obtained simultaneously for faster convergence of iterative methodology.

The equation (18) of author seems to be similar to Bhave's optimality criterion, if minor losses are ignored. Authors are requested to clearly point out the difference with Bhave's optimality criteria. Also, a systematic procedure or flow chart should be included to apply the proposed methodology to design water networks.

Even though outcome of the paper is general and nothing new in it, the paper can be recommended if the difference between the proposed CB method with Bhave's CB method is clearly indicated and proposed methodology is explained by giving procedure or flowchart.

**Additional References**

Deb, A. K. (1974). Least Cost Design of Branched Pipe Network Systems. Journal of Environmental Engineering, ASCE, 100(4): 821-835.
Bhave, P. R. (1978). Non-computer optimization of single source networks. Journal of Environmental Engineering, ASCE, 104(4): 799-814.
Chiplunkar, A. V. and Khanna, P. (1983). Least Cost Design of Branched Pipe Network Systemd. Journal of Environmental Engineering, ASCE, 109(3): 604-618.
Fujiwara, O. and Dey, D. (1988). Method of Optimal Design of Branched Network on Flat Terrain. Journal of Environmental Engineering, ASCE, 114(6): 1464-1475.
Young, B. (1994). Design of Branched Water Supply Network on Uneven Terrain. Journal of Environmental Engineering, ASCE, 120(4): 974-979.
Johnson, S. L., Gupta, R. and Bhave, P. R. (1996). Discussion of "Design of Branched Water Supply Network on Uneven Terrain" by B. Young. Journal of Environmental Engineering, ASCE, 122(5): 448.
Gupta, R., Sawarkar, V. R., and Bhave, P. R. (2003). Application of Newton-Raphson method in optimal design of Water Distribution Networks. Journal of Indian Water Works Association, 34(1), 31-37.

Author's Response:
1. Thank you for your detailed response to our paper.
2. As background, the motivation for cost optimization for the Q-specified condition (we refer to this as a demand-driven design), on which all our designs and studies are based, is twofold:
    a. To produce the lowest possible cost for a GDWN that satisfies the minimum-head requirement at all nodes subject to the flow rate constraints
    b. To produce a unique solution to the design problem, which in the absence of cost minimization, produces no unique solution for $D$. This is noted on pp. 2, 3, and 6 in the paper.
3. The problem considered in this paper is that of selecting a single pipe diameter for each link in a water network to optimize the material cost. This problem has three major categories of methods that are applicable to it: enumeration methods (including both complete enumeration and partial enumeration), nonlinear programming methods, and metaheuristic methods. For each of these categories we have proposed and tested one representative algorithm: backtracking (partial enumeration), the Jones calculus-based algorithm (nonlinear programming), and a genetic algorithm (metaheuristic). While there are many other types of metaheuristic algorithms (simulated annealing, Tabu search, cellular automata, ant colony optimization, and particle swarm optimization), the genetic algorithm is the most representative of these and is also the most commonly used (Zhao et al. 2016).

Note that we have categorized the backtracking algorithm as a partial enumeration method and not a heuristic algorithm. The backtracking algorithm, like heuristic methods, follows a set of deterministic rules to find better solutions, however, those rules are strictly formulated to find cost-optimal solutions and does so without missing the global optimum. In contrast, heuristic algorithms follow rules which achieve some proxy of an optimum solution but do not guarantee the global optimum. One example of a heuristic algorithm comes from Suribabu (2012), whose algorithm uses the uniformity of a solution's flow velocity as a proxy for its cost-optimality. As such, the Suribabu algorithm increments a pipe diameter when its flow velocity is high and decrements the diameter when its flow velocity is low, in an attempt to approach more optimal solutions. It was not necessary to include such a heuristic algorithm for comparison in this paper, since the Backtracking algorithm presented follows a more strictly formulated set of rules which do guarantee a cost optimum.

We note that other methods which can be applied to water network design do not address the problem addressed in this paper. For example, linear programming methods provide split-pipe solutions for each link, while the problem addressed here calls for each link to have a single diameter solution. Multi-objective optimization methods, which typically involve new implementations of metaheuristic methods, are also outside the scope of (single-objective) cost optimization, although it should be noted that cost-optimization algorithms, such as the ones used in this paper, can be used within multi-objective implementations. In addition, decomposition methods, where networks are broken down into smaller sub-networks, can use any of the methods listed above, and are therefore not an exclusive method category. Given the appropriateness of the presented algorithms to the network sizes of gravity water networks, decomposition was not necessary.

We have added a more thorough description to our introduction section that gives clearer context to the selection of these algorithms, highlighting the category of method to which they belong and the key features that make them distinct from one another.

Suribabu, C. R.: Heuristic-based pipe dimensioning model for water distribution networks, J. Pipeline Syst. Eng. Pract., 3(4), 115–124, doi:10.1061/(ASCE)PS.1949-1204.0000104, 2012.

Zhao, W., Beach, T., and Rezgui, Y.: Optimization of Potable Water Distribution and Wastewater Collection Networks: A Systematic Review and Future Research Directions, IEEE Transactions on Systems, Man, and Cybernetics: Systems, 46 (5), 659-681, doi:10.1109/TSMC.2015.2461188, 2016.

4. Equations (15) and (18) are the principal results of our CB modeling, **where (18) includes minor losses** and a two-part pipe cost model, which is more robust compared with the one-part pipe cost model of (15). The greater extent of agreement with pipe cost data shown in Figure 3 affirms this. The one-part pipe cost model is used as a simple case to help with reader understanding of the CB results.

5. At the request of the reviewer, we have the following observations when comparing Equation (18) and, in its more-restrictive form, Equation (15) to the Bhave equation above.

    a. The fundamental difference of the Bhave approach compared with ours is that Bhave uses an iterative method to solve the design problem and we do not. As such, there are several qualifications leading up to the Bhave equation (as above) in his paper. Among them is that variations in $H_j$ between two iterations is small so that the terms in the cost function may be approximated as constants (Equation (13) in Bhave). This apparently also includes the cost exponent $m$ in the Bhave paper, where normally $m = m(D)$. Upon taking the derivative of Equation (15) in the Bhave paper, $m$ must be assumed constant or else there would result an additive term to each side of Equation (16) in Bhave that is $H_j \ln(H_j)$ $dm/dH_j$ to account for the $m$-dependence on $H_j$. This term does not appear. Thus, we reason this is because $\ln(H_j)_{new} - \ln(H_j)_{old}$ between two successive iterations is taken to be zero $((H_j)_{new}/(H_j)_{old}=1)$. Our method uses our Equations (15) or (18) written at all internal nodes and includes this equation along with the energy equation for each link to solve for the head at each internal node and all link diameters that collectively minimize network cost. The constraint of $h_j \geq h_{min}$ at each node is, of course, included in the solution. Algorithms to solve a general set of independent, nonlinear algebraic equations using, for example, the Levenberg-Marquardt, Quasi-Newton, Newton-Raphson, or Conjugate Gradient methods are available in most commercial math packages including Matlab and Mathcad (we use the latter with the Quasi-Newton method). In particular, the cost-function coefficient and exponent ($b$ in our paper) are not assumed constant at any node joining two sets of links; see our Equation (18). Nor do we make any assumptions on the orders of magnitude of the terms in our equations to simplify them. In Equation (18) in our paper, the cost-function derivative (Equation (21)), term $C'$, explicitly accounts for the variation in $b$. A continuous function (a cubic polynomial; the smallest possible order) is fit over the transition between the small-$D$ and large-$D$ regions of the pipe cost function. Because there is no iterative procedure, there is no flow chart to illustrate our solution. However, based on your comments we have added a few sentences describing the difference between our approach and that of Bhave and a new figure to show the notation $ij$ and $jk$ at a node. Also included is a more-complete description of our method of solution. We believe these additions have added to the level of quality of our paper. Thank you.

    b. Because of the lack of restrictions of Equations (15) and (18) in our paper, the CB algorithm can be applied to any pipe network including serial, branch, and loop. For serial networks, in particular, where the flow rate across a node is constant, one can clearly see from inspection of Equation (15) the optimal solution of a constant hydraulic gradient ($S_{ij}$) or $D_{ij} = D_{jk} =$ constant for any two links provided $h_j \geq h_{min}$.

    c. The development of Equation (16) in Bhave is not explicitly given, as the steps between his Equations (15) and (16) are not shown in his paper. Conversely, all steps in the development of our Equation (15) appear in our paper, so readers are unambiguously informed of the origin of this equation and its variant, Equation (18).

[revised manuscript text omitted]
} \\[2mm] c_1 + c_2 \dfrac{D_{ij}}{D_u} + c_3 \left(\dfrac{D_{ij}}{D_u}\right)^2 + c_4 \left(\dfrac{D_{ij}}{D_u}\right)^3, & D_{co} < D_{ij} < D_{co+1} \\[2mm] a_\ell (\dfrac{D_{ij}}{D_u})^{b_\ell}, & D_{ij} \geq D_{co+1} \end{cases} \tag{17}$$

5    In Eq. (17), $a_s$ and $a_\ell$ are the coefficients for the small and large pipe size regions, respectively, and $b_s$ and $b_\ell$ are the exponents for the small and large pipe size regions, respectively. A cubic spline is fit between pipe sizes $D_{co}$ and $D_{co+1}$ to complete the transition between small and large pipe sizes. The coefficients of this polynomial are $c_1$, $c_2$, $c_3$, and $c_4$ as seen in Eq. (17). These coefficients are evaluated by matching the cubic polynomial and pipe data at $D_{co}$ and $D_{co+1}$ and the first derivative of the polynomial with respect to $D_{ij}/D_u$ to $a_s b_s (\frac{D_{co}}{D_u})^{b_s-1}$ at $D_{ij} = D_{co}$ and to $a_\ell b_\ell (\frac{D_{co+1}}{D_u})^{b_\ell-1}$ at $D_{ij} = D_{co+1}$. An

10    example of data for Polyvinyl chloride (PVC) pipe and the curvefit is shown in Fig. 3. The results of the curvefit are: $D_{co} = 2.067$ in., $D_{co+1} = 2.469$ in., $a_s = \$1.349$ m$^{-1}$, $b_s = 1.157$, $a_\ell = \$1.381$ m$^{-1}$, $b_\ell = 1.344$, $c_1 = \$237.516$ m$^{-1}$, $c_2 = -\$316.125$ m$^{-1}$, $c_3 = \$140.450$ m$^{-1}$, $c_4 = -\$20.499$ m$^{-1}$. It is clear from inspection of Fig. 3 that a one-part cost model would not have produced an acceptable curve-fit to pipe-cost data.

With the inclusion of the two-part cost model and minor loss term, Eq. (15) becomes

$$0 = \sum_{ij,in} \frac{C'_{ij} A_{ij}^{\frac{4}{19}} (1+\epsilon_{ij})^{\frac{4}{19}} S_{ij}^{-\frac{23}{19}} \left(\frac{Q_{ij}^7 v}{g^4 D_u^{19}}\right)^{\frac{1}{19}}}{1 - B A_{ij}^{\frac{4}{19}} \epsilon_{ij}' (1+\epsilon_{ij})^{-\frac{15}{19}} S_{ij}^{-\frac{4}{19}} \left(\frac{Q_{ij}^7 v}{g^4 D_u^{19}}\right)^{\frac{1}{19}}}$$
$$- \sum_{ij,out} \frac{C'_{ij} A_{ij}^{\frac{4}{19}} (1+\epsilon_{ij})^{\frac{4}{19}} S_{ij}^{-\frac{23}{19}} \left(\frac{Q_{ij}^7 v}{g^4 D_u^{19}}\right)^{1/19}}{1 - B A_{ij}^{\frac{4}{19}} \epsilon_{ij}' (1+\epsilon_{ij})^{-\frac{15}{19}} S_{ij}^{-\frac{4}{19}} \left(\frac{Q_{ij}^7 v}{g^4 D_u^{19}}\right)^{1/19}} \tag{18}$$

15    where $B = 0.1989$ and

$$\epsilon_{ij} = \sum_k \left(\frac{L_e}{D}\right)_{k,ij} \frac{D_{ij}}{L_{ij}} \tag{19}$$

$$\epsilon_{ij}' = \sum_k \left(\frac{L_e}{D}\right)_{k,ij} \frac{D_u}{L_{ij}}$$

$$A_{ij} = \begin{cases} 0.318, & \text{smooth pipe} \\ 0.420, & \text{steel pipe} \end{cases} \tag{20}$$

and $A$ accounts for the effect of pipe roughness (smooth and commercial steel). The term $C'_{ij}$ is the derivative of the cost function per unit length with respect to $D/D_u$. For the two-part cost model from above, obtain

$$C'_{ij} = \begin{cases} a_s b_s (\frac{D_{ij}}{D_u})^{b_s-1}, & D_{ij} \leq D_{co} \\ c_2 + 2c_3(\frac{D_{ij}}{D_u}) + 3c_4(\frac{D_{ij}}{D_u})^2, & D_{co} < D_{ij} < D_{co+1} \\ a_\ell b_\ell (\frac{D_{ij}}{D_u})^{b_\ell-1}, & D_{ij} \geq D_{co+1} \end{cases} \tag{21}$$

Equation (18), and its simpler form Eq. (15), forms the basis for calculus-based optimization in this work and is applied at all internal nodes to uniquely determine $h_j$. Equation (18) is valid over the range of ~4000 < Re < ~300,000. Algorithms to solve a general set of independent, nonlinear algebraic equations using, for example, the Levenberg-Marquardt, Quasi-Newton, Newton-Raphson, or Conjugate Gradient methods are available in most commercial math packages including Matlab (1 Apple Hill Drive, Natick, MA USA 01760) and Mathcad (http://www.ptc.com). We used the package Mathcad in the present work. Thus, compared with an iterative solution procedure, a solution flowchart is not relevant here.

Bhave (1978) first proposed an algorithm like Eq. (15). However, Bhave used an iterative method to solve the design problem. As such, there are several qualifications leading up to the cost minimization equation in Bhave. These include the assumption of smallness in variation of the static pressure head between two iterations. This allowed the terms in the cost function to be approximated as constants. In the present work, the cost-function coefficient and exponent are not assumed constant at any node joining two sets of links; see Equation (18). Nor do we make any assumptions on the orders of magnitude of the terms in our equations to simplify them. For clarity, we re-present Eq. (15) using Bhave's (1978) notation as

$$0 = \sum Q_{ij}^{7b/19} S_{ij}^{-(1+4b/19)} - \sum Q_{jk}^{7b/19} S_{jk}^{-(1+4b/19)}$$

where the $ij$ and $jk$ notation are shown in Fig. 4. Index $j$ spans all internal nodes along the distribution main.

[revised manuscript text omitted]

Two backtracking methods can be found in the literature, namely those by Gessler (1985) and González-Cebollada et al. (2011). The algorithm proposed by Gessler, however, also proposes a pipe-grouping criteria that risks pruning the global optimum and represents a tricky optimization problem in and of itself (Raad 2011). The González-Cebollada algorithm, on the other hand, does not include such criteria to potentially prune a global optimum from consideration, but it halts its search after finding the first feasible solution. In contrast, the BT algorithm in the present study guarantees a global optimum by continuing its search of the solution tree even after the first solution has been found. In addition, the present BT algorithm utilizes Pre-Processors 1 and 2 to further reduce the search space, without risk of pruning the global optimum, in advance of its search routine. Thus, out of the two reported backtracking algorithms in the literature, both do not guarantee a global optimum, while the BT algorithm presented in this work does. It should be noted that BT is known to scale poorly with large network sizes and would not be appropriate for use on large urban networks, though its appropriateness is demonstrated here for GDWNs, given that the test cases used in this paper representative of the sizes of GDWNs that would be expected in practice.

[revised manuscript text omitted]
. While a two-point crossover technique was considered, the results were not found to have any benefit over a single-point technique, which was chosen for its greater simplicity. The fitness, $f_i$, of each candidate is assessed with penalties associated with the solution's pipe cost, $C_{pipe,i}$, and violations of the static pressure head requirement, $C_{hyd,i}$, or

$$f_i = \frac{1}{C_{pipe,i} + C_{hyd,i}} \tag{29}$$

The hydraulic cost is obtained for each individual by identifying nodes in which the static pressure head is less than $h_{min}$ and multiplying the total amount of head violation by a hydraulic penalty coefficient, $a_{hyd}$:

$$C_{hyd,i_c} = a_{hyd} \sum_{1}^{N_L} (h_{min} - h_{i_N}) \mid h_{i_N} < h_{min} \tag{30}$$

To allow for a hydraulic penalty coefficient to produce similar results in both small-scale (inexpensive) network and a large-scale (more expensive) cases, the hydraulic penalty coefficient is made directly proportional to the average solution

cost. With each generation, $a_{hyd}$ is updated by multiplying the normalized penalty coefficient, $a_{hyd,norm}$, by the average pipe cost of the population,

$$a_{hyd} = a_{hyd,norm} \frac{\sum_1^{N_c} C_{pipe,i_C}}{N_c} \tag{31}$$

The algorithm then selects candidates to be carried into the next generation through a proportionate selection method, where each candidate has a probability of being selected, $p_{sel,i}$ in direct proportion to its fitness relative to the sum of all fitness values in the population

$$p_{sel,i} = \frac{f_i}{\sum_1^{N_c} f_i} \tag{32}$$

The algorithm replaces the parent generation with a generation of equal size and tends to select more fit individuals in successive generations. In this study, the genetic algorithm parameters used were $p_{mut} = 0.02$, $N_c = 50$, $p_{xover} = 0.5$, $N_{gen} = 100$, $a_{hyd,norm} = 0.1$. The first four of these parameters were chosen based on typical values presented in the literature and then tuned with a sensitivity analysis for the first test case. Simspon et al. (1994) present typical values for $N_c$ (30 - 200), $p_{xover}$ (0.7 - 1.0), $p_m$ (0.01 - 0.05), and $N_{gen}$ (100 - 1000). The normalized hydraulic penalty coefficient, $a_{hyd,norm}$, was chosen such that the GA converged on solutions which tended to satisfy the minimum static pressure constraint, but still allowed the population to gravitate towards smaller diameters with static pressures close to $h_{min}$.

**5  Cases Studied**

Five cases were studied based on actual GDWN in Panama, Nicaragua, and the Philippines. Global characteristics of each network are presented in Table 1 and the details of each network are presented in Table 3(a)-(e). Each network is a branching type without loops. The total lengths of the networks range from less than 1 km to over 15 km. Two serial networks are tested to demonstrate the effect of a local high point on the algorithm solutions. Elevation plots for each case are shown in Fig. 5.

The choice of $h_{min}$ is not standardized, and should appropriately balance the risk of negative pressure in pipes and the increase in network cost due to the requirement of using larger diameters. The choice of $h_{min}$ in GDWN design is typically in the range of 5 m – 20 m (Arnalich 2010; Bouman 2014; Swamee and Sharma 2008). In the present study $h_{min} = 7$ m, although this requirement was reduced at selected nodes at the beginning of networks where changes in elevation are still small. At the source node, the static pressure head is fixed at atmospheric pressure. All cases assumed minor-lossless flow, although all algorithms (e.g., Eq. (18) for CB-Theor) are capable of handling minor loss coefficients through the equivalent length method as presented above.

**6 Mapping the Theoretical $D$ to Discrete Pipe Sizes**

The mapping between continuous diameters and the discrete nominal pipe sizes was accomplished in our solution by one of the following ways:

1. For small and moderate size networks, the designer may manually adjust the pipe sizes (downward, normally one pipe size) starting from the first link downstream from the source and continuing along the rest of the distribution main to the end. A nearby plot of the static pressure heads compared with the theoretical $D_{ij}$ from our CB approach (on the same Mathcad page) will highlight the acceptability or unacceptability of any change. This exercise also gives the designer an understanding of the sensitivity of the design to small changes in pipe sizes.

2. Based on the theoretical $D_{ij}$ from the CB approach, a composite pipeline can be created for each link. That is, the lengths for the two discrete pipes sizes that bound the theoretical $D_{ij}$ from above and below are calculated such that the pressure drop between two consecutive nodes in the distribution main matches between the composite pipeline and the CB approach. This also provides discrete pipe sizes that nearly matches the CB solution in terms of cost.

[revised manuscript text omitted]
. 1996) were attempted in a less systematic way (Ie. a parameter tuning study and three attempts at scaling the fitness function with an increasing exponent), but these did not result in a noticeable on performance. However, it is possible that if these techniques were followed systematically in full, the GA performance may have been improved. Still, the GA algorithm presented has undergone

10   reasonable attempts to adapt its design and parameters for real-world GDWN cases, and therefore presents a useful point of comparison to the BT and CB algorithms.

**8   Conclusions**

Algorithms to optimize the cost of branching gravity-driven water networks are evaluated on five test cases from real networks in the Philippines, Nicaragua, and Panama. A calculus-based algorithm produced a solution composed of theoretical

15   diameters from a continuous set (CB-Theor), which are then mapped onto discrete commercially available diameters (CB-Disc). Backtracking (BT), a recursive algorithm, systematically searches discrete candidate solutions and is guaranteed to find the global optimum by following rules that prune only higher-cost or hydraulically infeasible candidates. The BT algorithm was modified (BT-NoUp) to improve computational speed by also rejecting all candidates that included a small diameter directly upstream of a larger diameter. This criterion allowed BT-NoUp to prune more candidate solutions but allowed for the

20   possibility of missing the global optimum. The third type of algorithm evaluated was a genetic algorithm (GA) that used single-point crossover and proportionate selection

BT was able to find the global optimum in all test cases with relatively little computational effort, and could be applied to other GDWNs composed of a similar number of links. In addition, while BT-NoUp completed its search in less time than BT, the time required to complete BT would not be burdensome on a designer and therefore BT-NoUp did not produce a

25   compelling relative advantage over BT. BT, however, could become prohibitively time-consuming when dealing with networks with significantly more links, as would be the case with large urban networks. While the test cases represent the range of GDWN sizes encountered in the authors' experience, future work would be needed to verify the suitability of the BT and BT-NoUp algorithms on other large GDWNs. The calculus-based algorithm produced consistently good results for the networks tested, although a more robust mapping scheme from theoretical diameters to discrete diameters would further

30   improve on these results as discussed above. In potential future work, the CB-Theor solutions could be used to prune the BT search space, similar to Kadu et al. (2008), by only including the two diameters above and below the CB-Theor diameters,

producing four diameter choices per link. The calculus-based methodology provides an additional benefit to the designer by explicitly revealing the sensitivities to cost for a design. The calculus-based algorithm requires greater computational effort than backtracking for smaller networks, however, this effort scales more linearly with the number of network links, while backtracking scales exponentially. Furthermore, backtracking's computational time is sensitive to the number of available

5    diameters. Still, when applied to GDWNs with a similar number of links to the test cases, backtracking can quickly find a global optimum. In addition, because it is guaranteed to find the global optimum, it can be useful for benchmarking the performance of other algorithms which scale better with more network links. While the genetic algorithm produced solutions with decent closeness to the global optimum, run-to-run results vary due to the stochastic nature of the algorithm. Overall, the genetic algorithm as implemented did not produce results which deemed it compelling over deterministic methods as applied

10   to GDWNs.  However, for more complex networks and problem formulations, a genetic algorithm may be more advantageous. In this case, the present study's GA could be greatly improved on through many improvements reported in the literature (Nicklow et al. 2010).

For all test cases, the calculus-based algorithm's theoretical diameter solutions (CB-Theor) produced a lower cost than the discrete-domain global optimum. This result is made possible because of it is not constrained to a discrete set of

15   diameters. As such, the *CB-Theor results represent a lower-bound on the optimum solution within the problem formulation*, which could be approached with a finer selection of pipe diameters.

[revised manuscript text omitted]

**Table 3(e): Case #5 El Guabo network properties, diameter (D) results (inch nominal sizes, with CB-Theor in inches), and nodal h.**

| | Link | 1-2 | 2-3 | 3-4 | 4-5 | 5-6 | 6-7 | 7-8 | 8-9 | 9-10 | 2-11 | 3-12 | 4-13 | 5-14 | 6-15 | 7-16 | 8-17 | 9-18 | |
|---|---|---|---|---|---|---|---|---|---|---|---|---|---|---|---|---|---|---|---|
| Network | Length (m) | 383 | 486 | 1030 | 600 | 150 | 400 | 187 | 450 | 227 | 230 | 240 | 110 | 270 | 130 | 130 | 260 | 110 | |
| | Q (L/s) | 17.72 | 14.68 | 12.76 | 11.96 | 10.04 | 7.72 | 6.60 | 3.12 | 1.20 | 3.04 | 1.92 | 0.80 | 1.92 | 2.32 | 1.12 | 3.48 | 1.92 | |
| | $\Delta z$ (m) | 10.9 | 10.0 | -5.6 | 3.2 | -2.6 | 5.7 | -4.1 | 4.2 | -3.1 | 2.0 | 2.5 | -1.2 | 2.0 | -1.1 | 0.0 | 1.0 | 2.0 | |
| D solutions | BT | 8 | 6 | 6 | 6 | 6 | 5 | 5 | 4 | 2 | 2-½ | 1-½ | 1-½ | 2 | 3 | 1-¼ | 3 | 1-½ | |
| | BT-NoUp | 8 | 6 | 6 | 6 | 6 | 5 | 5 | 4 | 2 | 2-½ | 1-½ | 1-½ | 2 | 3 | 1-¼ | 3 | 1-½ | |
| | CB-Theor | 6.875 | 6.408 | 6.144 | 6.008 | 5.691 | 4.800 | 4.576 | 3.494 | 2.649 | 2.364 | 1.608 | 1.529 | 1.932 | 3.250 | 1.395 | 3.076 | 1.647 | |
| | CB-Disc | 8 | 8 | 8 | 6 | 6 | 5 | 5 | 4 | 3 | 2-½ | 1-½ | 1-½ | 2 | 4 | 1-½ | 4 | 2 | |
| | Node | 1 | 2 | 3 | 4 | 5 | 6 | 7 | 8 | 9 | 10 | 11 | 12 | 13 | 14 | 15 | 16 | 17 | 18 |
| h (m) | BT | 0 | 10.34 | 18.50 | 9.91 | 11.53 | 8.61 | 13.16 | 8.65 | 12.11 | 7.25 | 8.48 | 7.23 | 7.35 | 8.84 | 7.03 | 7.16 | 7.68 | 7.80 |
| | BT-NoUp | 0 | 10.34 | 18.50 | 9.91 | 11.53 | 8.61 | 13.16 | 8.65 | 12.11 | 7.25 | 8.48 | 7.23 | 7.35 | 8.84 | 7.03 | 7.16 | 7.68 | 7.80 |
| | CB-Theor | 0 | 9.76 | 18.35 | 9.93 | 11.49 | 8.46 | 12.70 | 7.94 | 10.67 | 7.00 | 7.00 | 7.00 | 7.00 | 7.01 | 7.00 | 7.00 | 7.00 | 7.01 |
| | CB-Disc | 0 | 10.34 | 19.85 | 13.47 | 15.09 | 12.17 | 16.72 | 12.21 | 15.67 | 12.27 | 8.48 | 8.58 | 10.91 | 12.40 | 10.94 | 13.84 | 12.67 | 15.76 |

[Figure]

**Figure 1:  Element schematic of a GDWN.**

[Figure]

**Figure 2: Three-pipe branch network.**

[Figure]

**Figure 3:  PVC pipe cost from 2011 data.**

[Figure]

**Figure 4. Bhave (1978) index notation at an internal node, *j*.**

[Figure]

**Figure 5: Network elevation (z) and hydraulic grade lines (HGLs) of algorithm solution for main distribution links.**

[Figure]

**Figure 6: Diameter sizes of calculus-based (CB-Disc) solutions above the global optimum solutions.**

---

## Author Comment (AC4) · 7 Jun 2017

We have the following responses and questions concerning the requested revisions:

1. Authors have incorporated some of the suggested changes. However, many of the rebuttals to the authors comments needs to be suitably incorporated in the manuscript.

Author's Response: We are now reviewing the paper for places where our responses to the reviewer's comments might be incorporated.

2. Manuscript has many loose ends. It is still looks like a first draft.

Author's Response: Before we address this request, we will need guidance as to where in the paper the loose ends appear. Clearly, a general statement such as this needs to

be supported by details before any actions can be taken.

3. Authors should clearly bring out the quantifiable difference with the Bhave's method.

Author's Response: Quantitative differences between the cost-minimization function in our paper and the one from Bhave are not the issue. It was never an intent of our paper to compare the two or encourage the use of one method over the other. Ours is simply an alternate equation, based on Taylor series fundamentals applied to a simple branch network, and embedded in an alternate method. We have proved that it works because the fundamentals are correct, and graphs of cost vs. nodal pressure heads around the optimal head have demonstrated this. The real issue is the basis for the Bhave equation vs. that for ours. Each method needs to stand on its own merit, which comes from how each was developed. We have made the development of our equation as clear as possible by explaining all assumptions and showing nearly every step in its development. In addition, we have explained in detail the differences that we see between the two, including the iterative method of Bhave vs. the use of a math-package-based nonlinear equation solver in ours. If you require further evidence of the correctness of our method, please advise us and we will be happy to include this in the paper.

4. Detailed description of different methods may be presented in tabular form. It would result in replacing many paras with a table.

Author's Response: We are unclear as to what this means. Perhaps the reviewer who requested this could point to a published example so we can determine the exact meaning.

Thank you.

---

## Author Comment (AC5) · 16 Jun 2017

We uploaded our comments to the Topical Ed on June 7 and have not yet heard a response. Please let us know the status of these comments.

Thank you.
* * *